# Geomorphic signatures of the transient fluvial response to tilting

Helen W. Beeson[1] and Scott W. McCoy[1]

[1]Department of Geological Sciences and Engineering, University of Nevada, Reno, NV, 89557 USA

**Correspondence:** Helen W. Beeson (hbeeson@nevada.unr.edu)

**Abstract.** Nonuniform rock uplift in the form of tilting has been documented in convergent margins, postorogenic landscapes, and extensional provinces. Despite the prevalence of tilting, the transient fluvial response to tilting has not been quantified such that tectonic histories involving tilt can be extracted from river network forms. We used numerical landscape evolution models to characterize the transient erosional response of a river network initially at equilibrium to rapid tilting. We focus on the case of punctuated rigid-block tilting, though we explore longer duration tilting events and non-uniform uplift that deviates from perfect rigid-block tilting such as that observed when bending an elastic plate or with more pronounced internal deformation of a fault-bounded block. Using a model river network composed of linked 1-D river longitudinal profile evolution models, we show that the transient response to a punctuated rigid-block tilting event creates a suite of characteristic forms or geomorphic signatures in mainstem and tributary profiles that collectively are distinct from those generated by other perturbations such as a step change in uniform rock uplift rate or major truncation of headwater drainage area that push a river network away from equilibrium. These signatures include 1) a knickpoint in the mainstem that separates a downstream profile with uniform steepness (i.e., channel gradient normalized for drainage area) from an upstream profile with nonuniform steepness, with the mainstem above the knickpoint more out of equilibrium than the tributaries following forward tilting towards the outlet, versus the mainstem less out of equilibrium than the tributaries following back tilting towards the headwaters; 2) a pattern of mainstem incision below paleotopography markers that increases linearly up to the mainstem knickpoint, or vice-versa following back tilting; and 3) tributary knickzones with nonuniform steepness that mirrors that of the mainstem upstream of the slope-break knickpoint.

Immediately after a punctuated tilting event, knickpoints form at the mainstem outlet and each mainstem-tributary junction. Time since the cessation of rapid tilting is recorded by mainstem knickpoint location relative to base level and by the upstream end of tributary knickzones relative to tributary-mainstem junctions. Tilt magnitude is recorded in the spatial gradient of mainstem incision depth and, in the forward tilting case, also by the spatial gradient in tributary knickzone drop height. Heterogeneous lithology can modulate the transient response to tilting and, post-tilt, knickpoints can form anywhere in a stream network where more erodible rock occurs upstream of less erodible rock. With a full 2-D model, we show that stream segments flowing in the tilt direction have elevated channel gradient early in the transient response. Tilting is also reflected in network topologic changes via stream capture oriented in the direction of tilt. As an example of how these geomorphic signatures can be used in concert to estimate timing and magnitude of a tilting event, we show a sample of rivers from two field sites: the Sierra Nevada, California, USA, and the Sierra San Pedro Mártir, Baja California, Mexico, two ranges thought to have been tilted westward towards river outlets in the late Cenozoic.

# 1 Introduction

In unglaciated mountainous terrain, bedrock rivers are the primary drivers of landscape evolution (Howard, 1994). Bedrock rivers evolve in response to external forcing, including rock uplift rate relative to baselevel and climate, set base level for bounding hillslopes and the steep headwater valley network where debris flows can dominate, define the relief structure of the landscape as it is carved into valleys and ridges, and can transmit changes in external forcing through the river network (Burbank et al., 1996; Howard, 1994; Howard and Kerby, 1983; Ouimet et al., 2009; Snyder et al., 2000; Stock and Dietrich, 2003; Whipple, 2004; Whipple and Tucker, 1999). The channel steepness index, or the rate at which channel slope changes with drainage area (Hack, 1957; Flint, 1974; Morisawa, 1962), of rivers that have reached an equilibrium grade (Mackin, 1948) has been shown to reflect spatial patterns in uplift rate, millennial-scale erosion rates, rock erodibility, and climate (e.g., Bonnett and Crave, 2003; DiBiase et al., 2010; Duvall et al., 2004; Ouimet et al., 2009; Snyder et al., 2000; Wobus et al., 2006). Transient river profiles can record discrete, persistent, or cyclic changes in climate, lithology, relative base level, or drainage area as the river profile adjusts to the changes and evolves towards an equilibrium channel steepness that reflects modern boundary conditions (Whipple, 2013). Knowledge of the transient response of bedrock rivers to different perturbations thus comprises an important geomorphic tool to characterize the history of rock uplift rates, climate, or changes in river network topology from disequilibrium landscape form (e.g., Beeson et al., 2017; Ferrier et al., 2013; Kirby and Whipple, 2012; Lease and Ehlers, 2013; Tucker and Whipple, 2002; Whittaker et al., 2008; Willett et al., 2014; Wobus et al., 2006). Such histories are critical for testing geodynamic models of orogenesis and quantifying the relative importance of external forcing, such as climate and tectonics, versus internal complex system response, on the evolution of mountainous landscapes (e.g., Beeson et al., 2017; Clark et al., 2005; Gallen, 2018; Kirby and Whipple, 2012; Whipple et al., 2017; Willett et al., 2018; Yang et al., 2015).

Previous studies have illustrated the expected transient response in bedrock rivers to step changes in uniform rock erodibility or uplift rate (Baldwin et al., 2003; Bonnet and Crave, 2003; Howard, 1994; Royden and Perron, 2013; Tucker and Whipple, 2002; Whipple and Tucker, 1999), sudden base-level fall or uniform pulses of rock uplift (Grimaud et al., 2016; Rosenbloom and Anderson, 1994; Whipple and Tucker, 1999), erosion through layered stratigraphy (Forte et al., 2016), and cyclic fluctuations in rock erodibility, base level, or uplift rate (Goren et al., 2016; Snyder et al., 2002). Kirby and Whipple (2001) predict that steady-state bedrock rivers adjusted to uplift gradients with maximum uplift either at the channel head or the channel outlet will have increased and decreased concavities, or the rate of change of river slope with distance downstream, respectively. Whittaker et al. (2008) and Attal et al. (2011) explore the transient response to a step increase in nonuniform uplift rate on fault-bounded tilted blocks, but with a primary focus on how different erosion formulations modulate the mainstem response. Despite this progress, well-defined characteristics of the transient response across an entire bedrock river network to nonuniform uplift owing to a punctuated tilting event is still lacking.

Nonuniform rock uplift in the form of tilting has been documented across many tectonic settings. Convergent boundaries where tilting has been documented include the western flank of the central Andes (Farías et al., 2005; Jordan et al., 2010; Lamb and Hoke, 1997; Saylor and Horton, 2014; Wörner et al., 2002), the Siwalik Hills in the foothills of the Himalaya (Delcaillau et

al., 2007; Kirby and Whipple, 2001; Lavé and Avouac, 2000; Singh and Tandon, 2007), and the Manawatu region of northern New Zealand (Jackson et al., 1998). In the postorogenic North American Cordillera, regional tilting has been documented in the Sierra Nevada (Huber 1981; Jones, 2004; Lindgren, 1911; Unruh, 1991; Wakabayashi, 2013; Wakabayashi and Sawyer, 2001), the Rocky Mountains (McMillan et al., 2002; Riihimaki et al., 2007), the Salmon River basin (Mitchell and Yanites, 2019), and the Colorado Plateau (Liu and Gurnis, 2010; Moucha et al., 2009, Moucha et al., 2008; Sahagian et al., 2002). Tilting on a smaller scale has been documented on fault-bounded blocks in extensional terrain in the Appenines (Whittaker et al., 2008 and references therein) and throughout the Basin and Range (Stewart, 1980), including the Teton Range (Byrd et al., 1994), the Wassuk Range (Gorynski et al., 2013), the White Mountains (Stockli et al., 2003), and the Wasatch Range (Armstrong et al., 2003). Although tilting is widely documented, characteristic forms of bedrock rivers during the transient response to tilting have yet to be quantified.

Here, we seek to answer the question: *What are the geomorphic signatures of the transient fluvial response to tilting and can these signatures be used to quantify uplift histories in terms of timing and magnitude?* We focus on tilting towards the river outlet in which the mainstem river network is everywhere steepened by tilting about a horizontal axis located at the river outlet and oriented perpendicular to the mainstem (referred to throughout as forward tilting), though we simulate the transient response to other tilt directions relative to the mainstem flow direction to highlight general patterns. Specifically, we simulated forward, back (i.e., tilting towards the channel head about a horizontal axis located at the river outlet and oriented perpendicular to the mainstem) and lateral tilting (i.e., tilting about a horizontal axis located along the mainstem and oriented parallel to the mainstem) in homogeneous lithology and forward tilting with the simplest case of vertically-bedded heterogeneous lithology. Additionally, we explore perturbations that generate river profiles with similar characteristics to those produced by tilting. Many of these perturbations likely generate depositional signatures as well, but in this paper we focus exclusively on the erosional response in bedrock rivers. Although we focus on the simple case of a single short-duration rigid-block tilting event that briefly increases rock and surface uplift rates well above background, we explore both forward tilting over longer timescales and non-uniform uplift that deviates from perfect rigid-block forward tilting such as that observed when bending an elastic plate or with more pronounced internal deformation of a fault-bounded block. Lastly, we document the expression of geomorphic signatures of a punctuated rigid-block tilting event proposed to have occurred in the Sierra Nevada of California, USA (Huber 1981; Jones, 2004; Lindgren, 1911; Unruh, 1991; Wakabayashi, 2013) and onset of rapid continuous tilting in the Sierra San Pedro Mártir of Baja California (Rossi et al., 2017). We use these field examples to demonstrate how signatures of tilt in river profiles and river networks can be applied to estimate the timing and magnitude of tilt in both these regions, but we stress that in neither case do we consider our analysis to be a robust reconstruction of the regional tectonic histories owing to the analysis of only a single river basin from each landscape.

## 2    Methods

We used 1- and 2-dimensional (1- and 2-D) numerical landscape evolution modeling to explore the transient response of river longitudinal profiles and river networks to various perturbations that move a river away from equilibrium, with particular

emphasis on punctuated rigid-block forward tilting, and then compared these results to topographic analysis of river geomorphology on the western slopes of the Sierra Nevada and the Sierra San Pedro Mártir of Baja California, ranges proposed to have been tilted westward in the late Cenozoic. For the topographic analysis of the Sierra Nevada and the Sierra San Pedro Mártir, we used 3 arc-second (∼90 m) digital elevation models (DEMs) derived from the Shuttle Radar Topography Mission

and downloaded from Open Topography (http://opentopography.org). To map both real and simulated river networks, we calculated flow direction and accumulation using a steepest descent flow algorithm. All topographic analysis on real and simulated DEMs was completed using the Matlab-based software package TopoToolbox (Schwanghart and Scherler, 2014).

## 2.1   Numerical landscape evolution modeling

For all numerical models, we used the stream power model of bedrock incision, which assumes the rate of bedrock incision

is proportional to the stream power per unit channel width and that incision is limited by the rate at which the river can detach bedrock particles, as opposed to the rate at which detached particles can be transported (Howard, 1994; Perron et al., 2008; Siedl and Dietrich, 1992; Whipple, 2004; Whipple and Tucker, 1999). To simulate the evolution of the land surface, we numerically solved the following governing equation using a forward-time upwind-space finite-difference solver:

$$
\begin{aligned}
\frac{\delta z}{\delta t} &= U - K A^m |\nabla z|^n & |\nabla z| \leq S_c \\
\frac{\delta z}{\delta t} &= U - \xi_l & |\nabla z| > S_c
\end{aligned}
\tag{1}
$$

where $z$ is land surface elevation, $t$ is time, $U$ is rock uplift rate relative to base level, $K$ is an erodibility coefficient, $A$ is drainage area, $m$ and $n$ are empirical constants, $S_c$ is a critical gradient above which landsliding is initiated, and $\xi_l$ is the erosion rate required to reduce slopes to $S_c$ across the domain in a single time step. The equation in the bottom row imposes a maximum hillslope gradient and is a simple representation of threshold-controlled landsliding in which rock required to decrease gradient down to $S_c$ is removed from each over-steepened cell through an iterative process during each time step until

the gradient of all grid cells in the domain is less than or equal to $S_c$.

To simulate just the evolution of a river longitudinal profile, the problem is one-dimensional, and equation 1 can be simplified to

$$
\frac{\delta z(x,t)}{\delta t} = U(x,t) - K(x,t)A(x,t)^m \left| \frac{dz(x,t)}{dx} \right|^n
\tag{2}
$$

where $x$ is distance along the river. Equation 2 is only valid some distance down slope, $x_c$, of the drainage divide where

fluvial processes are active. In these 1-D river profile models we assumed drainage area along the channel can be described using Hack's law, $A = k_a x^h$ , where $k_a$ and $h$ are empirical constants (Hack, 1957). Equations 1 and 2 take the form of a nonlinear wave equation with a source term $U$ and thus perturbations to river profile slope move up through a river network in a wave-like manner with celerity, $C$, dependent on erodibility, $K$, and drainage area, $A$, as well as river slope, $S = dz/dx$ in 1-D, if $n$ is not unity such that $C = K A^m S^{n-1}$ (Tucker and Whipple, 2002). We use the terminology presented by Haviv et

al. (2010) and Whipple et al. (2013) to call a point that separates portions of a river profile with dissimilar channel steepness

a 'slope-break knickpoint' and a point at which offset of similar steepness channel profiles occurs a 'vertical-step knickpoint'. We use 'knickzone' to denote a portion of the river profile that has locally high channel steepness.

## 2.2   $\chi$ transformed river profiles for identifying equilibrium and transient forms in river profiles

For rivers in simulated topography and in the Sierra Nevada and Sierra San Pedro Mártir, we calculated the channel length-drainage area scaling relationship, $\chi$, that can be derived by solving equation 2 for steady-state or equilibrium conditions with uniform rock uplift rate, $U$, and rock erodibility, $K$.

$$z(x) = z_b + \left( \frac{U}{K A_0^m} \right)^{\frac{1}{n}} \chi \tag{3}$$

where

$$\chi = \int_{x_b}^{x} \left( \frac{A_0}{A(x')} \right)^{\frac{m}{n}} dx' \tag{4}$$

We define steady-state or equilibrium for a bedrock river as the state in which the time rate of change of river profile elevation is equal to zero, which occurs once river network area and slope adjust such that there is a perfect balance between input of rock by rock uplift relative to base level and removal of rock by erosion. For these conditions, equation 3 shows that river elevation scales linearly with $\chi$. An equilibrium river profile with uniform rock uplift rate and erodibility will be linear on a $\chi$ plot (i.e., a plot of river elevation as a function of $\chi$) rather than concave-up, as it would be in an untransformed longitudinal profile (Perron and Royden, 2013). Thus $\chi$ can be used as a proxy for the steady-state elevation of the river network as well as a convenient transformation variable that removes the effect of basin geometry and the downstream increase in drainage area on river longitudinal profile shape (Perron and Royden, 2013; Willett et al., 2014). Tributaries in equilibrium with the same uplift rate and with the same erodibility as the mainstem will be co-linear with each other as well as with the mainstem such that all portions of an equilibrium river network collapse towards a single straight line on a $\chi$ plot, provided the correct reference concavity ($m/n$) has been chosen and the analysis has been limited to the fluvial portions of the network (Clubb et al., 2014; Perron and Royden, 2013).

Deviations from a linear $\chi$ plot can be used to identify river profiles in a state of transient adjustment in response to changes in rock uplift rate, rock erodibility, or basin geometry that move a river away from equilibrium or deviations from assumed uniformity in uplift and rock erodibility (Perron and Royden, 2013; Willett et al., 2014). $\chi$ plots are particularly useful for identifying transient knickpoints propagating through a river network that share a common origin because these knickpoints will collapse to the same $\chi$ value. In the same manner that transformed profiles remove the effect of downstream increases in drainage area on river longitudinal profile shape, transformed profiles remove drainage area effects on the perturbation travel distances. We exploit these properties of transformed river profiles to identify portions of the river network that are near equilibrium (i.e., linear $\chi$ plots), versus out of equilibrium (i.e., nonlinear $\chi$ plots), as well as to test whether transient signals have a common origin (transient signals located at the same point in $\chi$ space).

With $A_0 = 1$, the coefficient in front of the integral quantity $\chi$ in equation 3 is the channel steepness, $k_s$ (Perron and Royden, 2013):

$$k_s = \left(\frac{U}{K}\right)^{\frac{1}{n}} \tag{5}$$

and thus $k_s$ is the slope of a $\chi$ plot. Throughout the paper we use channel steepness or profile steepness to refer to $k_s$. We never use steepness as a synonym of channel gradient.

## 2.3 Estimating time since perturbation

The fluvial response time, $\tau$, defined as the time for a perturbation originating at base level to travel to any point on the river network, is given by the upstream integral of the inverse wave speed:

$$\tau = \int\limits_{x_b}^{x} \frac{1}{KA^m S^{n-1}} dx \tag{6}$$

When $(1/KA_0^m)^{\frac{1}{n}}$ is included in the integral in equation 4, $\chi$ has units of time and the integral yields the fluvial response time for the case of $n = 1$ (Whipple and Tucker, 1999; Willett et al., 2014). Thus, if $n = 1$ and $K$ is uniform, $\tau$ is simply $\chi/K$. For other cases of $n$, $\tau$ can be estimated analytically for quasi-equilibrium river profiles by solving for slope under steady-state conditions in equation 2 (Willett et al., 2018), such that

$$S \approx \left(\frac{U}{K}\right)^{\frac{1}{n}} A^{\frac{-m}{n}} \tag{7}$$

Substituting equation 7 into equation 6 gives the following expression for $\tau$

$$\tau \approx k_s \frac{\chi}{U} \tag{8}$$

We used 1-D simulations to explore more quantitatively how deviations away from $n = 1$ may influence $\tau$ values estimated for real landscapes if it is assumed that $n = 1$.

## 2.4 Parameter values

For the simulations, we selected common values from published studies for all parameters in the stream power model of bedrock incision. The concavity index, $\theta = m/n$, has been shown to commonly range between $0.4 - 0.7$ for equilibrium channels (Lague, 2014; Stock and Montgomery, 1999; Tucker and Whipple, 2002; Whipple and Tucker, 1999). Although the slope exponent, $n$, has been shown to be commonly greater than unity from relationships of channel steepness with erosion rate (DiBiase et al., 2011; Harel et al., 2016; Lague, 2014; Ouimet et al., 2009), data on knickpoint propagation is best explained with $n = 1$ (Lague, 2014), and mechanistic approaches yield estimates ranging between $2/3$ and $5/3$ (Whipple et al., 2000; Larimer et al., 2019). Given the uncertainty in the value of $n$ and the simplicity of the $n = 1$ case, we assume $n = 1$ for all simulations and analyses. However, we also ran 1-D simulations with both $n = 2/3$ and $n = 5/3$ and present these results in the

supplement. For $n \neq 1$ simulations we adjusted $m$ and $K$ such that both concavity and fluvial relief remained constant between these simulations and those in which $n = 1$. We chose a reference concavity index, $\theta_{ref}$, of $0.45$ for all simulations. Thus $k_s$ as calculated in equation 5 is equivalent to the normalized channel steepness index, $k_{sn}$, as described by Wobus et al. (2006), and hereafter will be referred to as such. We used $6.69$ for the reciprocal Hack coefficient, $k_a$ (Whipple and Tucker, 1999) and $1.8$

for the reciprocal Hack exponent, $h$ (Hack, 1957). We chose a low background rock uplift rate of $50 \ \mathrm{mMyr}^{-1}$ and a value of $1 \times 10^{-6} \ \mathrm{m}^{0.1}\mathrm{yr}^{-1}$ for the erodibility coefficient, $K$, as this value allows $\chi$ to be read as response time in millions of years and because it is similar to other published values (e.g., Beeson et al., 2017; Stock and Montgomery, 1999; Willett et al., 2018).

To calculate $\chi$ and $k_s$ in the Sierra Nevada and Sierra San Pedro Mártir, we assumed uniform $U$ and $K$ and describe below how parameter values were estimated in these landscapes. Although we know rock type is nonuniform in both landscapes,

this approach reveals whether changes in channel steepness correspond with lithologic contacts or whether changes in channel steepness occur independent of rock type and might thus reflect temporal changes in boundary conditions. Furthermore, we do not know $K$ for each formation and therefore cannot calculate $\chi$ with local $K$ values inside the integral (Willett et al., 2014). With a 1-D model we demonstrate the signature that heterogeneous lithology would impart on $\chi$ plots in tilted landscapes. For all river profile analysis, we clipped both DEMs by the mountain front on the western side to limit our analysis to bedrock

rivers. In the Sierra Nevada, we identified the mountain front using a threshold slope of $0.01$ on a DEM that was smoothed with a 20 km moving window. In the Sierra San Pedro Mártir, we used the upstream boundary of Quaternary alluvium as mapped by the Mexican National Institute of Statistics and Geography (INEGI, 1984). We used a scaling area, $A_0$, of $1 \ \mathrm{m}^2$ and defined channel heads using a critical drainage area, $A_c$, of $0.5 \ \mathrm{km}^2$.

## 3    Modeling equilibrium fluvial longitudinal profiles

We solved equation 2 analytically to simulate equilibrium fluvial longitudinal profiles for the case where $U$ is uniformly $50$ $\mathrm{mMyr}^{-1}$ (Fig. 1b) and numerically for the case in which $U$ is a linear gradient from zero at the channel outlet to a maximum uplift rate of $50 \ \mathrm{mMyr}^{-1}$ at the channel head (Fig. 1c) and for the case with the reverse uplift gradient of maximum uplift rate of $50 \ \mathrm{mMyr}^{-1}$ at the channel outlet to zero at the channel head (Fig. 1d). We used a 200 km long mainstem river with three 40 km long tributaries that entered the mainstem at 20, 80, and 140 km upstream of the outlet (Fig. 1a). The tributaries were

made to run perpendicular to the mainstem such that the uplift rate for each tributary was equal to the rate experienced by the mainstem at the tributary confluence.

At steady-state, longitudinal profiles with uniform $U$ are straight on $\chi$ plots (Fig. 1b) with the slope equal to the channel steepness, but for the case in which $U$ is a linear gradient from zero at the channel outlet to a maximum at the channel head, profiles have positive curvature in $\chi$ plots, particularly near the channel outlet (Fig. 1c). The positive curvature $\chi$ plot results

from channel steepness increasing toward the channel head, which allows erosion rates to increase moving towards the channel head to balance the gradient in rock uplift rate. In contrast, for the case in which $U$ is a linear gradient decreasing from a maximum at the channel outlet to zero at the channel head (Fig. 1d), longitudinal profiles have negative curvature on a $\chi$ plot.

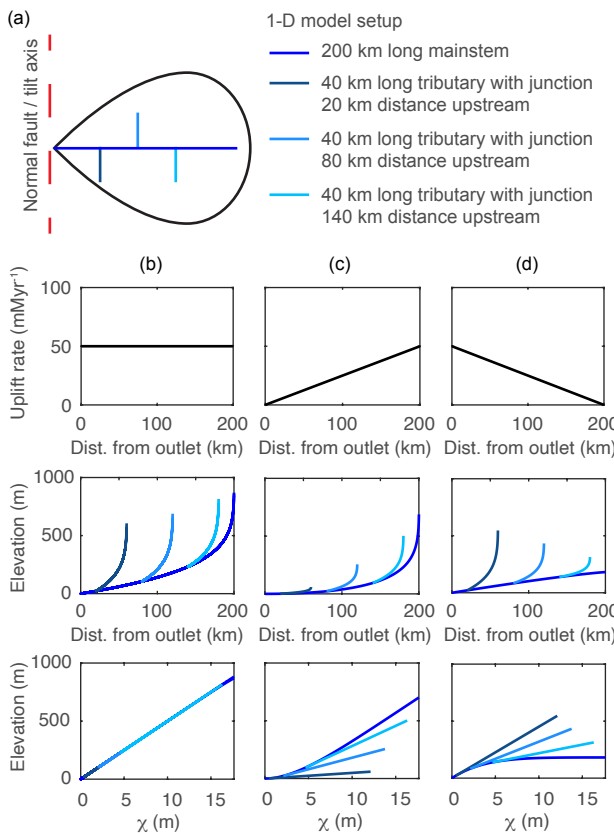

**Figure 1.** Equilibrium river longitudinal profiles. (**a**) Planform schematic of simple river network composed of linked 1-D river-profile models. River profiles equilibrated to (**b**) uniform $K$ of $1x10^{-6}$ and uniform $U$ of $50\ \mathrm{mMyr^{-1}}$, (**c**) a linear uplift gradient with maximum $U$ at the channel head of $50\ \mathrm{mMyr^{-1}}$, or (**d**) a linear uplift gradient with maximum $U$ at the channel outlet of $50\ \mathrm{mMyr^{-1}}$. Upper row shows uplift field, middle row shows longitudinal profiles, and lower row shows $\chi$ plots.

## 4 Modeling fluvial longitudinal profile response to perturbations

We solved equation 2 numerically to simulate fluvial longitudinal profile response to various perturbations that move a river away from equilibrium. Specifically, we focus on nonuniform uplift due to forward tilting as well as perturbations that result in disequilibrium profile forms that share characteristics with those generated in response to forward tilting. Starting from initial conditions with equilibrated river profiles, we simulated four main scenarios: 1) an instantaneous uniform pulse of rock uplift that increased the elevation of all points along the river profile 1 km with respect to the outlet; 2) a step decrease in equilibrium profile steepness achieved through either a uniform step increase in bedrock erodibility or a step decrease in uniform rock uplift rate; 3) major truncation or beheading of the mainstem river; and 4) forward tilting owing to an instantaneous rigid-block tilt about a horizontal axis perpendicular to the mainstem river and located at the river outlet that increased the headwater

elevation by 1 km with respect to the outlet (tilt of $\sim 0.3°$). For each of these scenarios we conducted analogous simulations but run with $n = 2/3$ and $n = 5/3$ and present those results in the supplement. Although the transient response to both uniform rock uplift/base level fall and step changes in uplift rate or erodibility have been extensively researched (e.g., Baldwin et al., 2003; Bonnet and Crave, 2003; Grimaud et al., 2016; Howard, 1994; Rosenbloom and Anderson, 1994; Royden and Perron, 2013; Tucker and Whipple, 2002; Whipple and Tucker, 1999), we include these perturbations herein to provide comparisons between well-known transient responses and those induced by tilt or truncation. We hope that this comparison highlights the fact that many perturbations disrupt co-linearity of mainstems and tributaries and that examination of the relationship between mainstems and tributaries can facilitate reconstruction of tectonic histories.

To explore likely deviations from the idealized instantaneous rigid-block forward tilt simulation, we simulated ten additional tilting scenarios: 1) backward tilting owing to an instantaneous rigid-block tilt about a horizontal axis perpendicular to the mainstem river and located at the channel head that increased channel outlet elevation by 200 m; 2) lateral tilting owing to an instantaneous rigid-block tilt about a horizontal axis parallel to the mainstem river and located along the mainstem that increased tributary channel heads by 800 m, 3-6) a rapid rigid-block tilting event that uplifted the mainstem channel head 1 km over 1, 3, 5, and 10 Myr; 7) instantaneous forward tilting but with a non-uniform uplift field that deviates from perfect rigid-block tilting, 8) an instantaneous rigid-block forward tilting scenario with a vertical bed of more erodible rock mid-profile; 9) an instantaneous rigid-block forward tilting scenario with a vertical bed of less erodible rock near the outlet; and 10) a step increase in forward tilting rate with a vertical bed of more erodible rock mid-profile.

For all simulations, we used the model setup as described above and shown in Figure 1. For the majority of simulations we imposed a steady, uniform background rock uplift rate of $50\,\mathrm{mMyr^{-1}}$, but for simulations 3-6 and 9, we imposed a nonuniform background uplift rate of zero at the channel outlet and $50\,\mathrm{mMyr^{-1}}$ at the channel head. We used different tilt angles for each of the simulations of instantaneous tilt in different directions (forward, back, and lateral tilting) because using the same high tilt angle for back tilting as we used for forward tilting resulted in river reversal but did not result in significant rock uplift at the channel head in lateral tilting owing to the much shorter length of tributaries compared with the mainstem river. Therefore, we used a lower tilt angle for back tilting and a higher tilt angle for lateral tilting. A list of all simulations, the background uplift field used for each, and the associated figure can be found in Table 1. For each simulation, we tracked the elevation of the river profile with respect to the river profile immediately after the perturbation to simulate the location of paleoriver deposits as well as incision depth below this paleotopography marker. Animations of simulations described herein are available as an online resource (see Beeson, 2019).

## 4.1 Rapid pulse of uniform rock uplift

The transient response to a 1 km instantaneous pulse of uniform rock uplift is illustrated in Figure 2a. The pulse of uniform uplift raises the surface elevation of the equilibrated initial river profile by 1 km and results in a 1 km high vertical-step knickpoint at the outlet. Tributaries and the mainstem are brought above equilibrium (shown as a dashed blue line) by an equal amount (Fig. 2a middle row). The knickpoint then propagates upstream through the river network in a wave like manner, lowering the fluvial profile by 1 km to return it back to equilibrium. Below the mainstem knickpoint, tributary knickzone

**Table 1.** List of simulations presented herein and run with 1-D model river network composed of linked river profile evolution models. In each simulation a single perturbation moves the river network away from equilibrium. Aside from perturbations, the background rock uplift field is the only parameter that varies among simulations, though in all simulations maximum background $U$ is 50 mMyr$^{-1}$. In all simulations except those with heterogeneous lithology, $K = 1 \times 10^{-6}$ m$^{0.1}$yr$^{-1}$, and in all simulations, $\theta = 0.45$, $n = 1$, $h = 1.8$, $k_a = 6.69$.

| Perturbation | Background uplift field | Figure |
|---|:---:|:---:|
| Instantaneous uniform pulse of uplift | Uniform | 2a |
| Step increase in K | Uniform | 2b |
| Step decrease in K | Uniform | S3 |
| Instantaneous beheading of mainstem river | Uniform | 3a |
| Instantaneous forward tilting of 1 km at crest | Uniform | 3b |
| Instantaneous back tilting of 200 m at outlet | Uniform | S6 |
| Instantaneous lateral tilting of 800 m at tributary channel head | Uniform | S7 |
| Forward tilting of 1 km at crest over 1 Myr | Tilt | 4a |
| Forward tilting of 1 km at crest over 3 Myr | Tilt | 4b |
| Forward tilting of 1 km at crest over 5 Myr | Tilt | 4c |
| Forward tilting of 1 km at crest over 10 Myr (continuous tilting) | Tilt | 4d |
| Instantaneous forward tilt of 1 km at crest in an elastic half-space | Uniform | S8 |
| Instantaneous forward tilt of 1 km at crest with vertical bed of more erodible rock | Uniform | 5a |
| Instantaneous forward tilt of 1 km at crest with vertical bed of less erodible rock | Uniform | 5b |
| Step increase in tilt rate with vertical bed of more erodible rocks | Tilt | S9 |

height is uniform and equal to the 1 km pulse of rock uplift, which is equal to the surface uplift following the perturbation, whereas incision depth below the uplifted initial river profile is also uniform (Fig. 2a lower row) but equal to the total rock uplift accumulated since the beginning of the simulation (background rate plus 1 km pulse). As highlighted in the $\chi$ plots (Fig. 2a middle row), at all times during the transient evolution, the knickpoint simply offsets profiles of equal and uniform steepness and each knickpoint within the tributaries and the mainstem collapse to a common $\chi$ value as expected from their common
5   point of origin. With $K = 1 \times 10^{-6}$ and uniform, $\chi$ values of the vertical-step knickpoint can be interpreted as knickpoint travel times, $\tau$, in millions of years (Equations 4 and 6). For example, at $t = 5$ Myr, the vertical-step knickpoint is located at a $\chi$ value equal to five, which with $K = 1 \times 10^{-6}$ and uniform corresponds to $\tau = 5$ Myr. If $n = 2/3$ or $5/3$, the first-order fluvial response is similar to the case in which $n = 1$ except that the vertical-step knickpoint broadens with time because of
10  slope-dependent knickpoint celerity (Figs. S1 and S2). The change in celerity with increasing slope also decreases the celerity of the base of the knickpoint in the $n = 2/3$ case and increases it in the $n = 5/3$ case as compared to the $n = 1$ case, increasing and decreasing overall response time, respectively (Figs. S1 and S2).

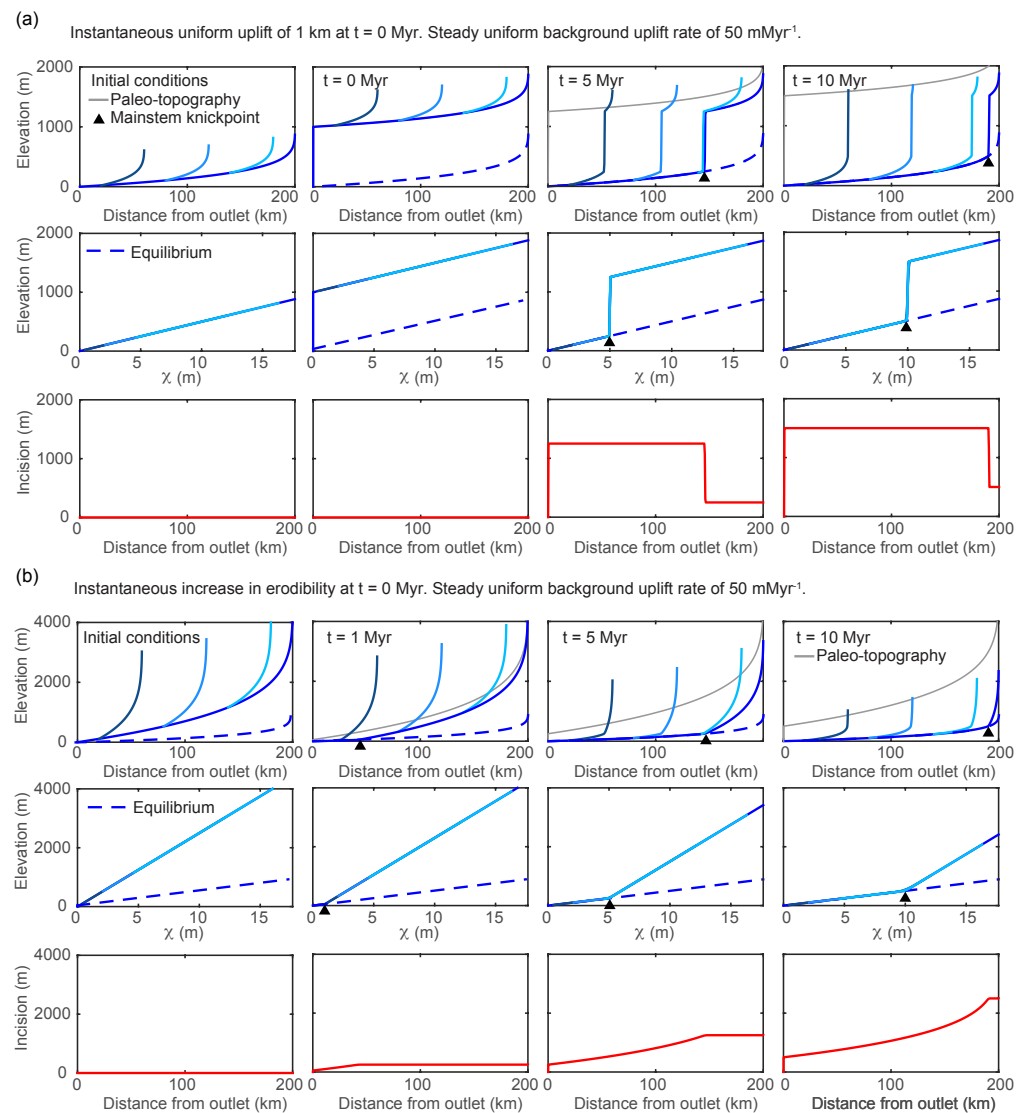

**Figure 2.** Results from 1-D simulations of a river network equilibrated to a uniform background uplift rate of $50 \ \mathrm{mMyr}^{-1}$ and subject to various perturbations that move profiles away from equilibrium, but maintain uniform steepness upstream of the propagating knickpoint. (**a**) Results from multiple timesteps for simulation of instantaneous uniform uplift of 1 km at $t = 0$ Myr versus (**b**) a step increase in erodibility, $K$, or equivalently a step decrease in rock uplift rate. In (a) and (b), the upper row of plots shows longitudinal profiles with shades of blue corresponding to mainstem and tributaries shown in Fig. 1a. The grey line tracks the pre-perturbation elevation of the river or the potential elevation of paleoriver deposits, i.e. "paleo-topography". The middle row shows $\chi$-elevation profiles ($\chi$ plots) for the mainstem and tributaries with shades of blue corresponding to mainstem and tributaries shown in Fig. 1a, and the lower row shows incision below the "paleo-topography" markers since $t = 0$ Myr. In all plots the dashed blue line denotes the final equilibrium state post-perturbation.

## 4.2 Step increase in bedrock erodibility or step decrease in rock uplift rate

A step decrease in fluvial relief can be accomplished by increasing bedrock erodibility or decreasing uniform rock uplift rate. We ran both simulations using different parameters to generate equilibrium initial conditions but the same parameters following each step change such that, following perturbation, the profiles in both simulations were evolving towards the same equilibrium profile and as such the transient response was identical between the two simulations. The transient response to a step increase in bedrock erodibility, $K$, or equivalently a step decrease in uniform rock uplift rate, is illustrated in Figure 2b. To perturb erodibility, the initial condition is a river profile equilibrated to the background uplift rate of 50 $\mathrm{mMyr}^{-1}$, but with uniform $K = 2 \times 10^{-7}$ $\mathrm{m}^{0.1}\mathrm{yr}^{-1}$, which is then increased to $1 \times 10^{-6}$ $\mathrm{m}^{0.1}\mathrm{yr}^{-1}$. To perturb uniform rock uplift rate, the initial condition is a river profile equilibrated to uniform erodibility of $1 \times 10^{-6}$ $\mathrm{m}^{0.1}\mathrm{yr}^{-1}$ and uniform rock uplift rate of 250 $\mathrm{mMyr}^{-1}$, which is then decreased to 50 $\mathrm{mMyr}^{-1}$. The step increase in erodibility (or decrease in rock uplift) decreases the new equilibrium steepness such that the upper reaches are brought farther above equilibrium as compared to the lower reaches that remain closer to equilibrium (Fig. 2b middle row). Post-perturbation, channel steepness is greater than that required to balance the background rock uplift rate and the river begins to incise more rapidly than the rock uplift rate (Fig. 2b upper and middle row). The imbalance between uplift and incision allows the river profile to decrease in elevation until the river profile is brought back down to an equilibrium grade, which occurs first at the outlet. This results in a positive-curvature slope-break knickpoint that originates at the outlet and propagates upstream through the river network in a wave-like manner. The knickpoint separates downstream reaches that have achieved the new lower equilibrium steepness from the upstream reaches that retain the original higher steepness. As the transient progresses, a distinct pattern of incision depth emerges, the magnitude of which is everywhere greater than the total rock uplift accumulated during the simulation, with the exception of that at the outlet. Incision depth increases away from the mountain front up until the location of the knickpoint, but then remains uniform upstream due to the uniform steepness, and hence uniform incision rate. Again, the knickpoint within the tributaries and the mainstem collapse to a common $\chi$ value as expected from them starting at a common point of origin (Fig. 2b middle row). If $n = 2/3$ or $5/3$, the first-order fluvial response is similar to the case in which $n = 1$, except knickpoint celerity is reduced or accelerated, respectively (Figs. S1 and S2). A step decrease in erodibility or step increase in uplift rate produces the inverse of these signatures, with a negative-curvature slope-break knickpoint separating a downstream section that has achieved the new higher equilibrium steepness and an upstream section with uniform, lower steepness (Fig. S3). In both cases, incision depth is greatest at the outlet and is everywhere less than the magnitude of accumulated uplift, except at the outlet.

## 4.3 Truncation of mainstem

The transient response to truncation or beheading of a river network is illustrated in Figure 3a. To simulate truncation, a 100 km section of river profile is removed from the headwaters of a 300 km long river profile equilibrated to a background uplift rate of 50 $\mathrm{mMyr}^{-1}$ and an erodibility of $1 \times 10^{-6}$ $\mathrm{m}^{0.1}\mathrm{yr}^{-1}$ (Fig. 3a upper row). Channel steepness of the truncated mainstem river profile is everywhere less than that required to balance the background uplift rate and the river profile begins to increase in elevation and steepness (Fig. 3a upper and middle row). Owing to the larger fractional decrease in drainage

area in the upper reaches, truncation brings the upper reaches farther below equilibrium than the lower reaches, resulting in negative curvature of the mainstem $\chi$ plot. Thus, a uniform background rock uplift rate combined with a nonuniform erosion rate mirroring the channel steepness results in a nonuniform surface uplift rate that brings the truncated mainstem river profile back to an equilibrium steepness in a wave-like manner that progresses upstream from the outlet. The tributaries retain the original equilibrium steepness, having not lost drainage area, but are pulled below equilibrium by the area perturbation on the mainstem. Paleotopography markers delineate the location of the original 300 km river that uplifts uniformly with the background uplift rate (Fig. 3a upper row). Only at the channel outlet, where equilibrium channel steepness was retained, does incision reflect rock uplift accumulated from the background uplift rate over the course of the simulation. Upstream of the outlet, incision below paleotopography markers tapers nonlinearly towards zero at the channel head (Fig. 3a lower row). If $n = 2/3$ or $5/3$, the first-order fluvial response is similar to the case in which $n = 1$, except knickpoint celerity is reduced or accelerated, respectively. It should be noted that all truncation simulations are limited in that they do not include the positive feedback often associated with drainage area loss (Willett et al., 2014) that may limit the ability of the river basin to achieve equilibrium steepness following truncation.

## 4.4 Rapid pulse of nonuniform rock uplift due to forward tilting

The transient response to an instantaneous rigid-block tilt about a horizontal axis perpendicular to the mainstem river and located at the river outlet that increases the headwater surface elevation by 1 km with respect to the outlet (tilt of $\sim 0.3°$) is illustrated in Figure 3b. The rapid nonuniform rock uplift rate brings the upper reaches farther above equilibrium as compared to the lower reaches, which remain closer to equilibrium (Fig. 3b upper and middle row). Post tilt, the entire mainstem that drains perpendicular to the tilt axis experiences a uniform increase in profile gradient (equal to the tilt angle), but owing to the nonlinear downstream increase in drainage area, profile steepness, $k_{sn}$, is increased by a greater degree near the outlet, resulting in negative curvature of the mainstem $\chi$ plot. In contrast, the tributaries that drain parallel to the tilt axis are uplifted a uniform amount equal to the rock uplift at each respective tributary junction and are thus offset above equilibrium, but their steepness is not affected (just as described above for uniform uplift). Together, these tilt-induced perturbations to profile steepness and elevation give the $\chi$ plot of a river network tilted toward the mainstem outlet a unique signature with all portions of the network plotting above the equilibrium line, but with the mainstem having negative curvature and plotting above uniform steepness (straight) tributaries (Fig. 3b middle row).

After tilting, channel steepness along the entire mainstem is greater than that required to balance the background rock uplift rate and all points on the mainstem begin to incise. The imbalance between uplift and incision allows the mainstem river profile to decrease in elevation until it is brought back down to the pre-tilt equilibrium grade, which occurs first at the outlet owing to the greater steepness and proximity to equilibrium. This results in a positive-curvature slope-break knickpoint that originates at the outlet and propagates upstream through the river network in a wave-like manner. The knickpoint separates a downstream section that has returned to the original pre-tilt equilibrium steepness from an upstream section that is over-steepened and retains the nonuniform tilt-induced steepness that plots as a profile with negative curvature on a $\chi$ plot. If $n = 2/3$, the first-order fluvial response is similar to the case in which $n = 1$, except knickpoint celerity is reduced (Fig. S4). If

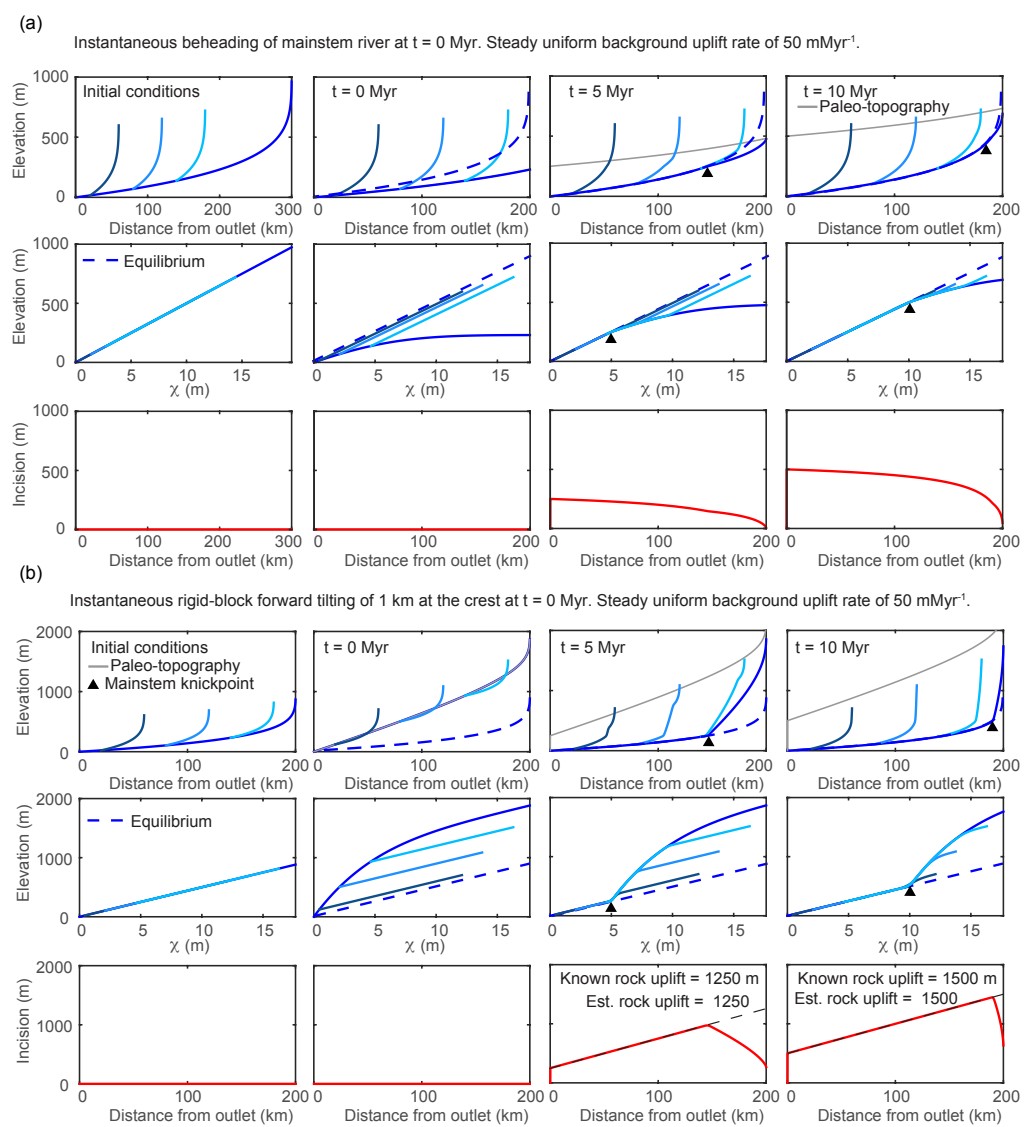

**Figure 3.** Results from 1-D simulations of a river network equilibrated to a uniform background uplift rate of $50 \ \mathrm{mMyr}^{-1}$ and subject to various perturbations that move profiles away from equilibrium and induce nonuniform steepness upstream of the propagating knickpoint. Results from multiple timesteps for simulation of (**a**) instantaneous truncation of a 300 km long river to a 200 km long river and (**b**) instantaneous tilting at $t = 0$ Myr with tilt axis at the river mouth, perpendicular to the main stem, and 1 km maximum uplift at the channel head. Known values of total rock uplift at the crest and those estimated using the spatial gradient in incision are noted in the lower row. Line color, basic description of plots filling each row, and network structure are the same as described in Fig. 2.

$n = 5/3$, knickpoint celerity is accelerated, but the characteristic negative curvature of the mainstem and tributary knickzones can actually become positive with time because of increasing celerity with increasing slope (Fig. S5).

### 4.4.1 Tributary knickzones record tilt timing

As the over-steepened mainstem incises back to equilibrium, knickzones form at the outlets of tributaries as a response to ongoing base level fall at the junction with the more rapidly incising mainstem. Thus, tributary knickzones begin to form the instant after the tilt perturbation at the outlet of every tributary regardless of position within the basin. The timing of tilt is recoverable from the tributary knickzones by subtracting the $\tau$ value at the corresponding tributary confluence from the $\tau$ value at the upstream end of the tributary knickzone. These knickzones are unique in that they are neither vertical-step knickpoints nor are they slope-break knickpoints. Rather, the base of the knickzone is marked by a slope-break knickpoint but upstream of this point they have significant length along which steepness is nonuniform and equal to that of the mainstem upstream of the tributary junction. This results in a perfect collapse of tributary knickzones with the mainstem on $\chi$ plots.

### 4.4.2 Tributary knickzone drop height and incision depth reflect tilt magnitude

Tributary knickzones reach their maximum height when the mainstem slope-break knickpoint is at the tributary junction. They then propagate upstream at this maximum height and leave an equilibrium grade tributary profile in their wake. Tributary knickzone height increases linearly moving up the mainstem at a rate equal to the tilt angle up to the mainstem slope-break knickpoint and then decreases moving further upstream. Below the mainstem knickpoint, tributary knickzone height is equal to the local rock uplift from the pulse of tilting, which is equal to the surface uplift following the perturbation, whereas incision depth below the uplifted initial profile (Fig. 3b lower row) is equal to the total rock uplift accumulated since the beginning of the simulation (background rate plus tilting pulse). At intermediate times steps in the transient evolution, a distinct triangular pattern of incision depth below paleotopography markers is evident (Fig. 3b lower row). Incision depth increases linearly with distance from the mountain front with the location of maximum incision corresponding to the slope-break knickpoint in the mainstem, above which incision depth rapidly decreases owing to the upstream decrease in steepness. The linear increase in incision depth with distance from the outlet reflects the linear rock uplift gradient due to tilting and projection of this gradient of incision depth to the channel head accurately recovers channel head surface uplift resulting from tilting plus uplift accumulated from the uniform background rate (Fig. 3b lower row).

### 4.5 Rapid pulse of nonuniform rock uplift due to back tilting

The transient response to an instantaneous rigid-block tilt about a horizontal axis perpendicular to the mainstem river and located at the channel head that increases the outlet surface elevation by 200 m with respect to the channel head (tilt of $\sim 0.06°$) is illustrated in Figure S6. The response is similar to the transient response to forward tilting in that tilting induces a mainstem knickpoint, nonuniform steepness in the mainstem, and distinct patterns in tributary steepness over which tributaries collapse with the mainstem on the $\chi$ plot. However, many of the specific characteristics of these signatures are reversed

when the uplift gradient relative to the mainstem flow direction is reversed in the back-tilted case (compare Fig. 3b with Fig. S6). Post-tilt, the lower reaches are brought farther above equilibrium compared to the upper reaches and, although the entire mainstem experiences a uniform decrease in gradient, $k_{sn}$ is decreased by a greater degree near the outlet owing to the nonlinear downstream increase in drainage area, resulting in positive curvature of the mainstem $\chi$ plot. Similar to the response to forward tilting, the tributaries that drain parallel to the tilt axis are uplifted a uniform amount equal to the rock uplift at each respective tributary junction and are thus offset above equilibrium, but their steepness is not affected. However, in the back-tilted case tributaries are farther above equilibrium than the mainstem and plot above the mainstem in the $\chi$ plot.

The primary difference in the transient response to back tilting as compared to forward tilting is the character of the mainstem knickpoint. Back tilting induces a vertical-step knickpoint, rather than a slope-break knickpoint, owing to uplift of the river outlet. The vertical-step knickpoint retains its original height as it propagates upstream, briefly raising mainstem elevation until the knickpoint passes and mainstem elevation returns to equilibrium. The vertical-step knickpoint propagates up tributaries, with tributaries collapsing with the mainstem on the $\chi$ plot over the vertical-step, similar to the transient response to a uniform pulse of uplift. The drop height of the vertical-step knickpoint thus only reflects the magnitude of the pulse of rock uplift at the river outlet. Although the mainstem temporarily rises during the transient response, the entire landscape has not experienced equivalent surface uplift. The location of the vertical-step knickpoint records the timing of perturbation in both the mainstem and tributaries. The transient mainstem response also inscribes a distinct incision pattern in which incision decreases linearly upstream up to the mainstem knickpoint where a sharp decrease is followed by increasing incision upstream. Incision downstream of the vertical-step knickpoint records total accumulated rock uplift post-tilt, that is, the sum of uplift from the punctuated tilt and uplift accumulated from the background uplift rate during the transient response.

Mainstem steepness upstream of the knickpoint is less than equilibrium steepness owing to the initial back tilting. As a result of the reduced steepness, mainstem erosion does not keep pace with uplift rate. The tributaries experience this as ongoing base level rise from the time of rapid tilting until the mainstem knickpoint passes the tributary junction. The ongoing base level rise induces low-gradient reaches in the tributaries that collapse with the mainstem, analogous to the tributary knickzones formed in the forward-tilting case as a result of ongoing lowering of tributary base level. These low-gradient reaches record tilt timing in the position of the upstream end relative to tributary junction, in the same manner that tributary knickzones record timing in the forward-tilting case. Unlike the forward-tilting case, tributary knickzones do not record the magnitude of rock uplift.

### 4.6 Rapid pulse of nonuniform rock uplift due to lateral tilting

The transient response to an instantaneous rigid-block tilt about a horizontal axis parallel to the mainstem river and located along the mainstem that increases the tributary channel head surface elevation by 800 m (tilt of $\sim 1°$) is illustrated in Figure S7. For this simulation, we used a slightly different model network configuration in which the three tributaries all joined the mainstem on the uplifted side. The response is identical to that of forward tilting except in this case the mainstem remains at equilibrium and the tributaries are steepened. Analogous to the mainstem in the forward-tilting case, all tributaries have negative curvature on the $\chi$ plot, positive-curvature mainstem knickpoints, and incision patterns that increase upstream up to the mainstem knickpoints and decrease upstream (Fig. S7).

## 4.7 Deviations in tilt duration away from the instantaneous tilting end-member scenario

Although we recognize that instantaneous tectonic perturbations are not geologically realistic, they are useful end-member scenarios to quantify the transient fluvial response to rapid perturbations in that an instantaneous perturbation generates the clearest signature and can represent scenarios in which the duration of the tectonic event is very short relative to the channel response time. To explore how tilt duration impacts signatures of tilt, we ran a series of simulations in which rigid-block forward tilting raised the channel head 1 km over a duration of 1, 3, 5, and 10 Myr (Fig. 4). For these simulations we used an initial river profile equilibrated to a tilting rate of 50 $\mathrm{mMyr^{-1}}$ at the channel head and zero uplift at the channel outlet. Uplift rate was then returned to this low magnitude tilting rate after the period of rapid tilting. We chose to use a low magnitude background tilt rather than a uniform background uplift rate because, if a uniform background uplift rate were used, lower tributaries would experience a step decrease in uplift rate and upper tributaries would experience a step increase and we wanted to isolate the effects of a rapid tilting event.

With increasing tilt duration, the first-order signatures of tilt become less pronounced because equilibrium channel steepness for the pulse of more rapid tilting becomes closer to that of the initial condition. During the rapid tilting phase, the ongoing uplift rate is higher than that to which the initial profiles were equilibrated and as such profiles rise to a higher equilibrium steepness and elevation (Fig. 4c and d). During this rising state there is a negative curvature slope-break knickpoint that forms at the outlet and tracks the most upstream extent of the reach equilibrated to the new rapid tilt rate. However, this negative curvature slope-break knickpoint is extremely hard to identify amongst the tilt-induced negative curvature of the entire profile (e.g., Fig. 4c) and thus duration of the pulse of tilt is hard to constrain using the mainstem profile form. Once the tilt rate returns back to the lower initial rate, the response is similar to that following the instantaneous tilt and a positive curvature slope-break knickpoint forms at the outlet and tracks timing of cessation of rapid tilt. During rapid tilting, the timing of the onset of tilt is recorded by the difference in $\tau$ between the mainstem-tributary junction and the point at which tributary channel steepness transitions from nonuniform to uniform (e.g., Fig 4c). After rapid tilting has ceased, the duration of tilting is recorded in the difference in $\tau$ between the point where tributary channel steepness diverges from that of the mainstem and the point at which tributary channel steepness transitions from nonuniform to uniform (e.g., Fig 4b) and the timing of cessation is recorded by the difference in $\tau$ between the mainstem-tributary junction and the point where tributary channel steepness diverges from that of the mainstem. Although identifying these points is possible in this simplified model, in natural landscapes where tributary steepness is impacted by numerous other processes it is likely not possible. However, it is still feasible to make a rough estimate of tilt duration using tributary channel steepness in that tributary knickzone steepness reflects the uplift gradient such that a more rapid tilt results in steeper tributary knickzones (compare Fig 4a with Fig 4d).

Although similar profile signatures form during the rapid tilting phase, the magnitude of rock uplift is not recorded until the uplift gradient returns to the initial low magnitude uplift gradient and the mainstem incises back toward the initial condition. With a tilt duration of 1 Myr, incision depth and tributary knickzone height exactly reflect tilt magnitude as in the instantaneous case (Fig. 4a). Mainstem incision and full tributary knickzone drop height slightly underestimate accumulated rock uplift in the cases where tilt duration is longer than 1 Myr (Fig. 4b-d).

(a) Rigid-block forward tilting beginning at t = 0 Myr that raised channel head elevation by 1 km over 1 Myr.
Background tilt rate of 50 mMyr⁻¹ at the channel head and zero at the mountain front.

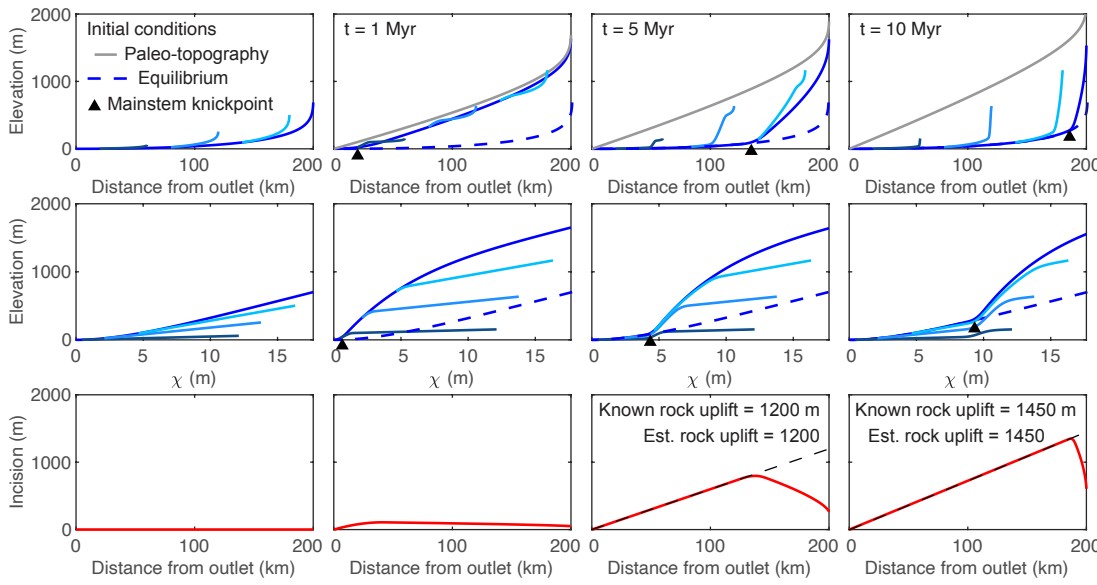

(b) Rigid-block forward tilting beginning at t = 0 Myr that raised channel head elevation by 1 km over 3 Myr.
Background tilt rate of 50 mMyr⁻¹ at the channel head and zero at the mountain front.

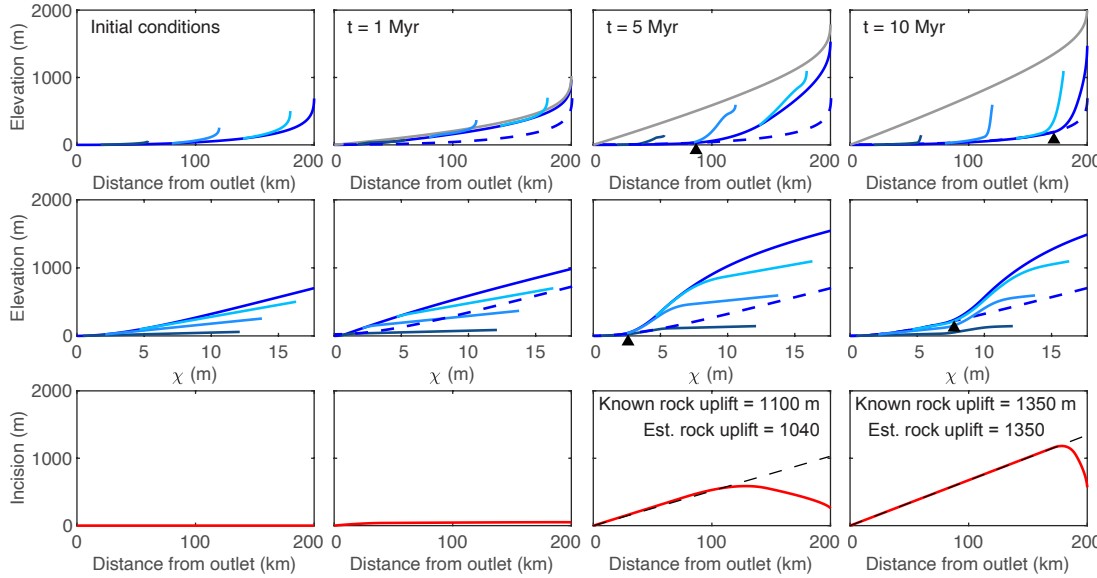

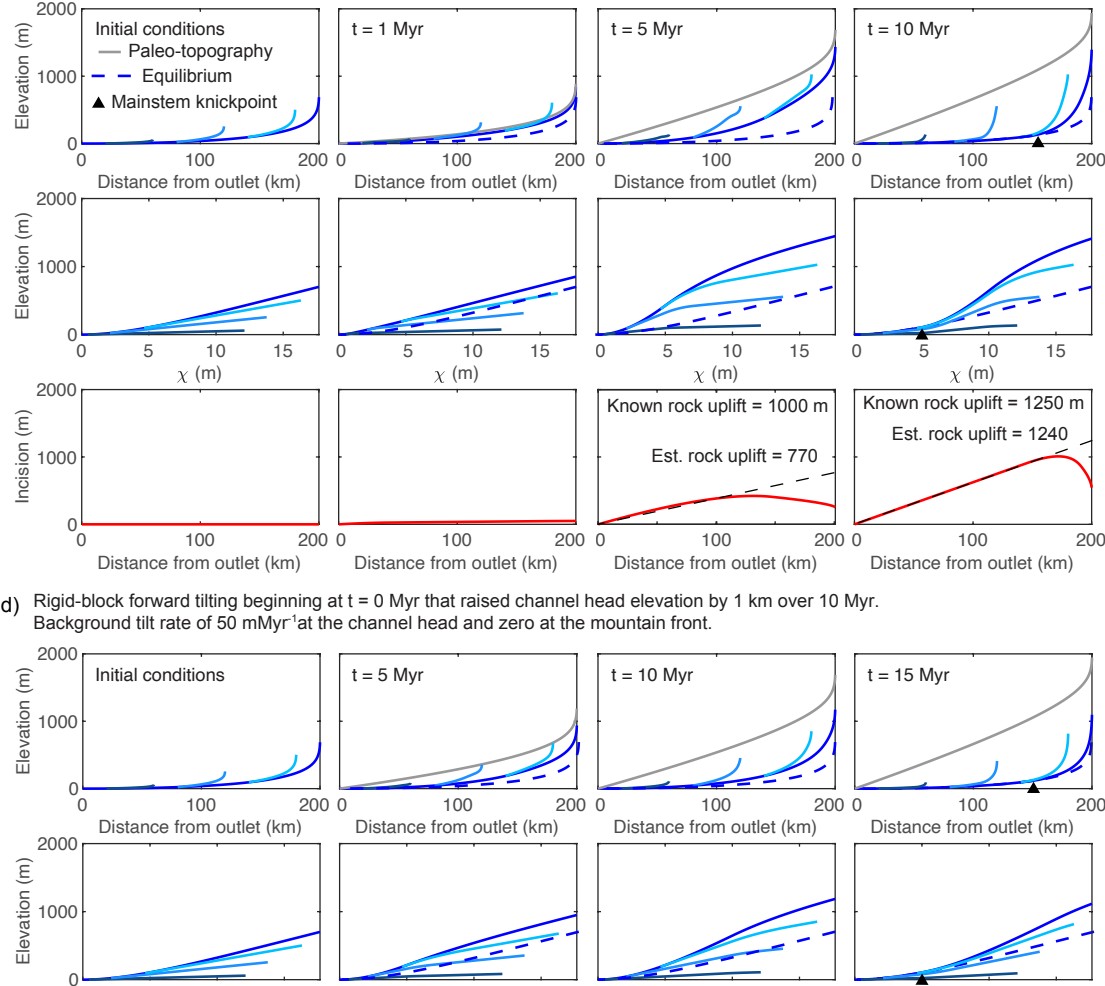

**(c)** Rigid-block forward tilting beginning at t = 0 Myr that raised channel head elevation by 1 km over 5 Myr. Background tilt rate of 50 mMyr⁻¹ at the channel head and zero at the mountain front.

**(d)** Rigid-block forward tilting beginning at t = 0 Myr that raised channel head elevation by 1 km over 10 Myr. Background tilt rate of 50 mMyr⁻¹ at the channel head and zero at the mountain front.

**Figure 4.** Results from 1-D simulations of equilibrated river network subject to various perturbations that move profiles away from equilibrium. Results from multiple timesteps for simulation of rigid-block tilting toward the channel outlet that raises channel head elevation by 1 km over 1 Myr (**a**), 3 Myr (**b**), 5 Myr (**c**), and 10 Myr (**d**). After 1 Myr, 3 Myr, 5 Myr, and 10 Myr, respectively, profiles experience a steady tilt rate of 50 $\mathrm{mMyr}^{-1}$ at the crest and zero at the mountain front. Basic description of plots filling each row are the same as described in Fig. 1. Known values of total rock uplift at the crest and those estimated using the spatial gradient in incision are noted in the lower row.

## 4.8 Non-uniform uplift that deviates from perfect rigid-block forward tilting

We simulated a deviation to rigid-block style tilting by setting the magnitude of instantaneous uplift using the vertical displacement predicted by a single-dislocation model with slip-boundary conditions in an elastic half-space as explained by Martel et al. (2014; Fig. S8). We used a fault dip of $90°$ and a model fault depth of 40 km. In this model, uplift decays to $1/10$ of maximum uplift 55 km from the model fault (where maximum uplift occurs), which we place at the channel head. The transient fluvial response to forward tilting with this simple elastic model generates similar signatures to rigid-block tilting in that positive curvature in the $\chi$ plot near the river outlet transitions to negative curvature in the headwaters, with tributaries plotting below the mainstem (Fig. S8). A positive curvature knickpoint forms at the channel outlet and propagates upstream and tributary knickzones collapse with the mainstem and record both timing and magnitude of rock uplift. However, much of the river near the outlet remains near equilibrium following tilting and the positive-curvature slope-break knickpoint that separates the equilibrated section downstream from the section with negative curvature upstream is much less distinct owing to high positive curvature in the uplift field. Further, the pattern in incision mirrors the nonuniform uplift field, with nearly uniform incision rising rapidly with positive curvature near the channel head, rather than the distinctly linear pattern generated by the response to rigid-block tilting.

## 4.9 Rapid pulse of nonuniform rock uplift due to rigid-block forward tilting with heterogeneous lithology

We ran three simulations to explore lithologic effects on transient river profiles resulting from nonuniform uplift: 1) a 1 km instantaneous forward tilt of a 200 km long equilibrium river with a 50 km wide vertical bed of rock with anomalously high erodibility from $70 - 120$ km upstream in the mainstem only (Fig. 5a); 2) a 1 km instantaneous forward tilt of a 200 km long equilibrium river with a 20 km wide vertical bed of rock with anomalously low erodibility from $30 - 50$ km upstream in the mainstem only (Fig. 5b); and 3) a step increase in rigid-block forward tilting from a maximum uplift rate of $50\,\mathrm{mMyr}^{-1}$ to $200\,\mathrm{mMyr}^{-1}$ at the channel head with a 50 km wide vertical bed of rock with anomalously high erodibility from $70 - 120$ km upstream in the mainstem only (Fig. S9). In all simulations, we used $K = 1 \times 10^{-6}\,\mathrm{m}^{0.1}\mathrm{yr}^{-1}$ for the majority of the 1-D river and the tributaries, $K = 1 \times 10^{-5}\,\mathrm{m}^{0.1}\mathrm{yr}^{-1}$ for the vertical bed of more erodible rock, and $K = 2 \times 10^{-7}\,\mathrm{m}^{0.1}\mathrm{yr}^{-1}$ for the vertical bed of less erodible rock.

The transient responses observed in the first two simulations in which tilting is instantaneous are similar to the one described above in section 4.4 in that a positive-curvature slope-break knickpoint forms at the outlet of the mainstem and propagates upstream, hereafter referred to as the "main slope-break knickpoint". However, in the instantaneous tilting simulations with heterogeneous lithology, additional positive-curvature slope-break knickpoints form at the downstream end of the vertical bed of more erodible rock (Fig. 5a) and the upstream end of the vertical bed of less erodible rock (Fig. 5b), hereafter referred to as "rock-type slope-break knickpoints". As these additional knickpoints propagate upstream, they have a similar effect on tributary profiles and incision depth as the main slope-break knickpoint in that they generate the same unique tributary knickzones and a triangular pattern of increasing incision depth upstream.

The important distinction between the main slope-break knickpoint and the rock-type slope-break knickpoints is that rock-type slope-break knickpoints can form immediately following a punctuated tilting event anywhere along the river profile where more erodible rock occurs upstream of less erodible rock. This can result in localized deviations to the pattern in tributary knickzone height described above in section 4.4.2. In both simulations of instantaneous tilting with heterogeneous lithology, large tributary knickzones and local maxima of incision depth occur upstream of the main slope-break knickpoint where propagating rock-type knickpoints have lowered the mainstem.

In the high-erodibility simulation of instantaneous tilting, the rock-type slope-break knickpoint that forms at the downstream end of the vertical bed of more erodible rock rapidly propagates upstream owing to greater celerity in the more erodible rock (Fig. 5a). In the wake of the rock-type knickpoint a reach of quasi-equilibrium steepness is formed. The quasi-equilibrium steepness is equilibrated to the rate of baselevel fall at the downstream end of the vertical bed of more erodible rock, which during the transient is higher than the background rate of rock uplift. Thus the quasi-equilibrium steepness is steeper than the final equilibrium steepness, but is much less than that found immediately post-tilt, which combined with the rapid response in the more erodible bed generates a knickzone upstream of the rock-type slope-break knickpoint. At very early times in the simulation ($t = 0.2$ Myr) large changes in steepness correspond perfectly to lithologic boundaries. Even at $t = 2$ Myr a large increase in steepness occurs at the upstream contact. However, once the main slope-break knickpoint has propagated past the vertical bed of more erodible rock, changes in equilibrium steepness at the lithologic boundaries are much more subtle than those displayed during the transient evolution (Fig. 5a upper row, $t = 3$ Myr).

In the low-erodibility simulation of instantaneous tilting, the vertical bed of less erodible rock slows the propagation of the main slope-break knickpoint and an additional rock-type knickpoint forms at the upstream end of the vertical bed of less erodible rock (Fig. 5b). As the main slope-break knickpoint propagates slowly through the vertical bed of less erodible rock, channel steepness is elevated within the less erodible vertical bed for many time steps. In contrast, the upstream rock-type knickpoint rapidly propagates upstream and leaves a profile with quasi-equilibrium steepness in its wake. Similar patterns in both incision and tributary knickzone drop height emerge upstream of the vertical bed of less erodible rock as in the tilting simulation with uniform $K$. However, within the low-erodibility vertical bed and in particular at the upstream end of the vertical bed, little incision occurs, which results in a local minima in incision depth and tributaries with junctions within the vertical bed that do not exhibit knickzones until the main knickpoint propagates through (Fig. 5b upper and lower row). Once the main knickpoint has propagated through the vertical bed of less erodible rock, the changes in equilibrium channel steepness at the lithologic boundaries are much less prominent than those displayed during the transient evolution.

Similar profile forms develop in the simulation of a step increase in tilt rate with a vertical bed of more erodible rock (Fig. S9). As the river profile rises to a new equilibrium state, the section that traverses the bed of more erodible rock reaches its new equilibrium steepness faster than the remainder of the profile, which, owing to its smaller $K$ value, is rising to a higher fluvial relief and greater channel steepness than the section through the more erodible bed. As a result, the section through the bed has lower channel steepness than the rest of the river profile for the entirety of the simulation and, because this section has reached an equilibrium in which erosion equals uplift, the comparatively lower gradient never propagates upstream from the lithologic contact.

**(a)** Instantaneous rigid-block forward tilting of 1 km at the crest at t = 0 Myr with band of more erodible rock at 70-120 km upstream. Steady uniform background uplift rate of 50 mMyr⁻¹.

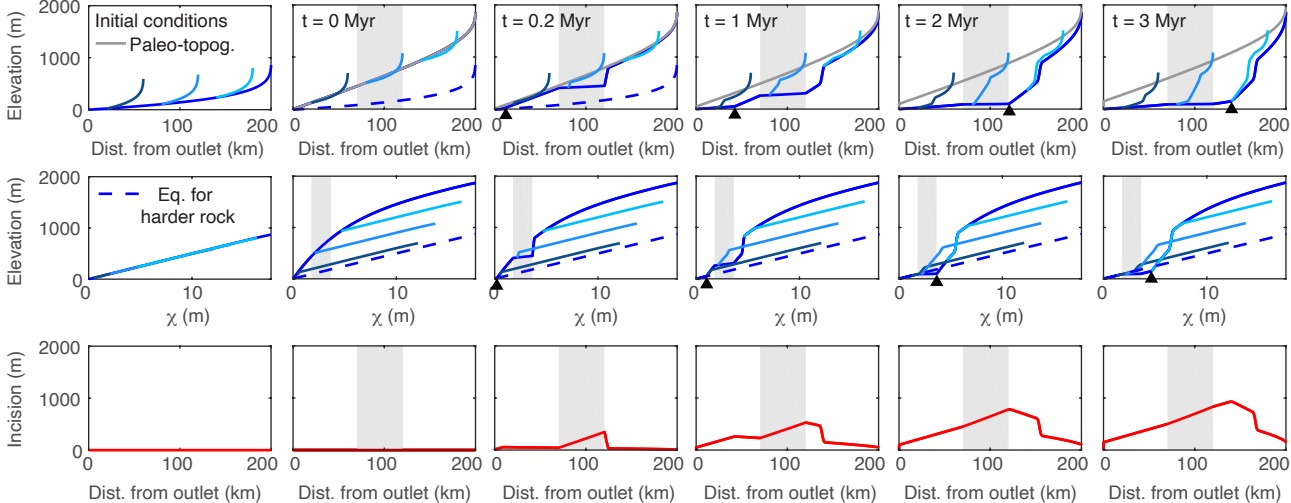

**(b)** Instantaneous rigid-block forward tilting of 1 km at the crest at t = 0 Myr with band of less erodible rock at 30-50 km upstream. Steady uniform background uplift rate of 50 mMyr⁻¹.

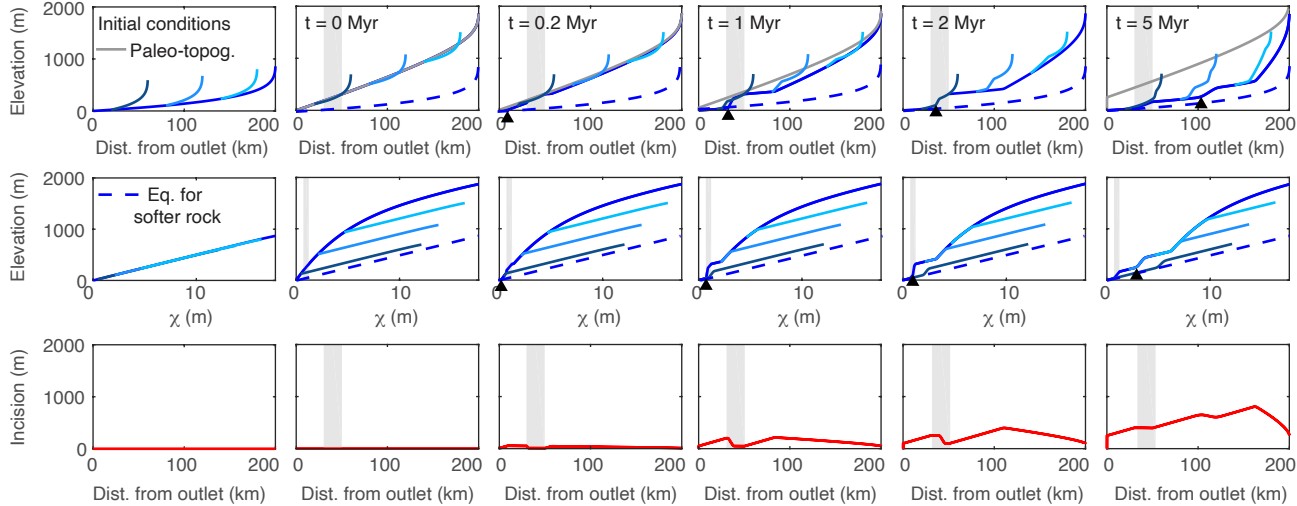

**Figure 5.** Results from a 1-D simulation of an equilibrated river network subject to instantaneous rigid-block forward tilting of 1 km at the channel head with (**a**) a band of more erodible rock at 70-120 km upstream of the mountain front and (**b**) a band of less erodible rock at 30-50 km upstream the mountain front. In both cases, only the mainstem traverses the heterogeneous lithology. Line color, basic description of plots filling each row, and network structure are the same as described in Fig. 2. In both (a) and (b), $K = 1 \times 10^{-6}$ m$^{0.1}$yr$^{-1}$ was used for the majority of the river profile aside from a 50 km vertical band (shown in grey) which was assigned a $K$ value of $1 \times 10^{-5}$ m$^{0.1}$yr$^{-1}$ in (a) and a 20 km band (shown in grey) which was assigned a $K$ value of $2 \times 10^{-7}$ m$^{0.1}$yr$^{-1}$ in (b).

### 4.9.1 Effects of heterogeneous lithology on $\chi$ plots

In both the high-erodibility and low-erodibility simulations, only tributaries with the same path integral of rock type as the mainstem collapse with the mainstem in the $\chi$ plot, in contrast to the simulation of tilting with uniform erodibility in which all tributaries collapse with the mainstem. We calculated $\chi$ with the incorrect assumption of uniform $K$ and, as $\chi$ is an integral quantity calculated in the upstream direction, this results in deviations away from linear scaling between $\chi$ and steady-state elevation not just within the vertical beds of anomalous erodibility, but for the entire profile upstream. Although we could easily calculate $\chi'$ (Willett et al., 2014) with the correct value of $K$ included in the integral, we chose not to in order to illustrate the effects that even small beds of heterogeneous lithology can have on $\chi$ plots when changes in $K$ are unaccounted for. In the high-erodibility simulation, $\chi$ values are higher than $\chi'$ would be for the mainstem and tributaries upstream of the vertical bed of more erodible rock and these profiles will thus plot below the equilibrium line for uniform $K$ (Fig. 5a middle row and Fig. 6). In the low-erodibility simulation, $\chi$ is lower than $\chi'$ for the mainstem and tributaries upstream of the vertical bed of less erodible rock and these will thus plot above the equilibrium line for uniform $K$ (Fig. 5b middle row and Fig. 6).

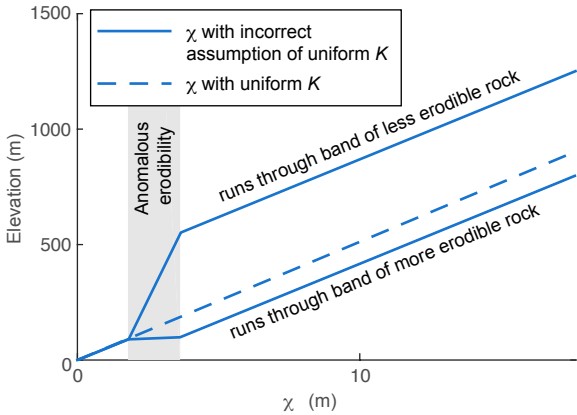

**Figure 6.** Influence of unaccounted for vertical bed of more or less erodible rock on equilibrium $\chi$ plots for a 200 km river. Dashed $\chi$-z profile has a uniform $K$ of $1 \times 10^{-6}$ m$^{0.1}$yr$^{-1}$. The lower solid profile has a vertical bed of more erodible rock with $K = 1 \times 10^{-5}$ m$^{0.1}$yr$^{-1}$ at $60 - 110$ km upstream as compared to upper solid profile that has a vertical bed of less erodible rock with $K = 2 \times 10^{-7}$ m$^{0.1}$yr$^{-1}$ at $60 - 110$ km upstream but both were calculated assuming $K$ is uniformly $1 \times 10^{-6}$ m$^{0.1}$yr$^{-1}$.

### 4.9.2 Estimating tilt magnitude from rock-type knickzone geometry in profiles with heterogeneous lithology

At early time steps in the simulation of instantaneous tilting with a vertical bed of more erodible rock, the river profile has a deeply incised section that appears as a triangular "bite" out of the tilted river profile where the river crosses the more erodible

rock. If the slope of the quasi-equilibrium reach (as described in the section above) in the vertical bed of more erodible rock is assumed to equal the pre-tilt river slope over the erodible rock, the geometry of this bite reflects the magnitude of tilt such that:

$$\theta_t = \theta_T - \theta_{qer} \tag{9}$$

$$\theta_T = \arctan\left(\frac{H_T}{L_T}\right) \tag{10}$$

$$\theta_{qer} = \arctan\left(\frac{H_{qer}}{L_{band}}\right) \tag{11}$$

$$\theta_{ssr} = \arctan\left(\frac{H_{ssr}}{L_{band}}\right) \tag{12}$$

where $\theta_t$ is tilt angle, $\theta_T$ is the angle off horizontal over the bite, $\theta_{qer}$ is the angle off horizontal over the quasi-equilibrium reach, $\theta_{ssr}$ is the angle off horizontal of the initial steady-state river profile, $H_T$ is the total elevation drop over the bite, $H_{qer}$ is the elevation drop over the quasi-equilibrium reach in the band, $H_{ssr}$ is the elevation drop of the initial steady-state river over the band, $L_T$ is the total length of river along the bite, and $L_{band}$ is the length of river through the band (Fig. 7).

To test whether tilt magnitude could be extracted from the geometry of these mid-profile rock-type knickzones, we ran ten simulations with vertical beds of rock with $K = 1 \times 10^{-5}$ m$^{0.1}$yr$^{-1}$ of varying length in which we tilted the same initial equilibrium profile by the same tilt magnitude. We ran simulations with low background uplift rate ($U = 50$ mMyr$^{-1}$) to achieve a low gradient initial river slope to then tilt $0.5°$ (Fig. 7b) and a high background uplift rate ($U = 200$ mMyr$^{-1}$) to achieve a high gradient initial river slope subject to then tilt $0.5°$ (Fig. 7c). In all simulations the initial profile was equilibrated to the same nonuniform values of $K$ used in the simulation. We identified the upstream and downstream ends of the rock-type slope-break knickpoint defining the knickzone base by identifying the appropriate peak and trough in profile curvature on the $\chi$ plot and stopped the simulation when the knickpoint was at the upstream end of the band and the quasi-equilibrium reach stretched only the full length of the band (e.g., Fig. 7a).

We were able to estimate the tilt angle to within a tenth of a degree using equation 9 by calculating $\theta_T$ and $\theta_{qer}$ from measurements of $H_T$, $L_T$, and $H_{qer}$ made at the end of each simulation. Estimates of tilt angle do not vary with band width (Fig. 7b and c). In all simulations, the method underestimates tilt magnitude because the quasi-equilibrium slope is greater than the initial equilibrium river slope through the band owing to the fact that the quasi-equilibrium slope is adjusted to a transient rate of base level fall that is greater than the background rate of rock uplift (e.g., Fig. 7b and c). We were also able to measure tilt angle in simulations using a larger magnitude tilt (Fig. S10a) or a smaller difference in erodibility between the bands, but with greater error (Fig. S10). The error in the tilt estimate increases with increasing tilt magnitude and decreasing erodibility because the gradient of the quasi-equilibrium reach becomes greater than the initial river slope across the erodible band in both

of these cases (Fig. S10). We found that this method can also be used to estimate tilt magnitude in the case of a step increase in tilt rate with a band of more erodible rock, though the method will always underestimate tilt magnitude in this case as a step increase in tilt rate will increase the slope of the segment of river that traverses the band.

## 4.10  Rapid pulse of nonuniform rock uplift with alternative initial and final conditions

The geomorphic signatures of the transient response to tilting outlined above apply under the conditions that tilting is the predominant perturbation and that the initial condition is a river profile near equilibrium. These conditions may not be met in real landscapes where multiple perturbations may occur with similar timing or where rivers may not have reached equilibrium prior to tilting. To illustrate which geomorphic signatures of tilt outlined above are most robust, we simulated a rapid pulse of nonuniform uplift due to tilting with different initial conditions and histories of background rock uplift rates.

We simulated an instantaneous tilt of an equilibrium profile with a simultaneous step increase in uniform background rock uplift rate from 50 $\mathrm{mMyr}^{-1}$ to 200 $\mathrm{mMyr}^{-1}$ such that final fluvial relief was greater than initial fluvial relief (Fig. S11). The initial condition was a river profile equilibrated to a uniform background uplift rate of 50 $\mathrm{mMyr}^{-1}$ and an erodibility of $1 \times 10^{-6}\, \mathrm{m}^{0.1}\mathrm{yr}^{-1}$. The new equilibrium steepness is similar to the maximum steepness of the tilted river profile, obscuring the slope-break knickpoint (Fig. S10). With a larger magnitude step increase in uplift rate, the sign of the slope-break knickpoint would flip from positive to negative. The pattern in channel steepness upstream of the mainstem knickpoint remains a robust signature of a punctuated tilting event despite the step change in uplift rate, with mainstem maximum steepness still occurring just upstream from the knickpoint and the mainstem plotting above the tributaries in the $\chi$ plot. A simultaneous step change in background uplift rate also affects estimates of surface uplift from the magnitude of incision below paleo-topography markers in that incision depth divided by surface uplift is inversely correlated with fractional change in fluvial relief (Fig. S12). For example, a simultaneous step increase in uniform background rock uplift rate along with tilting results in less incision and therefore a lower estimate of surface uplift.

To demonstrate how disequilibrium initial conditions affect the geomorphic signatures of tilt, we ran two simulations: 1) a uniform pulse of uplift at $t = 0$ Myr and an instantaneous tilt of 1 km at the channel head at $t = 5$ Myr (Fig. S13a); and 2) an instantaneous tilt of 1 km at the channel head at $t = 0$ Myr and another tilt of the same magnitude at $t = 5$ Myr (Fig. S13b). Following the second perturbations at $t = 5$ Myr, the positive-curvature mainstem knickpoint and the pattern in channel steepness upstream of the knickpoint remain robust signatures of tilt that are recognizable in both simulations despite the additional disequilibrium forms in the river profiles. Incision depth in each case reflects the combined surface uplift that has occurred over both perturbations. Thus, the expected pattern emerges from the simulation of two tilting events, but the expected pattern is obscured in the simulation in which the river is first subjected to a uniform pulse of uplift.

## 5  Modeling entire river network response to a rapid pulse of nonuniform rock uplift due to forward tilting

To investigate how landscapes respond to instantaneous tilting beyond the idealized channel network explored above, we used a 2-D numerical landscape evolution model described by Perron et al. (2008) to solve equation 1 and to simulate a $1°$ forward

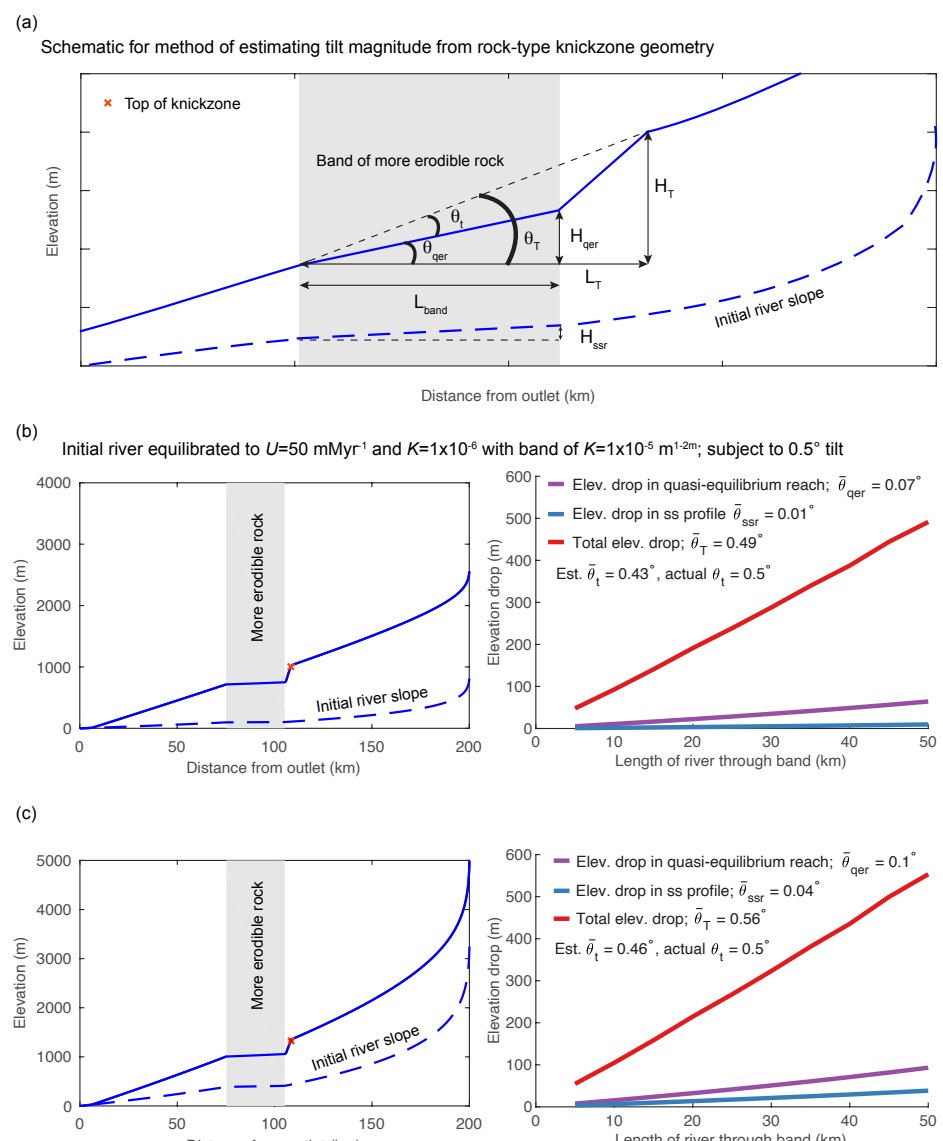

**Figure 7.** Measurements of tilt angle from the geometry of local rock-type related knickzones in a 1-D model of instantaneous forward tilt at $t = 0$ with vertical beds of more erodible rock of increasing length at 75 km upstream in a 200 km long river. Erodibility in the vertical bed of more erodible rock is 10 times greater ($K = 1 \times 10^{-5}$ m$^{0.1}$yr$^{-1}$) than the rest of profile ($K = 1 \times 10^{-6}$ m$^{0.1}$yr$^{-1}$) and initial river profile is equilibrated to these nonuniform values of $K$. (**a**) Schematic for variables used in equations 9-12. (**b**) Example of end time step transient river profile for simulations using low gradient initial river and subject to $0.5°$ tilt (left) and measurements made from ten simulations with this set up but variable band width (right). (**c**) Example of end time step transient river profile for simulations using steep initial river and subject to $0.5°$ tilt (left) and measurements made from ten simulations with this set up but variable band width (right).

tilt of a landscape (down towards the mainstem outlets) equilibrated to an uplift rate of $50~\mathrm{mMyr^{-1}}$, an erodibility coefficient of $1 \times 10^{-6}~\mathrm{m^{0.1}yr^{-1}}$ and river concavity, $\theta$, of $0.45$, with $n = 1$. We ran two simulations: 1) an instantaneous $1°$ forward tilt (Fig. 8a) and 2) step change from low magnitude uniform uplift to rapid continuous tilting of magnitude $1°$ over 3 Myr (Fig. 9a). We calculated $\chi$ for the main river networks that drain towards the lower boundaries and removed all rivers draining to

other boundaries from the analysis. $\chi$ plots for the simulated river networks show a similar pattern as the simplified network of linked 1-D profiles described above, with negative curvature along the majority of the profile, tributaries plotting below the mainstem, and a positive curvature slope-break knickpoint propagating upstream (Fig. 8b and 9b).

## 5.1 Estimating tilt magnitude from azimuth-gradient relationship in stream segments

We measured azimuth, $k_{sn}$, and gradient of 3 km long river segments for the entire river network in both the 2-D instantaneous
tilt case (Fig. 8) and the 2-D continuous tilting case (Fig. 9). We plotted $k_{sn}$ and gradient against azimuth and calculated running medians for both using a $10°$ moving window (green line in Fig. 8 and 9 c,d) to compare to the expected post-tilt signature of a cosine function with a maximum in the tilt direction of $180°$ (blue lines in Fig. 8 and 9 d). The distribution of azimuth across all 3 km segments was relatively uniform throughout the simulation (e.g., Fig. 8b inset). In the instantaneous tilt case, clear signatures of tilt appear immediately after tilting in both the relationships between azimuth and the moving median of $k_{sn}$ and
gradient as stream segments flowing in the direction of tilt ($180°$) have higher gradient and higher $k_{sn}$. However, the signature of increased gradient and $k_{sn}$ in segments flowing in the direction of tilt is much stronger than the signature of decreased gradient and $k_{sn}$ in segments that were back-tilted (Fig. 8c and d), likely because many back-tilted segments were depressions that were filled during flow routing immediately post-tilt. Both signatures in gradient and $k_{sn}$ are obscured by the ongoing transient erosional response by $t = 4$ Myr and, instead of returning to pre-tilt conditions, median gradient and $k_{sn}$ are greater
at every azimuth at $t = 4$ Myr (Fig. 8). Unlike the response to uniform uplift, which is dependent on the channel response time, channels respond instantaneously to tilt because steepness has changed everywhere in the river network (Fig. 3b). In the rapid continuous tilting case, signatures of tilt appear in both azimuth and gradient around $t = 3$ Myr but are obscured by $t = 6$ Myr (Fig. 9). However, in a simulation of continuous tilting with a lower magnitude tilt rate, no signatures of tilt appear at any time steps (Fig. S14).

## 5.2 River profile analysis of 2-D simulated river network

For the instantaneous tilting case, we analyzed mainstem and tributary river profiles at $t = 5$ Myr and made plots and measurements analogous to those made from the linked 1-D models (Fig. 10). The transient fluvial response to instantaneous tilting in the 2-D river network generates equivalent disequilibrium forms as those generated in the 1-D simulation: 1) a positive-curvature slope-break knickpoint that separates a downstream section with uniform equilibrium steepness from an upstream
section with nonuniform steepness; 2) negative curvature on the $\chi$ plot in which the mainstem plots above tributaries; and 3) tributary knickzones with nonuniform steepness that collapse with the mainstem on the $\chi$ plot (Fig. 10). We identified the tops of tributary knickzones as the point where tributaries diverge from the mainstem on the $\chi$ plot (filled black markers in Fig. 10c and d). Tributary junction angle does not impact the magnitude of tributary knickzone drop height nor the degree to

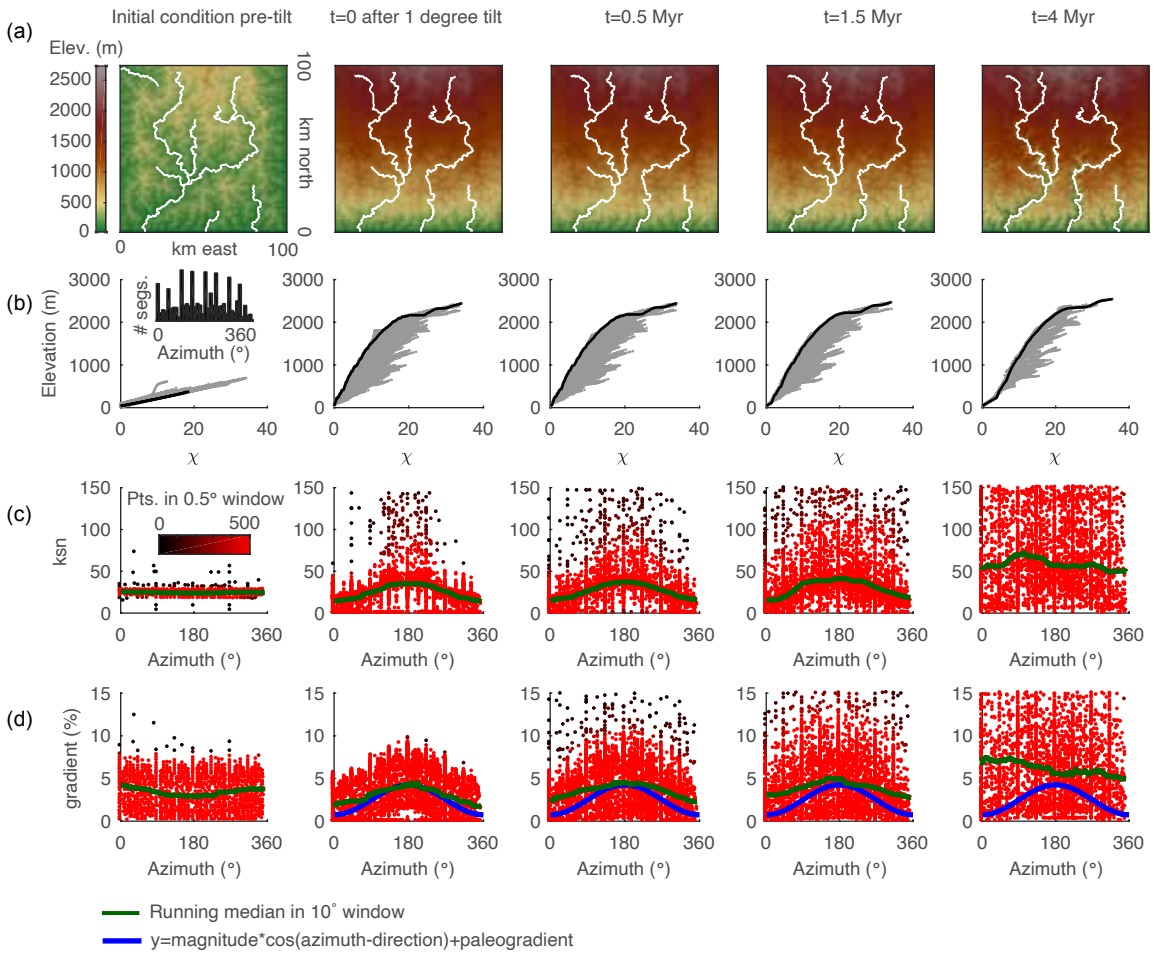

**Figure 8.** Evidence of tilt in relationships between azimuth and gradient and $k_{sn}$ of 3 km long stream segments of river networks generated in a 2-D simulation of a $1°$ instantaneous rigid-block forward tilt. (**a**) DEMs with mainstems of the stream network shown in white. (**b**) $\chi$ plots for entire river network with points on the mainstem shown in black and all tributary points shown in gray. (**c**) Plot of segment $k_{sn}$ against azimuth with point color indicating density of points within $0.5°$ azimuth window. (**d**) Plot of segment gradient against azimuth colors as in **c**. Inset in **b** shows the distribution of azimuth of 3 km stream segments demonstrating that the distribution of segments is approximately uniform. Blue line shows cosine function with known parameters of tilt perturbation in the simulation. Green line shows $10°$ wide moving-window medians.

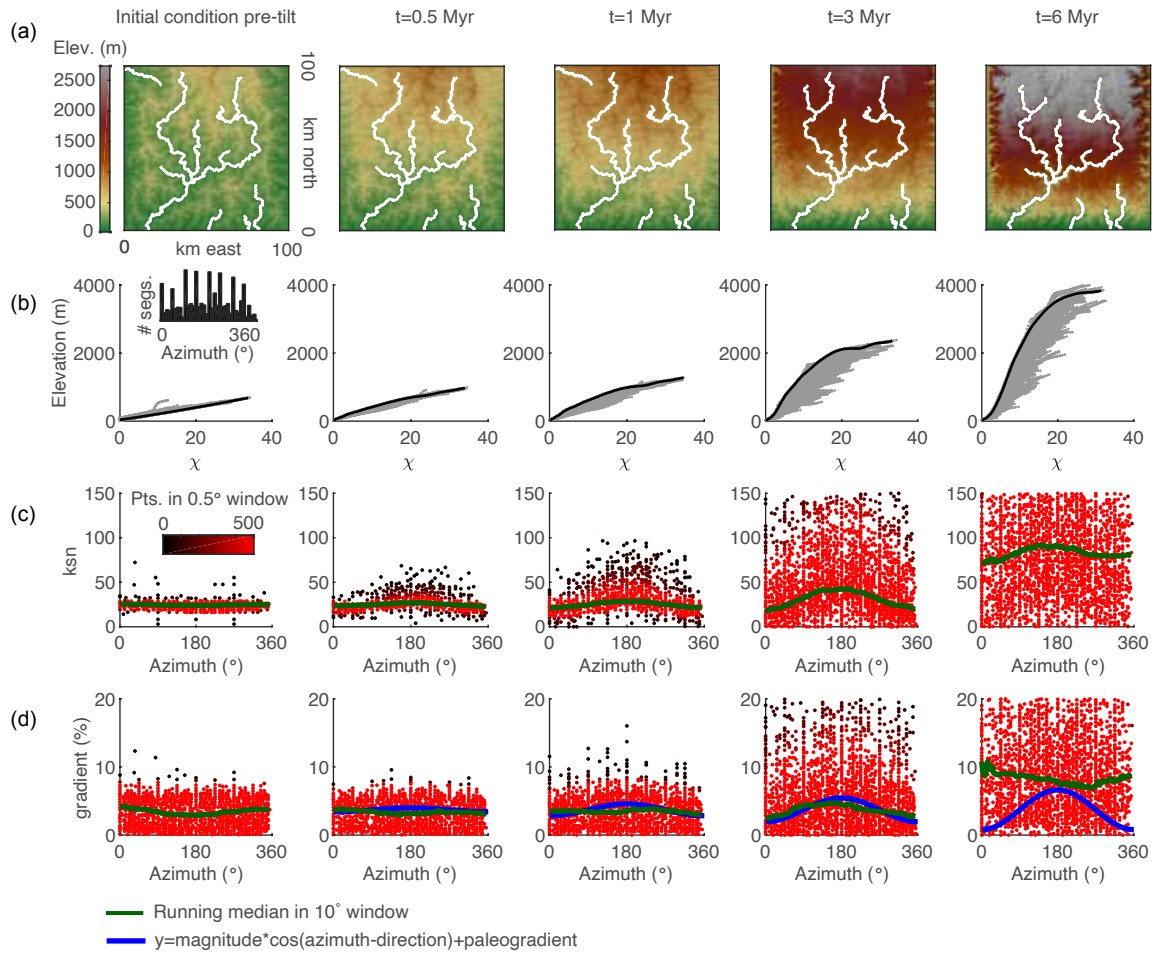

**Figure 9.** Evidence of tilt in relationships between azimuth and gradient and $k_{sn}$ of 3 km long stream segments of river networks generated in a 2-D simulation of a $1°$ rigid-block forward tilt that occurs over 3 Myr. (**a**) DEMs with mainstems of the stream network shown in white. (**b**) $\chi$ plots for entire river network with points on the mainstem shown in black and all tributary points shown in gray. (**c**) Plot of segment $k_{sn}$ against azimuth with point color indicating density of points within $0.5°$ azimuth window. (**d**) Plot of segment gradient against azimuth colors as in **c**. Inset in **b** shows the distribution of azimuth of 3 km stream segments demonstrating that the distribution of segments is approximately uniform. Blue line shows cosine function with known parameters of tilt perturbation in the simulation. Green line shows $10°$ wide moving-window medians.

which tributary knickzones collapse with the mainstem. Mainstem channel steepness upstream of the knickpoint and tributary channel steepness both within and upstream of the knickzone is impacted by the azimuth of stream segments, but this does not impact estimates of timing or magnitude.

We were able to recover tilt timing from tributary knickzones by subtracting the $\tau$ value at the mainstem-tributary junction from the $\tau$ value at the top of the corresponding tributary knickzone (mean $\tau = 4.8$ vs. 5Myr). We were also able to recover total magnitude of rock uplift at the crest using methods outlined with the 1-D model, though both methods slightly underestimated tilt magnitude (Fig. 10a,c). We regressed tributary knickzone drop height against the Euclidean distance to the mountain front measured at mainstem-tributary junctions using only tributaries with junctions downstream of the mainstem slope-break knickpoint. This yielded a slight underestimate of $0.8°$ or $\sim 1,575$ m of rock uplift at the crest compared to the known $1.0°$ tilt or $\sim 1,750$ m of rock uplift at the crest (Fig. 10c inset). To make an analogous measurement to the incision measurements in the 1-D model, we measured canyon incision below the proximal low-relief upland surface by running a 10 km wide swath profile down the mainstem and calculating the difference between the mean elevation of the upland surface and the mainstem river. We then regressed these incision measurements against Euclidean distance from the mountain front at the corresponding river nodes, which also yielded a slight underestimation of the magnitude of surface uplift ($\sim 1,560$; Fig. 10a inset), likely owing to erosion of the upland surface. As expected, regression of $\chi$-elevation data downstream of the mainstem knickpoint yielded a $k_{sn}$ value that perfectly reflects the input value of $K$ and the background uplift rate (Fig. 10d).

### 5.3 River network reorganization in response to rapid tilting

Network topologic changes via stream capture oriented in the direction of tilt characterized river network reorganization following the instantaneous tilting event. We observed two types of stream captures in the simulated river network. The first occurred immediately after tilting as a result of our routing algorithm that filled in closed depressions in back-tilted reaches until an outlet could be found, commonly a low point in the bounding ridge. Although these captures resulted purely from our particular routing algorithm, the analogous natural processes would be aggradation of sediments in back tilted reaches that eventually raise the bed elevation to the point of spilling over the ridge to complete a stream capture (e.g. Fig. 11b). At later times, the second type of stream capture occurred when knickzones in over-steepened streams flowing in the tilt direction eroded headward and breached ridges separating more rapidly eroding streams in the tilt direction from more slowly-eroding back-tilted reaches or reaches flowing oblique to the tilt direction (e.g., Fig. 11c). At $t = 4$ Myr, the captured stream network appear as regions with anomalous topology, including barbed tributaries and streams that cut through ridges (Fig. 11c).

### 6 Example of the geomorphic expression of a punctuated tilting event from the Sierra Nevada, California

We analyzed mainstem and tributary profiles of the Middle Fork American-Rubicon River basin, in the northern part of the Sierra Nevada (see Fig. 12 for location), as an example of how the geomorphic signatures of the transient response to rapid tilting as outlined above can be used in concert to reconstruct timing and magnitude of a tilting event in a range long thought to have been tilted westward in the late Cenozoic. The Rubicon River is the longest tributary to the Middle Fork American

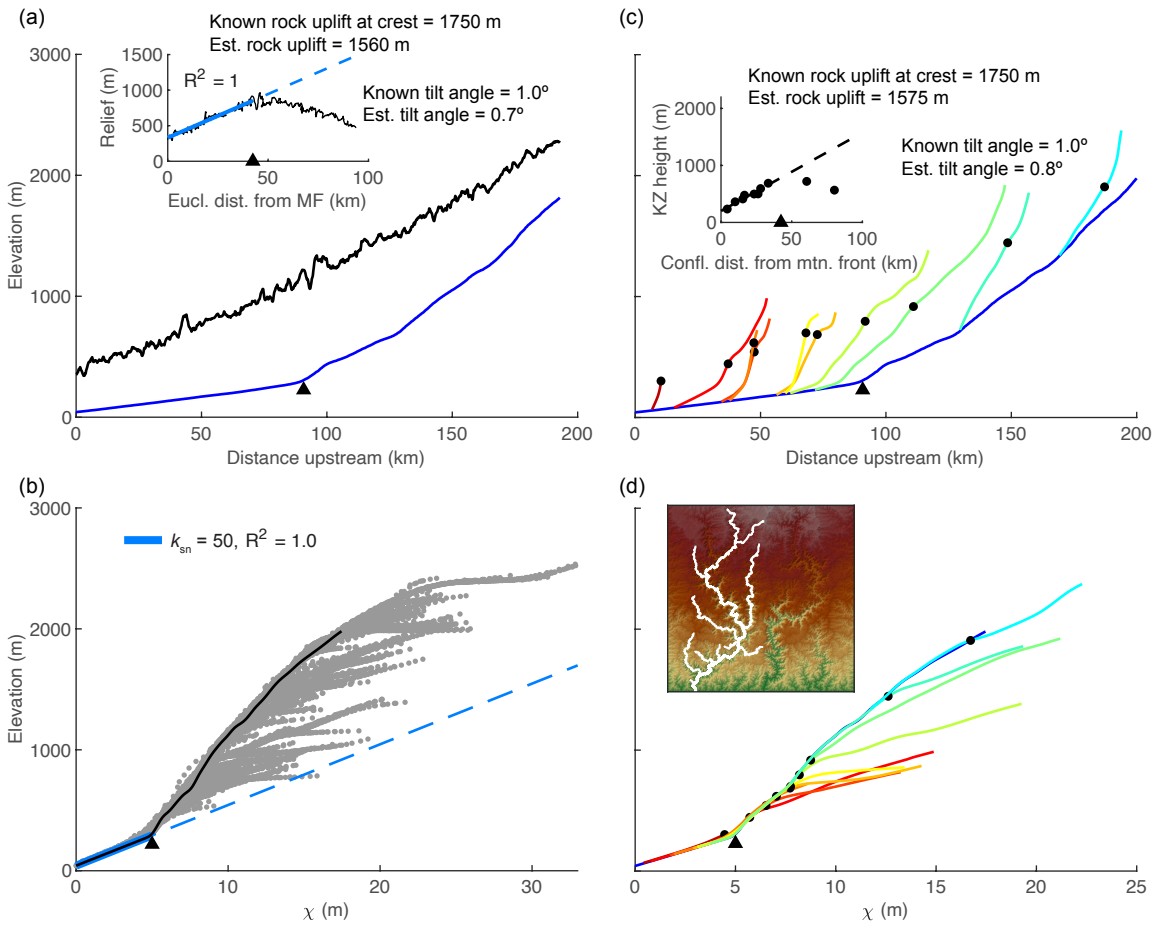

**Figure 10.** River profile analysis of 2-D simulated river network at $t = 5$ Myr following instantaneous rigid-block forward tilting of $1°$ which raised the upper boundary of the grid $\sim 1,750$ m. (**a**) Longitudinal profile of the mainstem (blue) with mean elevation of the proximal upland surface within a 10 km wide swath (black). Inset shows regression of canyon incision (approximated as canyon relief) with distance from the mountain front for points downstream of the mainstem slope-break knickpoint. (**b**) $\chi$ plot of the mainstem and tributaries with regression of data downstream of the mainstem slope-break knickpoint. (**c**) Longitudinal profiles and (**d**) $\chi$ profiles for the mainstem and major tributaries. Mainstem is shown in dark blue; tributaries change from warm to cool colors as distance upstream from the mountain front to the tributary confluence increases. Inset plot in (c) shows measurements of tributary knickzone drop height against tributary confluence distance from the mountain front along with a linear regression of points with confluences downstream of the mainstem slope-break knickzone. Inset in (d) shows DEM with analyzed mainstem and tributaries in white.

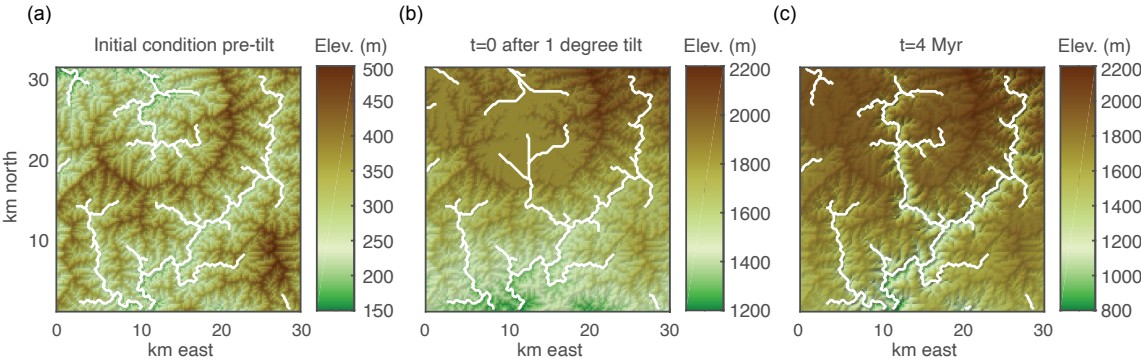

**Figure 11.** Examples of stream capture in the tilt direction from 2-D simulation of a 1 degree instantaneous tilt at $t = 0$ Myr. Maps show DEMs overlaid with stream network in white for (**a**) initial condition pre-tilt, (**b**) $t = 0$ after 1 $^\circ$ tilt, and (**c**) at $t = 4$ Myr.

River. For simplicity we refer to the drainage we analyzed as the Middle Fork American River basin hereafter, although there is another, smaller tributary called the Middle Fork American that was not part of our analysis. The Sierra Nevada is a $\sim 600$ km long $\sim 100$ km wide range striking northwest-southeast through eastern California (Fig. 12) that is thought to be tilted westward $\sim 0.5 - 1°$ in the northern Sierra (Jones et al., 2004; Lindgren, 1911; Unruh, 1991; Wakabayashi, 2013;

Wakabayashi and Sawyer, 2001) and $\sim 1 - 2°$ in the southern Sierra (Huber, 1981; McPhillips and Brandon, 2012). Evidence from a range of fields points to a late Cenozoic pulse of uplift in the Sierra Nevada (Christensen, 1966; Clark et al., 2005; Ducea and Saleeby, 1996; Huber, 1981; Jones et al., 2004; Lindgren, 1911; McPhillips and Brandon, 2012; Pelletier, 2007; Stock et al., 2004; 2005; Unruh, 1991; Wakabayashi, 2013; Wakabayashi and Sawyer, 2001; Yeend, 1974). However, recent geochemical studies suggest that the Sierra has been a high topographic feature since the Cretaceous (Cassel et al., 2009; 2012;

2014; Crowley et al., 2008; Hren et al., 2010; Mix et al., 2016; Mulch et al., 2006; 2008; Poage and Chamberlain, 2002) and thus the debate surrounding the uplift history of the Sierra Nevada remains active and unresolved.

## 6.1 Disequilibrium form of the mainstem Middle Fork American consistent with late Cenozoic tilting

A positive-curvature slope-break knickpoint is evident in the mainstem Middle Fork American River at 110 km upstream from the mountain front and 330 m elevation (black triangle in Fig. 13a and b). Downstream of the slope-break knickpoint, channel

steepness is low and uniform, whereas upstream of the knickpoint, channel steepness is nonuniform with maximum steepness occurring just above the knickpoint and decreasing upstream. Both the mainstem and the cloud of points defined by the tributaries exhibit negative curvature on the $\chi$ plot, with the mainstem plotting near the top of the cloud. These disequilibrium profile forms are similar to the signatures of the transient response to a rapid forward tilting event displayed in the 1- and 2-D simulations outlined above that then returns to a uniform uplift rate or a lower background tilt rate (compare Fig. 13b with Fig.

3b, Fig. 4 and Fig. 8b).

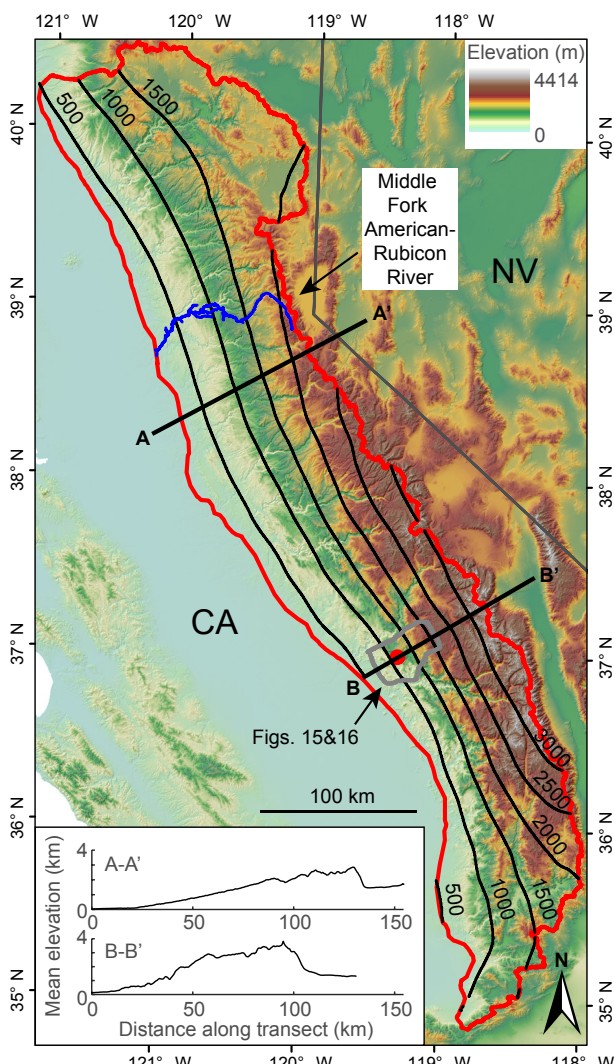

**Figure 12.** Digital elevation model (DEM) of eastern California and western Nevada. The extent of the western slope of the Sierra Nevada is enclosed by a red line with the eastern boundary defining the crest and the western boundary defining the mountain front. The Middle Fork American-Rubicon River basin is shown with a blue line, upland interfluve surface analyzed in Fig. 15 is enclosed by a thick grey line, stream capture shown in Fig. 16 is shown with red marker, and 500 m contours of elevation after smoothing DEM with a 50 km$^2$ moving-window mean are shown with black lines. State lines, shown in grey, of California and Nevada for reference. Inset shows elevation swaths for corresponding transects $A - A'$ and $B - B'$.

Estimating the regional equilibrium river concavity or the reference concavity, $\theta_{ref}$, is challenging in landscapes with transient river networks as both standard techniques using slope-area plots (e.g., Wobus et al., 2006) or $\chi$ plots (Perron and Royden, 2013) of entire basins cannot be applied given the disequilibrium state of Sierra rivers. Although more advanced techniques

of estimating $\theta_{ref}$ from $\chi$ plots have been developed (e.g., Mudd et al., 2018), these methods still require the assumption of co-linearity, which as we show above, is not retained in all transient responses. Therefore, we used slope-area plots of mainstem rivers downstream of slope-break knickpoints to estimate that $\theta = 0.41 - 0.48$, and thus we use $\theta_{ref} = 0.45$ for all analyses in the Sierra. In doing so we assume that equilibrium river concavity is uniform throughout the river network in the Sierra

Nevada, but not that modern river concavity is uniform.

    We calculated $K$ for the Sierra batholith using a selection of published $^{10}$Be-derived denudation rates and their associated $k_{sn}$ values. From a compilation of erosion rates for basins on an unglaciated, low-relief upland surface in the Sierra (Callahan et al., 2019), we selected basins with $R^2 > 0.9$ for linear regression of $\chi$ elevation data above the sampling point for catchment-average erosion rates such that erosion rates should be in approximate equilibrium with uplift rates (Fig. S15). This yielded

21 basins from which we calculated a mean $K$ of $1 \times 10^{-6} \pm 0.03 \times 10^{-7}$ m$^{0.1}$yr$^{-1}$. With $K = 1 \times 10^{-6}$ and assuming $K$ is relatively uniform, $\chi$ values can be interpreted as knickpoint travel times, $\tau$, in millions of years (equations 4 and 6). The mainstem slope-break knickpoint occurs at $\chi = 5$, thus we estimate that rapid tilting in the northern Sierra ceased $ca.$ 5 Ma. With values of $n \neq 1$, $\tau$ cannot be calculated directly. However, based on 1-D model results, we can see that if $n = 2/3$ but $n = 1$ is assumed when analyzing river profiles, response times increase and thus 5 Ma would be an underestimate (Fig. S4),

whereas if $n = 5/3$ but $n = 1$ is assumed when analyzing river profiles, response times decrease and thus 5 Ma would be an overestimate (Fig. S5).

    Widespread volcanism blanketed the northern Sierra from the mid-Miocene through the Pliocene (Bateman and Wahrhaftig, 1996; Busby et al., 2008; Christensen, 1966; Curtis, 1953; Durrell 1966; Slemmons 1966). These deposits still blanket modern interfluves throughout the northern Sierra along with Eocene-aged fluvial gravels (Cassel and Graham, 2011 and references

therein). The depth of incision into basement rock underlying Mio-Pliocene volcanic deposits and Eocene fluvial gravels has been proposed to reflect accumulated rock uplift since their deposition (Wakabayashi, 2013; Wakabayashi and Sawyer, 2001). We measured incision of the mainstem into basement rock below these Cenozoic deposits and plotted it against Euclidean distance from the mountain front (Fig. 13a inset). This revealed that incision depth increases linearly below the mainstem slope-break knickpoint with respect to distance from the mountain front as in the simulations of rapid tilt (compare Fig. 13a

inset with Fig. 3b lower row and Fig. 4a and b lower row). By calculating a linear regression of incision data (both incision below volcanics and below fluvial gravels) downstream of the mainstem knickpoint as described in section 4.4.2 (Fig. 3b lower row), we estimated a $0.8°$ tilt or, by projecting this tilt angle the $\sim 90$ km width of the range, $\sim 1,200$ m of late Cenozoic surface uplift at the crest (Fig. 13a inset).

    Local deviations in the mainstem longitudinal and $\chi$ profiles as well as the pattern of incision into basement rock may be

caused by heterogeneous lithology and/or glacial erosion. At 160 km upstream, the Middle Fork American travels for 9 km through metamorphosed Jurassic marine rocks. This reach has anomalously low channel steepness and occurs downstream of a steep knickzone at the contact with Mesozoic granitic rocks (Fig. 13a). This step exhibits similar characteristics as steps formed in the simulation of instantaneous tilt with a vertical bed of more erodible rock (compare Fig. 13a with Fig. 5a). From the geometry of this knickzone (Fig. 7a and equation 9), we estimated a tilt of $0.7°$ or $\sim 1,100$ m of surface uplift at the crest.

Such notable profile deviations are not observed in other locations where the river passes through Jurassic marine rocks, likely

because bands are downstream of the mainstem slope-break knickpoint where equilibrium changes in steepness across rock types are much more subtle or because this is a grouping of multiple formations (Ludington et al., 2005) and is thus likely comprised of rocks with a range in erodibility.

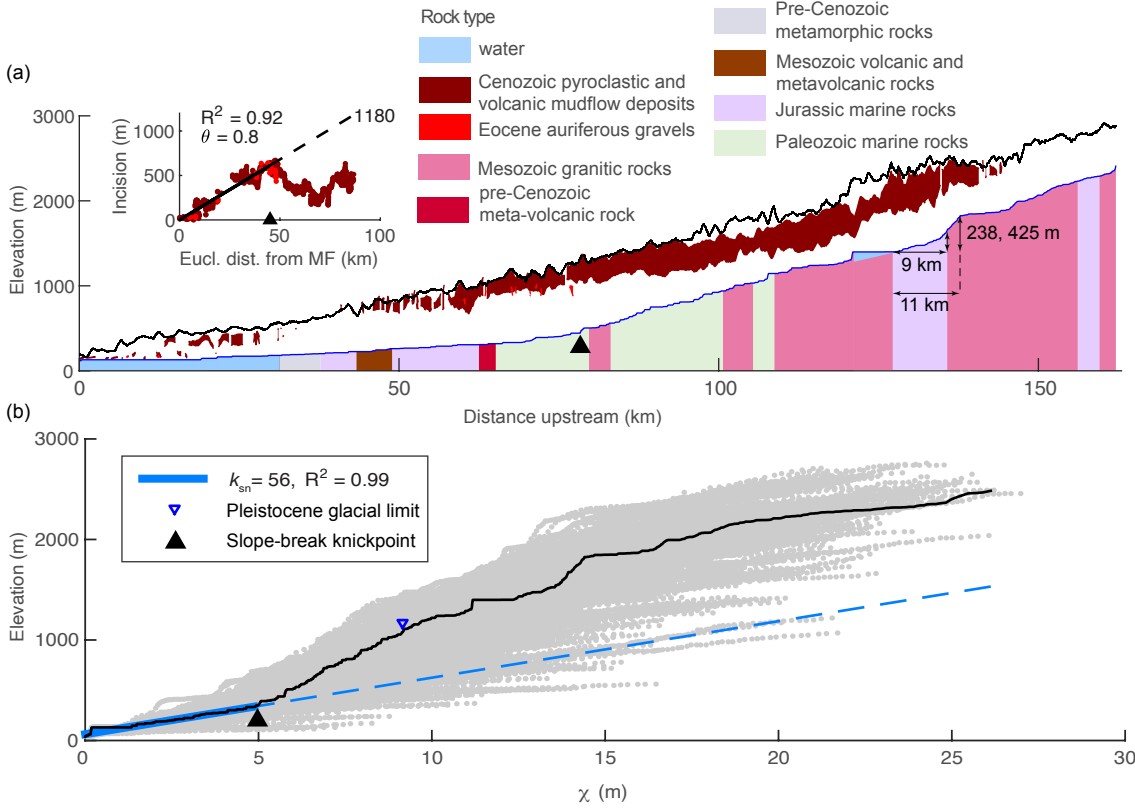

**Figure 13.** Mainstem river profiles for the Middle Fork American-Rubicon River, Sierra Nevada. (**a**) Longitudinal profile (blue line) for mainstem upstream of mountain front, with surface geology, proximal upland surface topography (black line), Cenozoic volcanic deposits within the basin (brown polygons below black line) and Eocene auriferous gravels (red polygons). Cenozoic volcanic cover and and Eocene auriferous gravels were taken from 1:250,000 geology maps (Saucedo and Wagner, 1992; Wagner et al., 1981). Other surface geology was taken from 1:500,000 geology maps (Ludington et al., 2005) and isolated bands of rock <2.5 km river length were removed. Surface geology is shown with vertical contacts to schematically represent the generally steep dips of contacts found in the Sierra. Inset plot shows valley incision into basement calculated as the difference between the minimum elevation of proximal Cenozoic volcanic deposits and the modern river profile with a linear regression of data below the mainstem slope-break knickpoint that was projected to the river's headwaters. (**b**) $\chi$ plot with mainstem $\chi$ profile (black) over data from entire basin (grey). Thick blue line highlights data below mainstem slope-break knickpoint that was used in a linear regression that was then projected to the headwaters (dashed blue line).

## 6.2 Middle Fork American tributary knickzones reflect late Cenozoic tilting

We analyzed all major tributaries to the Middle Fork American River that traverse terrain downstream of the Pleistocene glacial limit as mapped by Gillespie and Clark (2011) and we found that all tributaries have large knickzones between the tributary junction and an upstream section with low channel steepness (Fig. 14). These knickzones have a similar unique character as those formed in the simulations of rapid forward tilting, with anomalously high but nonuniform steepness that is similar to that of the mainstem upstream of the tributary confluence (compare Fig. 14 with Fig. 3b and Fig. 4). We identified the upstream end of tributary knickzones visually and measured their drop height as the elevation difference between the upstream end and the confluence with the mainstem. Tributary knickzone drop height increases up to the mainstem slope-break knickpoint and decreases upstream, similar to the pattern generated in the simulations of rapid forward tilting (compare Fig. 14 with Fig. 3b and Fig. 4).

We categorized tributary knickzones as those that appear visually to collapse with the mainstem on the $\chi$ plot and thus likely behave similarly to tributaries in the simulation of instantaneous tilt, and those that do not collapse with the mainstem on the $\chi$ plot and thus are likely influenced by additional factors that allow the knickzones to move in a manner not captured by the simple stream power law employed here. As described in section 4.4.1, the timing of the cessation of rapid tilting can be recovered by subtracting the $\tau$ value at the corresponding tributary confluence from the $\tau$ value at the upstream end of the knickzone. By using only knickzones that collapse with the mainstem on the $\chi$ plot, we estimated the rapid tilting event in the Sierra ended $ca.$ 1.9 Ma. By comparing disequilibrium profile forms in the Sierra with those generated by tilts of varying duration, in particular mainstem steepness upstream of the slope-break knickpoint and tributary knickzone steepness, we estimate that the duration of this tilting event was less than 5 Myr (compare Fig. 14 with Fig. 3b and Fig. 4).

We further categorized the tributary knickzones that collapse with the mainstem on the $\chi$ plot into two groups: those with tributary confluences downstream of the mainstem knickpoint and those with confluences upstream of the knickpoint. The drop height of tributary knickzones that collapse with the mainstem on the $\chi$ plot and that have junctions downstream of the mainstem slope-break knickpoint should reflect the magnitude of surface uplift that occurred as a result of the rapid tilting event as described in section 4.4.2 (e.g., Fig. 3b). By regressing these points, we estimated $0.6°$ tilt or $\sim 950$ m of surface uplift at the crest (Fig. 14 inset).

## 6.3 Tilt magnitude recorded in the stream network on a broad upland interfluve in the southern Sierra Nevada

We located an unglaciated, low-relief surface on the interfluve between the San Joaquin River and the Kings River in the southern Sierra Nevada (full extent of surface shown in Fig. 12 and close-up shown in Fig. 15a). We broke the river network on this surface into 3 km segments and measured azimuth, $k_{sn}$, and gradient for all segments just as we had done in the analysis of the 2-D simulations of rapid forward tilt (section 5.1). As with the simulated data, we plotted $k_{sn}$ and gradient against azimuth and calculated running medians for both using a $10°$ moving window and observed that gradient values are anomalously high for a $\sim 180°$ range of azimuth ($145 - 325°$; Fig. 15b and c). By fitting a cosine function to the azimuth-gradient data that fell between $145 - 325°$ (blue line in Fig. 15c), we estimated a tilt towards $237°$ (southwest) of magnitude $2.3°$, or $3,500$ m

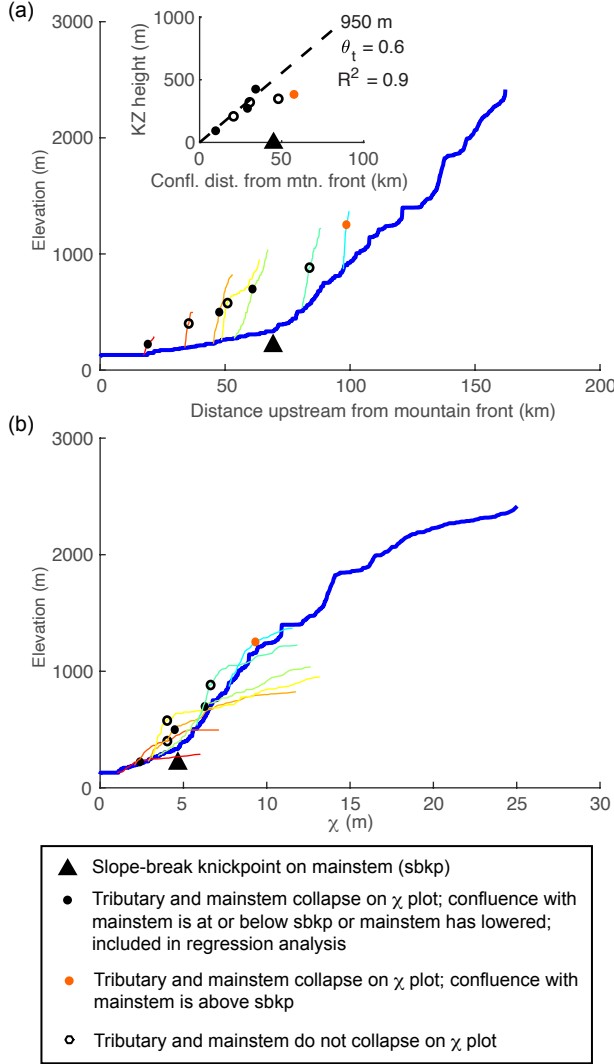

**Figure 14.** Mainstem and tributary profiles for the Middle Fork American-Rubicon River, Sierra Nevada. (**a**) Longitudinal profiles and (**b**) $\chi$ profiles for the mainstem and major tributaries. Mainstem is shown in thicker blue line; tributaries change from warm to cool colors as distance upstream from the mountain front to the tributary confluence increases. Inset plot shows measurements of tributary knickzone drop height against tributary confluence distance from the mountain front along with a linear regression of points with confluences downstream of the mainstem slope-break knickzone whose $\chi$ profiles collapse with the mainstem through the knickzone.

of surface uplift when projected the $\sim 90$ km width of the range in this region. This method yields maximums as gradient continues to rise following the cessation of rapid tilting. Therefore, we interpret the estimate of $2.3°$ as a maximum possible tilt magnitude for the southern Sierra. We also observed multiple possible stream captures in the direction of tilt on this interfluve (e.g., Fig. 16).

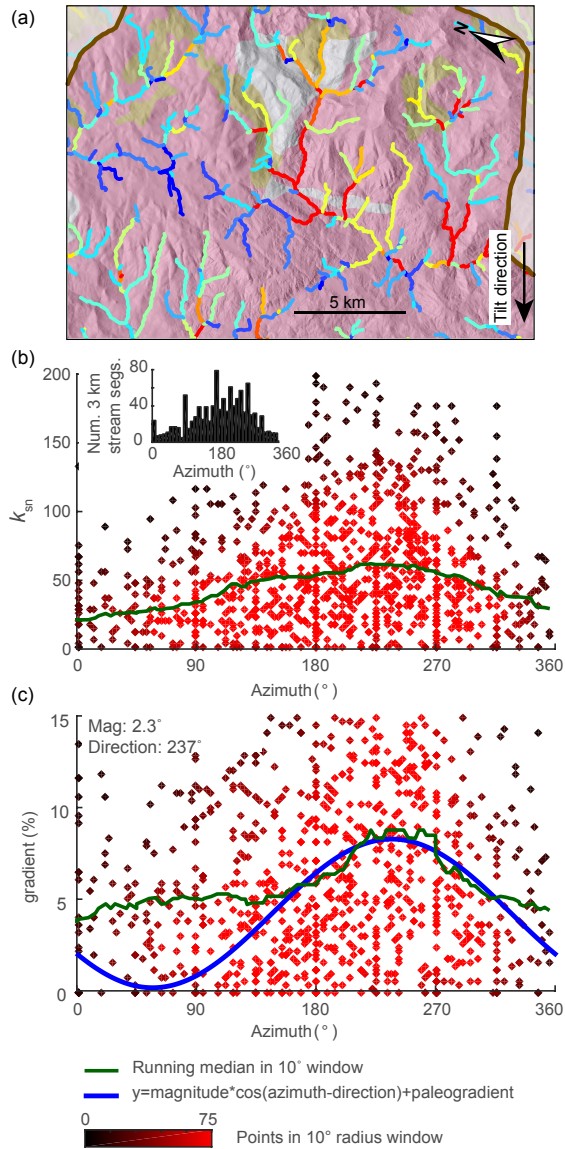

**Figure 15.** Evidence of tilt in relationships between azimuth and gradient and fluvial steepness, $k_{sn}$, of the modern river network on a broad upland interfluve between the San Joaquin and Kings Rivers in the Sierra Nevada. (**a**) Hillshade image rotated such that the tilt direction is towards the bottom of the page and colored by geology (colors same as Fig. 13) for a portion of the interfluve surface overlaid with the stream network colored by $k_{sn}$ measured over 3 km stream segments. Note higher $k_{sn}$ in segments flowing in the tilt direction and lower $k_{sn}$ in segments flowing against the tilt direction. Inset shows a histogram of the azimuth distribution of 3 km stream segments. (**b**) Azimuth-$k_{sn}$ plot. (**c**) Azimuth-gradient plot. Green line in (b) and (c) shows $10°$ wide moving-window median. Blue line in (c) shows the cosine fit which was used to estimate tilt magnitude, the azimuth of tilt direction, and the mean stream gradient pre-tilt ("magnitude", "direction", and "paleogradient" in the legend, respectively).

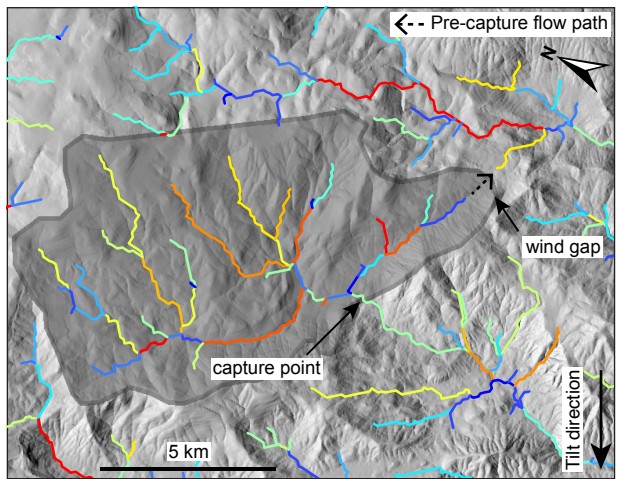

**Figure 16.** Proposed stream capture in the predicted tilt direction in the southern Sierra. Hillshade image with the stream network colored by $k_{sn}$ measured over 3 km stream segments. Proposed captured area is shaded with dark grey and enclosed by a darker grey line. The proposed pre-capture flow path is along the tributary with a noted wind gap, suggesting flow reversal along this short length.

## 7 Example of the geomorphic expression of the transient response to onset of rapid continuous tilting from the Sierra San Pedro Mártir, Baja California, Mexico

We analyzed mainstem and tributary profiles of the Rio San Simón on the western slope of the central Sierra San Pedro Mártir, Baja California, as an example of how the geomorphic signatures of the transient response to the onset of rapid tilting

5 can be recognized in real landscapes and used to estimate tilting rates. The Sierra San Pedro Mártir is a $\sim 100$ km long and $\sim 80$ km wide section of the Peninsular Ranges that contains the highest elevations on the Baja Peninsula owing to the large-displacement steeply-dipping active normal fault system that borders the eastern boundary of the range and is a prominent part of the Main Gulf Escarpment (Rossi et al., 2017; Sedlock, 2003). Rossi et al. (2017) used topographic analysis combined with [10]Be-derived denudation rates in river basins on the eastern side of the range to show that the fault system that borders the

10 eastern boundary of the range initiated during the mid-Miocene and increased in slip rate in the late Pliocene up to a mean of 130 mMyr$^{-1}$. The broad morphology of the range is similar to the Sierra Nevada, with a gentle western slope rising to a steep eastern escarpment (Fig. 17). Similar to the northern Sierra Nevada, lithology is heterogeneous (e.g., Fig. 18), but unlike the northern Sierra Nevada, the Sierra San Pedro Mártir has no evidence of glaciation.

### 7.1 Disequilibrium forms of the Rio San Simón consistent with onset of rapid tilting in the late Cenozoic

15 Profiles of the mainstem and tributaries of the Rio San Simón reveal disequilibrium forms that appear to reflect transient adjustment to a perturbation in boundary conditions (Fig. 18 and 19). In $\chi$ space, the mainstem profile has positive curvature near the outlet that transitions to negative curvature with tributaries plotting below the mainstem. These disequilibrium forms

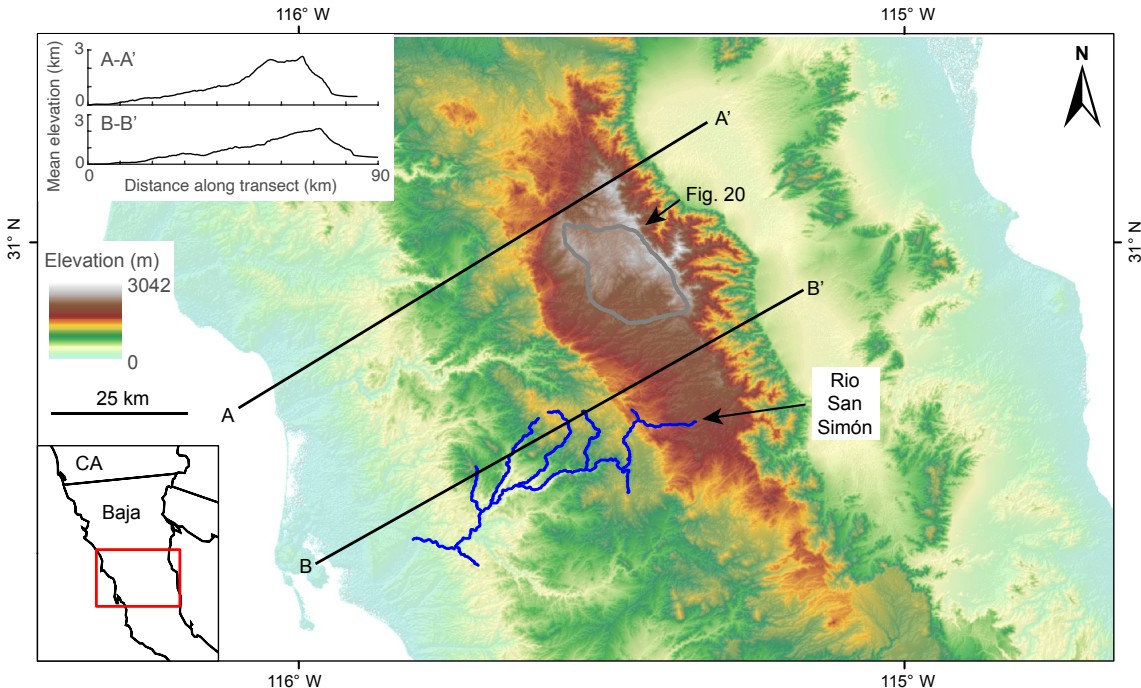

**Figure 17.** Digital elevation model (DEM) of the Sierra San Pedro Mártir in Baja California. The western boundary of the range is defined here by the eastern boundary of mapped Quaternary alluvium (INEGI, 1984). The Rio San Simón and its tributaries are shown in blue and the upland interfluve surface analyzed in Figure 20 is enclosed by a thick grey line. The inset in the bottom left shows the location in North America and the inset in the top left shows elevation swaths for corresponding transects $A - A'$ and $B - B'$.

are consistent with the transient response to onset of rapid tilting observed in 1- and 2-D simulations. Negative-curvature slope-break knickpoints form in mainstem rivers in response to rapid continuous tilting and are difficult to identify in a profile with negative curvature (e.g., Fig. 4 and 9). We identified a negative-curvature slope-break knickpoint at $\chi = 2.5$. However, given the difficulty in identifying knickpoints associated with continuous tilting, we stress that significantly more river profile

5    analysis would be necessary to conduct a robust reconstruction of tectonic history in the western Sierra San Pedro Mártir. We present these profiles as an example of the expression of signatures of rapid tilting in a real landscape that has been previously documented to be actively tilting. Rivers in this region are currently highly alluviated and appear transport-limited today. However, they are deeply incised into steep-walled bedrock valleys and we therefore assume that the alluvial state of modern rivers is a result of more recent changes in climate and does not reflect processes of bedrock incision over million year

10    timescales.

For all analyses, we assume river concavity is uniform throughout the river network and use $\theta_{ref} = 0.45$ as identified by detailed analysis of river networks on the eastern slope of the Sierra San Pedro Mártir (Rossi et al., 2017). We calculated $K$ from $k_{sn}$ values and [10]Be-derived denudation rates presented in Rossi et al. (2017). Excluding two outlier points identified by

Rossi et al. (2017) yields $K = 1 \times 10^{-6}$ ($4.2 \times 10^{-7} - 2.5 \times 10^{-6}$) m$^{0.1}$yr$^{-1}$ with $R^2 = 0.6$ (Fig. S16). Including the outlier points yields a $K$ only a factor of two greater but with $R^2 = 0.35$. As in the simulations and in the Sierra, with $K = 1 \times 10^{-6}$, $\chi$ can be read as time in millions of years and we can interpret the mainstem knickpoint as reflecting an increase in tilt rate $ca.$ 2.5 Ma. Given the positive profile curvature in the mainstem river $\chi$ plot near the outlet, we used 1-D simulations of continuous

tilting to find a best-fit equilibrium profile for the mainstem river downstream of the most prominent negative curvature slope-break knickpoint. We used $K = 1 \times 10^{-6}$ in simulations to find the best fit curve but show analogous results from simulations using the minimum and maximum in $K$ calculated as the 95% confidence interval on the regression that excluded the two outlier points (Fig. S16) as ranges in timing of the onset of tilting and estimated uplift rates. These fits suggest that $ca.$ 2.5 ($6.0 - 1.0$) Ma, tilting rate increased to $\sim 360$ ($150 - 890$) mMyr$^{-1}$ at the crest. Following a step increase in tilting rate,

surface uplift raises the river network to a higher equilibrium steepness and elevation and thus neither tributary knickzone drop height nor mainstem incision will reflect the magnitude of rock uplift. We therefore do not attempt to estimate the magnitude of rock uplift at the crest using these methods as we did in the Sierra where disequilibrium river profile forms appear to reflect a punctuated rapid tilting event in the late Cenozoic rather than continued tilting.

Local deviations in mainstem channel steepness occur upstream of the mainstem knickpoint in the Rio San Simón and may

reflect heterogeneous lithology or prior changes in boundary conditions (Fig. 18). The most prominent deviation is the steep knickzone upstream of the contact between Mesozoic volcanic rock (andesite, shown in brown) and Mesozoic granitic rock (granodiorite, shown in pink). This occurs just upstream of a section with slightly lower channel steepness in the Mesozoic volcanic rock that begins at $\sim 53$ km upstream. An additional low-steepness section occurs just upstream of the knickpoint at $\sim 23$ km upstream and downstream of the contact between the Mesozoic volcanic rock and the Pre-Cenozoic metamorphic

rock (gneiss, shown in grey). These deviations are similar to those generated in the 1-D simulation of a step increase in tilt rate with a band of more erodible rock (compare Fig. S9 with Fig. 18), though we did not attempt to estimate tilt magnitude from these forms as we did in the Sierra Nevada because the low-gradient reaches do not exactly correspond to mapped changes in lithology in the Rio San Simón at the map scale available.

## 7.2   Rio San Simón tributary knickzones reflect onset of rapid tilting in the late Cenozoic

We analyzed all major tributaries to the Rio San Simón. The majority of tributaries have knickzones with nonuniform steepness that collapse in part with the mainstem on the $\chi$ plot. Upstream, tributaries have negative curvature that transitions to relatively uniform steepness in tributary headwaters. These tributary forms are consistent with the transient response to onset of rapid tilting (compare Fig. 19 with Fig. 4c and d). In 1-D simulations of the transient response to onset of rapid tilting, the timing of the onset of rapid tilting can be recovered by subtracting the $\tau$ value of the mainstem-tributary junction from the $\tau$ value of

the upstream end of the corresponding tributary knickzone where channel steepness transitions from nonuniform to uniform. We identified these locations in tributaries to the Rio San Simón and used these to estimate that onset of rapid tilting occurred $ca.$ 6.3 (8.5-1.6) Ma. This estimate likely reflects initial onset of tilting, rather than the increase in slip rate that the mainstem knickpoint reflects, as it is more difficult to identify tributary knickzones associated with an increase in tilt rate than onset of

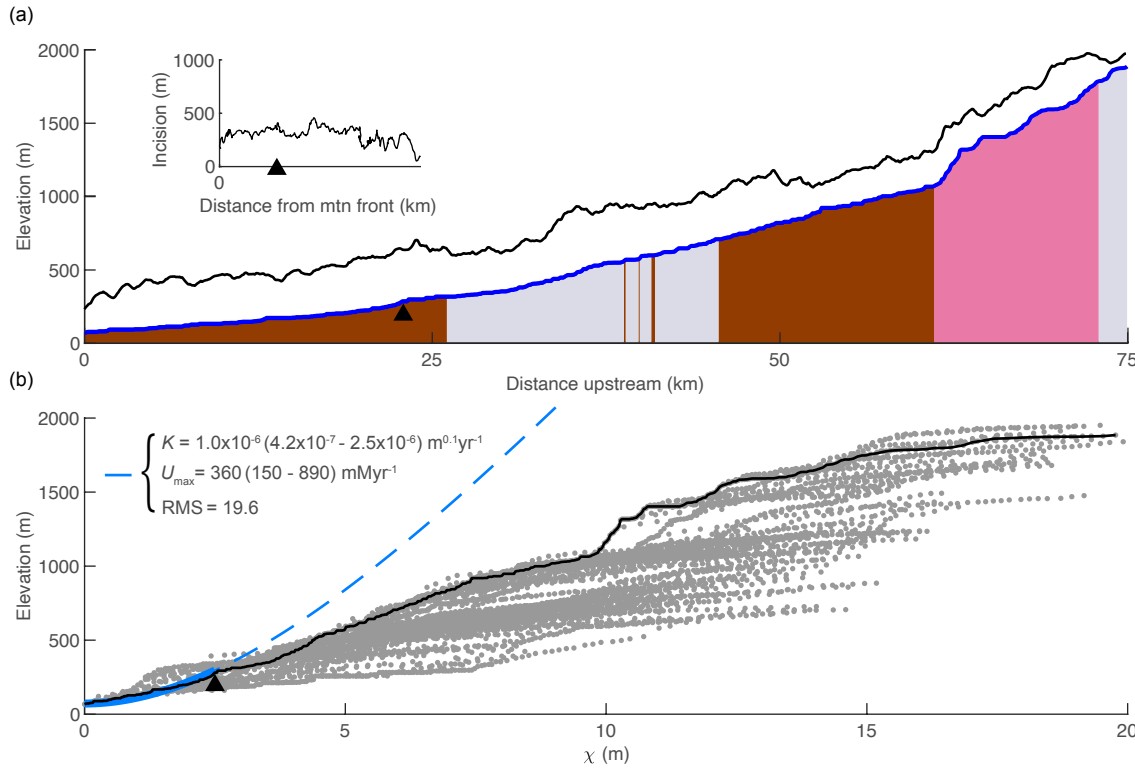

**Figure 18.** Mainstem river profile for the Rio San Simón in the Sierra San Pedro Mártir, Baja Californnia. (**a**) Longitudinal profile (blue line) for mainstem, with surface geology (INEGI, 1984), and topography of proximal upland surface (black line). See Fig. 13 for geology legend. Inset plot in (a) shows valley incision into basement calculated as the difference between the mean elevation of the proximal upland surface and the modern river profile. Black triangles mark the mainstem slope-break knickpoint. (**b**) $\chi$ plot with mainstem $\chi$ profile (black) over data from entire basin (grey). Thick blue line shows the best-fit curve to $\chi$-elevation data downstream of the mainstem knickpoint at $\chi = 2.5$ that was then projected to the channel head (dashed blue line).

tilting. The variability in this estimate likely reflects the challenge in identifying the upstream end of knickzones but the range supports onset of rapid tilting in the late Cenozoic.

### 7.3   Evidence of tilting recorded in the stream network on a broad upland interfluve in the Sierra San Pedro Mártir

We located a low-relief surface to the west of Cerro Picacho del Diablo where stream segments flowing down dip appear
5    steeper than stream segments flowing in other directions. We broke the river network on this surface into 3 km segments and measured azimuth, $k_{sn}$, and gradient for all segments as we did for simulated networks and river networks in the Sierra Nevada, plotted $k_{sn}$ and gradient against azimuth, and calculated running medians for both using a $10°$ moving window. We fit a cosine function to values of gradient that appear anomalously high (azimuth $= 105 - 285°$), yielding a tilt direction of $203°$ and a tilt magnitude of $0.3°$ (blue line in Fig. 20c). In the 2-D simulation of continuous rapid tilting presented above, this method

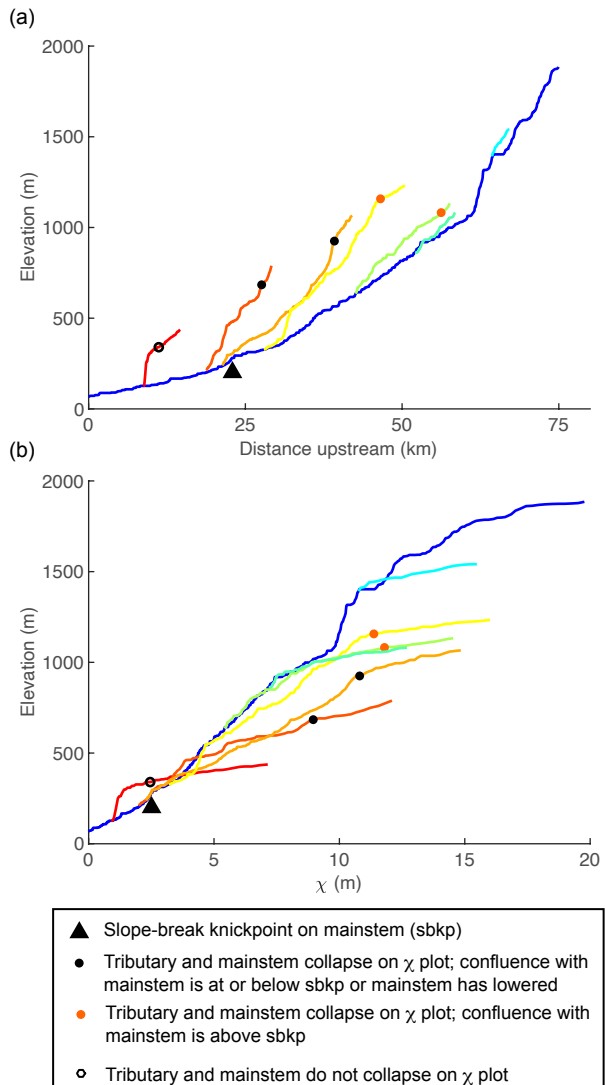

**Figure 19.** Mainstem and tributary profiles for the Rio San Simón in the Sierra San Pedro Maártir, Baja California. (**a**) Longitudinal profiles and (**b**) $\chi$ profiles for the mainstem and major tributaries. Mainstem is shown in thicker blue line; tributaries change from warm to cool colors as distance upstream from the mountain front to the tributary confluence increases.

reflects the true magnitude of tilt until $t = 3$ Myr, but at later time steps no signatures are apparent in either azimuth-gradient or azimuth-$k_{sn}$ relationships. This suggests that the observed signatures on the upland surface near Cerro Picacho del Diablo reflects a recent increase in tilt rate in the Sierra San Pedro Mártir.

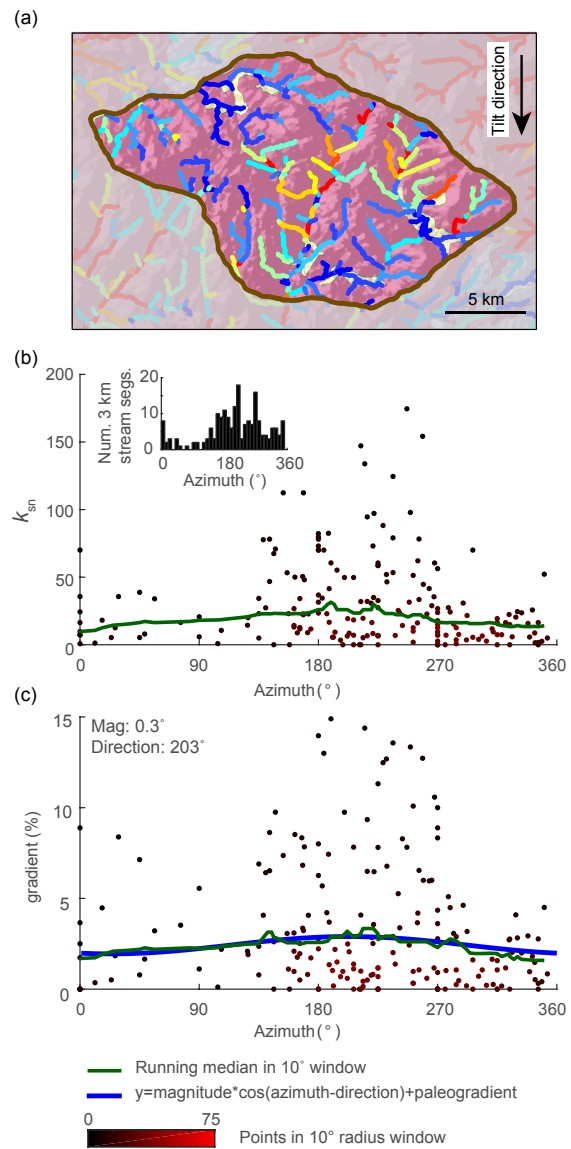

**Figure 20.** Evidence of tilt in relationships between azimuth and gradient and fluvial steepness, $k_{sn}$, of the modern river network on a broad upland surface to the west of Cerro Picacho del Diablo in the Sierra San Pedro Mártir. (**a**) Hillshade image colored by geology (colors same as Fig. 13) for a portion of the interfluve surface overlaid with the stream network colored by $k_{sn}$ measured over 3 km stream segments. Note higher $k_{sn}$ in segments flowing in the tilt direction and lower $k_{sn}$ in segments flowing against the tilt direction. Inset shows a histogram of the azimuth distribution of 3 km stream segments. (**b**) Azimuth-$k_{sn}$ plot. (**c**) Azimuth-gradient plot. Green line in (b) and (c) shows $10°$ wide moving-window median. Blue line in (c) shows the cosine fit which was used to estimate tilt magnitude, the azimuth of tilt direction, and the mean stream gradient pre-tilt ("magnitude", "direction", and "paleogradient" in the legend, respectively). Location of surface shown in Fig. 17.

# 8    Discussion

A long-standing goal of geomorphology has been to develop tools to extract Earth history from landscape form, but questions remain regarding the degree to which particular perturbations to Earth's surface, whether tectonic or climatic, are recorded in a recognizable manner. In the analysis presented above, we show that the fluvial response to the simple case of a punctuated rigid-block tilting event does inscribe a suite of identifiable geomorphic forms throughout the river network during the transient evolution back towards an equilibrium state that collectively comprise a robust signature of tilt. These signatures are composed of unique patterns of channel steepness, $k_s$, or channel gradient normalized for drainage area, in the mainstem as compared to the tributaries and a unique distribution of knickpoints originating at the mainstem outlet and at mainstem-tributary junctions. Tilting about a horizontal axis uniformly increases or decreases the gradient of stream segments perpendicular to the tilt axis, while the gradient of stream segments parallel to the tilt axis remain unchanged. Owing to the branched nature of river networks, the mainstem river and its tributaries commonly experience very different uplift fields, which, to first-order, is the root cause behind the tilt signatures described above. Further, the power law relationship between gradient and drainage area that is characteristic of a dendritic river network translates a uniform change in stream gradient to a nonuniform change in $k_s$. Analysis of the transient response to tilting clarifies that a uniform increase in channel gradient on the mainstem does indeed generate knickpoints.

Similar to other well-explored perturbations (e.g., Baldwin et al., 2003; Tucker and Whipple, 2002; Whipple and Tucker, 1999), a punctuated tilting event will generate a knickpoint in the mainstem that will propagate upstream from the outlet, the location of which will track the timing of the end of the perturbation. In all perturbations explored herein, the mainstem knickpoint separates a downstream section with new, equilibrium $k_{sn}$ from an upstream section with disequilibrium $k_{sn}$, as long as both $U$ and $K$ are steady and uniform following the perturbation. Following a uniform pulse of uplift or a step change in fluvial relief, the landscape upstream of the mainstem knickpoint retains the pre-perturbation uniform equilibrium $k_{sn}$ and is unaffected by the change in boundary conditions until the knickpoint propagates upstream (e.g., Baldwin et al., 2003; Tucker and Whipple, 2002; Whipple and Tucker, 1999).

Unlike a uniform pulse of uplift or a step change in fluvial relief, both rigid-block tilting about a horizontal axis perpendicular to the mainstem and truncation of the mainstem perturb $k_{sn}$ along the entire mainstem away from an equilibrium state and thus immediately initiate a response along the entire mainstem, but the perturbation to $k_{sn}$ is initially localized to the mainstem. Following a punctuated forward tilting event, the mainstem is oversteepened and begins to incise, whereas, following punctuated back tilting or beheading, the mainstem is everywhere less steep than equilibrium and begins to rise. Immediate, yet transient, mainstem response generates unique patterns in tributary channel steepness that reflect, in the forward-tilting case, a period of transient base level fall that generates a knickzone and, in the back-tilting or beheading case, a period of base level rise that generates a low-gradient reach. In all cases following cessation of rapid tilting, tributary knickzone $k_{sn}$ matches that of the mainstem upstream of the tributary junction and hence tributaries collapse with the mainstem on $\chi$ plots downstream of sections that retain equilibrium $k_{sn}$. Similar to how the $\tau$ value of a mainstem knickpoint tracks the timing of a perturbation (e.g., Whipple and Tucker, 1999), the $\tau$ value of the upstream end of tributary knickzones can also be used to track timing

because $k_{sn}$ begins to change upon perturbation. In the case of a rapid pulse of forward-tilting, tributary knickzones occurring in tributaries with junctions downstream of the mainstem knickpoint also record the magnitude of tilt induced rock uplift. As the duration of tilt increases, signatures of tilt grow less pronounced but estimates of the timing of cessation of tilting remain accurate. After a step change to ongoing and more rapid tilting, similar disequilibrium forms are generated but the magnitude

of tilt is not reflected in these forms as both channel steepness and elevation are rising to a new equilibrium. Additionally, the timing of onset of rapid tilting is more difficult to estimate in that it is reflected in negative-curvature knickpoints in mainstem and tributary profiles that also have negative curvature formed by the tilt. Each perturbation explored herein generates a distinct incision pattern as the mainstem river incises below paleotopography markers that record the magnitude of rock uplift in the perturbation plus rock uplift accumulated by the background rate. Forward tilting with heterogeneous lithology leaves partic-

ularly distinct incision patterns as knickpoints can form anywhere in forward-tilted river segments where more erodible rock occurs upstream of less erodible rock. This mechanism suggests that incision histories in tilted landscapes can vary widely both among neighboring basins and within an individual basin.

In addition to generating a robust suite of signatures in river profiles, the transient response to tilting leaves additional distinct signatures in river networks. In 2-D simulations of rapid forward tilting towards river outlets, stream segments flowing in the

direction of tilt have increased $k_{sn}$ and gradient, with the magnitude of tilt recoverable from the relationship between gradient and azimuth in over-steepened reaches, but only shortly after the tilting event. Following continuous rapid tilting, the magnitude of tilt is reflected in the river network only briefly (at $t = 3$ Myr) and only with a higher-magnitude tilting rate ($1°$ vs. $0.5°$), however, this response time is dependent on the chosen parameter values in the model. Following punctuated rapid forward tilting, gradient and $k_{sn}$ continue to rise and both remain elevated at all azimuths for millions of years, generally resulting in

an overestimate of tilt magnitude. A rapid tilting event can also generate topologic changes oriented in the direction of tilt that can occur coincident with tilting or anytime following tilting while the river network equilibrates. These signatures are unique to nonuniform uplift in that they result from simultaneous nonuniform changes to channel steepness, which in turn generate erosion gradients that allow drainage divides to become mobile throughout the river network.

Through river profile and network analysis in isolated localities in the Sierra Nevada, California, and Sierra San Pedro

Mártir, Baja California, we showed that proposals for late Cenozoic punctuated tilting in the Sierra Nevada and onset of rapid tilting in the Sierra San Pedro Mártir are consistent with the geomorphic signatures we quantify for a forward tilted landscape. Although our estimates of tilt timing and magnitude made by applying these signatures to both regions are consistent with published estimates, variability exists among the suite of independent measures of both timing and magnitude for each region. We present these analyses as examples to show that the geomorphic signatures of the transient response to tilting as outlined

herein exist in nature but not as a demonstration of a robust method for characterizing the uplift history of a landscape. For a robust characterization of a the uplift history of a landscape, river networks across the entire landscape should be analyzed, not isolated profiles as presented here. We present a thorough and systematic analysis of Sierra Nevada river profile and network structure in a forthcoming paper.

To quantitatively investigate the signatures of the transient fluvial response to tilting, we used the detachment-limited stream

power model (Howard, 1994; Whipple and Tucker, 1999), which has been shown to produce first-order features of equilibrium

and transient bedrock river profile forms as seen in nature (Howard and Kerby, 1983; Siedl and Dietrich, 1992, Tucker and Whipple, 2002, Whipple and Tucker, 1999). We demonstrate that the signatures of tilt are robust and unique by additionally simulating the transient fluvial response to a uniform pulse of uplift and a step change in fluvial relief, both of which have been studied extensively (e.g., Baldwin et al., 2003; Bonnet and Crave, 2003; Grimaud et al., 2016; Howard, 1994; Rosenbloom and Anderson, 1994; Royden and Perron, 2013; Tucker and Whipple, 2002; Whipple and Tucker, 1999), as well as the transient fluvial response to beheading of the mainstem river, the signatures of which have not been thoroughly explored previously. Although we exclusively use the detachment-limited stream power model for the sake of comparing first-order profile forms in the simplest case, the transient response to some perturbations may be more accurately simulated using a transport-limited model or a model that explicitly incorporates the competing effects of tools and cover (e.g., Sklar and Dietrich 2004). For example, Whipple and Tucker (2002) showed that the transient response to a decrease in uplift rate is eventually accompanied by a transition from detachment-limited to transport-limited conditions. Given the similar declining-state profile form, the transient response to a punctuated forward tilting event may also eventually involve a shift to transport-limited conditions. However, while this may change channel response times and profile forms at later time steps, it is unlikely to significantly change the distinct signatures that characterize first-order profile forms at early time steps. Similarly, perturbations that move mainstem and/or tributary $k_{sn}$ below equilibrium $k_{sn}$, such as mainstem beheading and back tilting, will likely generate an aggradational response. Future research could exploit the first order signatures of tilting or mainstem beheading as outlined herein to investigate the depositional response and possible transitions to transport-limited conditions.

Heterogeneity in real landscapes presents challenges to extracting uplift history from landscape form. We demonstrated that deviations in the expected river profile signatures of the transient fluvial response to rigid-block forward tilting can result from nonuniform uplift that deviates from perfect rigid-block tilting, heterogeneous lithology, changes in the stream power slope exponent, disequilibrium initial conditions, or coincident perturbations. Deviation from perfect rigid-block behavior creates similar signatures in mainstem and tributary profile forms that still record tilt timing and magnitude, though signatures are more subdued owing to the fact that the majority of uplift occurs at the channel head. Heterogeneous lithology, in contrast, can result in knickzones forming anywhere in the river profile where more erodible rock occurs upstream of less erodible rock during the transient response to tilting, with the potential to significantly alter expected signatures. Furthermore, we did not attempt to simulate the transient response to tilting in sub-horizontally layered stratigraphy, the dynamics of which are likely to be considerably more complex (e.g., Forte et al., 2016).

Certain systematic changes in process or mistakes in data analysis can also create $\chi$ plots that could be mis-interpreted as reflecting a signature of tilt. A shift in $m$ and/or $n$ owing to a change in erosional mechanism (Whipple et al., 2000) or a transition from detachment-limited to transport-limited between tributaries and mainstem rivers or along mainstem rivers would violate our assumption of uniform concavity and could generate negative curvature and/or tributaries with lower steepness than mainstems that would simply reflect a change in process rather than tectonic forcing. Similarly, extending $\chi$ analysis above the channel heads results in $\chi$ profiles with negative curvature that would also only reflect a change in process (e.g., Clubb et al., 2014). Choosing too high of a reference concavity can result in negative curvature in $\chi$ plots as well and could be confused with tilting. However, the patterns in channel steepness that characterize the transient fluvial response to rapid punctuated

tilting remain when $\chi$ is calculated with incorrect reference concavities (Fig. S17). Tilting generates a suite of signatures that collectively are robust and unique, but individual signatures share characteristics with those produced by other perturbations described herein. Furthermore, constraining initial conditions or teasing apart multiple perturbations presents a major challenge. Thus, it could be difficult to recognize signatures of tilt in river profiles and network patterns if tilting is not the predominant

perturbation recorded in landscape form or if erosional mechanisms are nonuniform. This highlights that care must be taken when interpreting tectonic histories from landscape form.

Estimates of timing made using methods outlined herein may also be sensitive to heterogeneity in rock properties, processes, and the dynamics of real landscapes, even in landscapes where signatures of tilt are strongly expressed. Estimating timing requires constraining $K$, $m$, and $n$, with standard methods utilizing equilibrium rivers or at least rivers that are colinear on $\chi$

plots - a difficult requirement in transient landscapes. Even if these can be constrained, any perturbation to knickpoint travel times will affect estimates of tilt timing. For example, nonuniformity in $K$ can result in either faster or slower knickpoint travel times and result in either overestimates or underestimates of tilt timing, respectively. Furthermore, calculations of travel times are generally based on the modern river network, but major topologic change in river networks (e.g., Willett et al., 2018) following tilting has the potential to impact estimates of timing. Drainage area gain that occurs after knickpoints have

propagated for some time would result in underestimation of time-since-tilt as area gain moves profiles to the left of equilibrium in $\chi$ plots (Willett et al., 2014), thus resulting in lower $\tau$ values.

We focused on moderate magnitude tilting ($0.5 - 1°$) at 100 km scales over which fluvial networks are well developed. Signatures of the fluvial response to high magnitude tilting over short length scales as is observed in the Basin and Range (e.g., Stewart, 1980) may be quite different as the tilt perturbation is large relative to the majority of channel gradients which

could effectively reset river network structure, river networks are much smaller, and hillslope processes may dominate over fluvial over a large part of the network. Similarly, fluvial signatures of very low magnitude tilting over large wavelengths as can occur through dynamic topography (e.g., Liu and Gurnis, 2010) may differ substantially from those presented herein as near-rigid-block behavior is less likely over long length scales (e.g., Martel et al., 2014) as are the required assumptions of uniform erosional processes and uniform concavity. Processes not explored here, such as sediment flux dependent erosion (Lague, 2014;

Sklar and Dietrich, 1998; 2004), thresholds in shear stress or stream power in controlling bedrock incision (e.g., DiBiase and Whipple, 2011; Lague et al., 2005; Snyder et al., 2003a; Snyder et al., 2003b; Tucker, 2005), or changes in channel width with uplift rate (Attal et al., 2011; Finnegan et al., 2005; Lague, 2014; Turowski et al., 2009; Whittaker et al., 2007; Wobus et al., 2006; Yanites and Tucker, 2010; Yanites et al., 2010), may also have the potential to confound signatures of tilt or impact estimates of timing or magnitude.

We showed that river networks in the Sierra Nevada and the Sierra San Pedro Mártir have profiles that are consistent with the suite of forms that comprise the robust geomorphic signatures of a punctuated tilting event and a step increase in tilt rate, respectively, that were developed using a 1-D river profile model with the simplest version of detachment-limited stream power and rigid-block tilting. We also measure an azimuth dependence in river channel gradient in both landscapes that is consistent with analogous measurements from simulations of punctuated and ongoing tilting in a 2-D landscape evolution

model. The estimates for timing and magnitude made using these signatures for both landscapes are in line with independent

estimates, suggesting that the signatures of tilt presented in this paper may be used alongside a thorough analysis of regional river networks to reconstruct robust tectonic histories from landscape form, despite the potential limitations outlined above. However, as previously mentioned, there is an ongoing and active debate surrounding the tectonic history of the Sierra Nevada and most recently the open discussion version of this paper served as a platform for this debate (Beeson and McCoy, 2019).

Within this forum and in a subsequent solicited review, Gabet argues that the knickpoint we identify as a migrating knickpoint of tectonic origin in the Middle Fork American is instead a stationary knickpoint that reflects lithologic control (Gabet, 2019a). We agree that lithology has a significant impact on river profiles in the northern Sierra, but we find that the observed patterns in incision into basement rock below Cenozoic deposits and the relationship between longitudinal profile form and lithologic boundaries in the Sierra are more consistent with the models presented herein in which the transient response to a punctuated

tilt is modulated by heterogeneous lithology (Beeson and McCoy, 2019), rather than reflecting a response to heterogeneous lithology with no tectonic forcing (Gabet, 2019b).

## 9 Conclusions

We investigated the geomorphic signatures of the transient fluvial response to tilting and whether these signatures can be used to quantify uplift histories in terms of timing and magnitude. With a model river network composed of linked 1-D river

longitudinal profile evolution models, as well as a full 2-D landscape evolution model, we demonstrated that the transient response to a punctuated tilting event leaves a suite of signatures throughout the river network that are distinct and, in the case of forward tilting toward the river outlet, provide multiple independent measures of tilt timing and magnitude. Further, we showed that the signatures of rapid punctuated forward tilting are consistent with profile forms and river network patterns observed in the Sierra Nevada, California, USA, a landscape proposed by many to have experienced a punctuated tilting event

in the late Cenozoic, and that signatures of recent onset of rapid tilting are consistent with profile forms and network patterns observed in the Sierra San Pedro Mártir, Baja California, Mexico, a landscape proposed to be experiencing ongoing rapid tilting that initiated in the mid-Miocene. However, we presented only a small sample of the river network draining the western slopes of these ranges. Although the resulting estimates of tilt timing and magnitude are within the range of previous estimates for both regions, the variability across our estimates indicate that a thorough analysis across all the river networks in both

landscapes is necessary to robustly characterize the uplift history using these methods.

Signatures of tilt are robust to the extent that they are not obscured by other later-stage perturbations of greater magnitude. Heterogeneous lithology and deviations in the stream power exponent away from $n = 1$ cause predictable deviations in the transient response to tilting, but generally do not completely obscure tilt signatures. In contrast, disequilibrium initial conditions and coincident perturbations may result in myriad permutations to predicted profile forms that may obscure signatures of tilt.

To test the uniqueness of tilt signatures presented, we simulated a range of perturbations that generate transient river profiles that share characteristics with profiles generated by tilting, including a pulse of uniform uplift, step changes in fluvial relief, and mainstem beheading. Thus, although this paper focuses on tilting, it provides a detailed summary of the suite of signatures that characterize the transient fluvial response to each of these perturbations in longitudinal profiles, $\chi$ plots, and patterns in

mainstem incision below paleotopographic markers that could help to interpret more diverse landscape histories recorded in river network forms.

*Video supplement.* Videos showing response of model river network composed of linked 1-D river longitudinal profile evolution models to all perturbations discussed within are available as an online resource (see Beeson, 2019)

5 *Author contributions.* H.W.B and S.W.M designed the experiments and H.W.B developed the model code and performed the simulations. H.W.B conducted analysis on rivers in the Sierra Nevada and the Sierra San Pedro Mártir. Both authors contributed to the writing of the manuscript.

*Competing interests.* The authors declare that they have no conflict of interest.

*Acknowledgements.* We thank three anonymous referees and Dr. Emmanuel Gabet for providing detailed, constructive feedback. We ac-
10 knowledge the Associate Editor Dr. Jens Turowski for handling this paper. Lastly, we thank Taylor Perron for providing a 2-D landscape evolution model that we modified for use in this paper.

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
