# Peer review of "Geomorphic signatures of the transient fluvial response to tilting"

_Earth Surface Dynamics, 2019_

## Short Comment (SC1) · 4 Jun 2019

The manuscript by Beeson and McCoy explores the role of tilting in affecting bedrock channel profiles, a worthwhile topic, via numerical modeling and GIS analysis. Unfortunately, the main conclusions concerning the geologic history of the Sierra Nevada are refuted by previously published field evidence. The present study concludes that tilting of the northern Sierra Nevada drove incision of the American River system beginning 5 mya (or 1.9 mya); however, the presence of older sediments deep within the canyons of the American River (and other major Sierran rivers) unambiguously falsifies this conclusion (Gabet, 2014): a canyon cannot be younger than the sediments found within it. This study also concludes that the southern Sierra Nevada experienced 2.3° of tilt over the past few million years, a result contradicted by the slope of a lava flow that

proves that there could not have been more than ~1.4° of tilt since the mid-Miocene (Huber, 1981). I describe these problems in more detail below, along with other issues; for example, the authors appear to have been confused about which river they were analyzing. I also provide suggestions for how the numerical analyses and the modeling might be improved. (I apologize if this 'short comment' seems long-winded, I've just written 2 papers on this subject and these issues are fresh in my mind.)

NORTHERN SIERRA NEVADA Although the authors refer to the Middle Fork of the American River throughout, the long profile shown in Figure 10a appears to be for the Rubicon River, which is a tributary to the Middle Fork, so I wasn't sure which river they actually analyzed. My comments below apply equally well in either case.

The approach used here is based on presumed incision depths below a series of late Cenozoic volcanic rocks found along the interfluves. This analysis is entirely dependent on the assumption that the present elevation of these volcanic rocks are faithful recorders of the pre-uplift topography. However, this assumption was falsified in Gabet (2014). In that paper, I provide examples of volcanic rocks and Eocene-Oligocene gravels that reach deep down into the canyon of the South Fork American River (note, the age of the volcanic rocks in Fig. 15b in that paper is late Miocene, not 3 my). In addition to those examples, 24 myo volcanic rocks can be found ~200 above the bed of the South Fork at an elevation where the ms claims that there has been ~500 m of post-5 my (or 1.9 my) incision on the Middle Fork, and 6.5 myo rocks can be found just 70 m above the bed of the South Fork at an elevation where the authors are claiming ~300 m of incision on the Middle Fork (Gabet and Miggins, in review). Although these examples come from the South Fork, it is part of the same drainage system as the Middle Fork and the Rubicon and subject to the same forcings, and, therefore, should have a similar history of incision. On the Middle Fork, at a bed elevation of 1050 m, Lindgren (1911) found Eocene-Oligocene gravels at 1236 m, thereby constraining the net bedrock incision over the past ~40-50 my to only 186 m (Gabet and Miggins, in review); the present ms, in contrast, suggests that there has been ~400 m of incision

at this elevation on the Rubicon (?) just in the late Cenozoic (Figure 10). I present other examples in Gabet and Miggins (in review), and they all demonstrate that there have been minimal amounts of basement incision in the late Cenozoic. To make this point more succinctly, unequivocal field evidence shows that the valleys of the Middle Fork and the South Fork of the American had already been incised down to ∼70% of their modern depths by the early Oligocene. Therefore the assumption underpinning the present manuscript's incision analysis is not valid – the position of the late Cenozoic volcanic rocks do not faithfully record the region's paleotopography. Moreover, the field evidence demonstrates that the beds of the Middle Fork and South Fork rivers were already near their present elevations by the early Oligocene, a conclusion incompatible with the results presented here (see also House et al, 1998). This conclusion applies equally well to the Rubicon because it's a tributary of the South Fork. The fact that uplift-related incision could not have occurred after deposition of the volcanic rocks has been known since the earliest days of California geology (a key point not appreciated in the Wakabayashi papers). For example, in 1880, after an exhaustive survey of the Tertiary gravels, Whitney concluded, with respect to uplift, that "There is also abundant evidence the volcanic epoch was not inaugurated in the Sierra until the range had approximately its present form." Also, since these canyons have been deep for a long time, knickpoints along the profiles of their tributaries cannot be used in the manner adopted here to provide evidence of recent uplift-driven incision (i.e. Section 6.2). (Note to the AE, I can send you a figure from my 'in review' paper showing evidence for the antiquity of these canyons).

I spent some time trying to understand why the results from the numerical analysis diverge from the actual field evidence to see how it might be improved. One issue is that the streampower framework begins with the assumption that there is uplift to be detected (egg, U in Eqn 5 cannot be zero). In other words, this technique cannot be used to determine whether or not there was uplift because it already assumes that U is not zero. Note that, although several years of measurements have found evidence for present uplift (Hammond et al., 2012), this is likely due to groundwater depletion (Amos

et al., 2014); moreover, there would be little justification for applying several years of modern data to the past several million years.

Another critical issue with the streampower analysis is the assumption that the erodibility along the river is uniform (Section 6.1). Rivers in the American River watershed flow across a wide variety of rock types with a range of erodibilities. In Gabet (2019), I demonstrate that lithology is the primary control on the channel steepness index; therefore, any attempt to extract tectonic information from channel profiles must first account for bedrock erodibility. I calculated the normalized channel steepness for 36 different rock units in the northern Sierra; I also determined an average Ksn for the four main lithological categories. If the authors (or the AE) would like, I would be happy to provide a pre-print of this paper – it is an unfortunate bit of timing that it was not published sooner. The authors could use my results to account for differences in erodibility among the different rock types and then re-run their analyses. On a related note, the authors should be aware that the GIS lithology layer (Ludington et al, 2005, 2007) is not accurate (eg, contacts are sometimes not where they are supposed to be) and can't be trusted for detailed analyses.

With respect to how the model was parameterized, the river incision rate used (0.06 mm/yr; Section 6.1) did not appear to be well-justified. The study used a value from Stock et al (2004), which is an incision rate through limestone from a site ∼100 km away. Aside from the issue of distance between the two sites, the differences in lithology between the two is critical: the American River system cuts through very resistant rocks like granite, metavolcanics, and quartzite (note, the 1:250000 map shows sandstone but Hietanen (1973) clearly describes this as quartzite). As shown in Sklar et al (2001), these rocks are 1-2 orders of magnitude more resistant to fluvial abrasion than limestone (see also Hack, 1957). A more appropriate estimate for an incision rate can be determined with data from Gabet and Miggins (in review) in which we dated a volcanic deposit that is just 70 m above of the South Fork of the American River. At this site, the river is cutting through granitic rock, which is similar in erodibility to

quartzite and metavolcanics (Gabet, 2019; Sklar and Dietrich, 2001). The age of the deposit is 6.5 Ma, yielding an incision rate of 0.01 mm/y over the past several million years; unfortunately, the data for this calculation are in a paper that is under review and, therefore, can't be cited until it is published but it highlights the issue with the rate used in the present ms. This incision rate also comes with an important caveat: in the latter half of the Cenozoic, the Sierras were buried by volcanic rocks from 2 major eruptive episodes, the first beginning about 28 mya and the second ending about 5 mya. Therefore, any incision rate based on the elevation of a volcanic deposit only constrains the maximum rate. For the example I just gave above, it is possible that portions of flow descended all the down to the river's present elevation but were eroded away, meaning that a bedrock incision rate of zero over the last several million years is fully consistent with the field evidence.

SOUTHERN SIERRA NEVADA For the southern Sierras, this ms analyzes the steepness index of the modern rivers according to their azimuth and concludes that there has been 2.3° of tilting along an azimuth of 237°. This conclusion is based on a resemblance between a DEM analysis of terrain south of the San Joaquin River and the results of a landscape evolution model in which instantaneous tilt was imposed as a boundary condition. However, the 2.3° of predicted tilt in the late Cenozoic in the southern Sierra is refuted by the field evidence. Along the San Joaquin River (in close proximity to where the authors made their slope measurements), a 10 myo trachyandesite flow has created a set of table mountains at the southern edge of Millerton Lake (Huber, 1981); from the lower table mountain to Squaw Leap, the upper, uneroded surface of this flow has a slope of 1.37° at an azimuth of 226°. Therefore, even if the upper surface of the lava flow had a slope of 0° when it cooled, it limits the total amount of possible tilt since 10 mya to only 1.39° (at an azimuth of 237°) because any more tilting would mean that the lava had flowed out of the Central Valley and into the Sierras, which would be nonsensical since the source of the flow was in the mountains. Furthermore, the mid-Miocene fluvial deposits underneath the lava flow can also be used as a tilt-meter (albeit with a bit more uncertainty). A paleo-slope analysis based

on the size of the gravels shows that they were deposited on a slope of ∼0.15°; their present slope is now at 1.26°, meaning that the maximum amount of tilt experienced by the region since the mid-Miocene cannot exceed ∼1.1° (Gabet, 2014). Therefore, two sets of field evidence refute the conclusion that there has been 2.3° of tilt at any time since the mid-Miocene in the southern Sierras. Similarly, Mix et al (2019) finds no evidence for significant late Cenozoic uplift in the southern Sierras.

MODELING I was concerned by the tectonic boundary conditions imposed in the model, specifically the instantaneous tilt of 1000 m, the instantaneous baselevel drop of 1000 m, and the instantaneous 1000-m tilt with uniform background uplift. It is obvious, of course, that none of these scenarios are realistic (i.e. mountain ranges don't instantaneously increase their elevations by 1000m); but that's not necessarily a problem. I understand the value in whacking a model and seeing what happens. The problem arises, however, when the results from these types of modeling scenarios are used to interpret real landscapes. For example, results from a model run with instantaneous tilt (Fig. 7c) are compared to a DEM analysis of terrain near the San Joaquin River (Fig. 12c), and a similarity between the two are used as evidence for tilting of the southern Sierra. Because the key element (tilting) differs in a fundamental way between the two (instantaneous in the model vs gradual in the real world), a comparison between the two seems like an apples-and-oranges situation. This may explain why the manuscript's estimate of tilt magnitude in the region is contradicted by the field evidence. Nevertheless, I think this technique could have value (in fact, I once considered trying something similar, albeit not as sophisticated) if the modeling adopted more realistic boundary conditions. I provided, in an earlier comment, some calculations that could be used to place constraints on a tilt rate over the past ∼10 my.

The rigid-block assumption over a distance of 180-km is also problematic because real crust will bend, which means that tilting is not distributed uniformly across the landscape (Martel et al., 2014). To demonstrate this, I did some calculations using the standard elastic half-space beam flexural model with the 'broken plate' assumption

(appropriate because of the range-front faults along the eastern Sierras), regional flexural rigidity parameters (Martel et al., 2014), and the assumption that tilting is due to loading at the lower end of the block (i.e., the eastern edge of the Central Valley, at x=0 km). The results indicate that 2° of tilt at x=0 km decays rapidly with distance such that at x=100 km the block is only tilted by 0.7°, and the end of block (i.e., x=180 km) experiences essentially no tilt. Of course, this point loading scenario is a simplification and represents an end-member scenario; I use it here because it has an analytical solution. It nevertheless emphasizes the point that crustal flexure can't be ignored at these spatial scales, especially if one is trying to use slopes to detect tectonic signals. I would recommend looking at Martel et al.; they explore other uplift scenarios for the Southern Sierra which also show that tilting decays rapidly with distance. I think that incorporating crustal flexure into a tilting landscape evolution model would be a neat and interesting contribution. It may also help damp tilting in the model that led to the overestimation of tilt in the southern Sierra.

CONNECTIONS WITH PREVIOUS WORK The final issue was that the incorporation and analysis of previous work on the geological history of the Sierra Nevada could be more balanced. All but one of the geomorphic papers (Stock, 2004) marshaled to support late Cenozoic tilting are known to be flawed in one respect or another, or they are not appropriate in the context of this manuscript. In contrast, important recent papers challenging the hypothesis of recent uplift were dismissed a bit too easily or were missing from the discussion. With respect to the former group, I explore, in detail, their problems in Gabet (2014) but provide a brief synopsis here (I've uploaded a copy of this paper). I present these in the same order (more or less) as the list on page 23, line 20 and only focus on those presenting field evidence (pure modeling papers always limited by their assumptions).

1) Christensen (1966) concluded that the northern Sierra had been tilted by comparing the gradients of Lindgren's Tertiary paleochannels to modern analogs. However, the modern counterparts were small, ephemeral sand-bedded streams in semi-arid landscapes and, thus, were inappropriate and irrelevant analogs for the large, subtropical rivers capable of transporting meter-sized boulders that were draining the Sierra during the Eocene.

2) Clark et al (2005) analyzed river profiles in the southern Sierra and concluded that uplift-driven incision began carving the Kern River canyon 3.5 mya. However, this study was refuted by noting the presence of 3.5-my volcanic rocks deep within Kern Canyon.

3) The results from Huber (1981) can only constrain tilting of the southern SN, and they can only constrain tilting since the mid-Miocene. In other words, Huber (1981) cannot be used to support significant tilting over only the past 5 my, as demonstrated above.

4) Lindgren (1911) based his tilt estimate with the assumption that he correctly reconstructed the Tertiary paleochannels. Both Gabet (2014) and Cassel (2012b) demonstrated that his reconstructions were fundamentally flawed, either because they imply that water flows uphill or because he was linking channel segments that were unrelated in time and space. Moreover, Gabet (2019) demonstrates that, even if Lindgren had correctly reconstructed the channels, the differences in their gradients that he attributed to tilting can be wholly explained by differences in bedrock erodibility.

5) Jones et al (2004) based their tilt estimates on Lindgren's faulty reconstructions (see above) and are, therefore, inconclusive. This is evident by noticing that Jones et al's untilted South Yuba River has 4-5 reaches that defy gravity (i.e. water flows uphill in their Figure 3C). Note, also, that the good gradient-azimuth relationship in their Figure 3A is only possible because ~1/3 of the data points were ignored. Finally, like Lindgren, the authors did not account for bedrock erodibility.

6) Unruh (1991) based his tilt estimate of 1.4° on the gradient of Central Valley sediments. By using simple geometry, one can demonstrate that a consequence of this result is the prediction that, at some point in the mid-Cenozoic, Tertiary gravels at an elevation of ~700 m along the Yuba river were once ~500 m below sea level. In other words, untilting the northern Sierra by 1.4° places sections of the mountain range deep

underwater sometime in the past 30-50 my. This study is refuted, therefore, by the absence of deep Cenozoic marine sediments in the Sierra.

7) The Wakabayashi papers mostly base their analyses on incision depths determined from volcanic rocks, which are contradicted by the presence of these rocks deep within the canyons. The Wakabayashi papers present some other analyses which are also refuted (or challenged) by the field evidence.

8) Yeend's (1974) tilt estimate is based on Lindgren's paleochannels and suffers from the same problems as Jones et al (2004): (1) using the gradients of channels that never existed as continuous features in time or space and (2) not accounting for the role of lithology.

Therefore, the present manuscript leans on some discredited or inconclusive papers to support its results. In contrast, the published evidence for tectonic quiescence of the range throughout the Cenozoic is somewhat elided. Admittedly, the isotopic data from the leeward side of the Sierra may be confounded by the issue of terrain-blocking and, thus, citing Molnar (2010) is a reasonable counterpoint to some of these studies; however, several recent papers have avoided this problem. Mix et al. (2019) specifically addresses the terrain-blocking issue and finds no evidence for recent uplift of the southern Sierra (to be fair, this paper was just published so the authors may not have seen it). Cassel et al (2009) and Mix et al (2015) present isotopic data from the windward side of the northern Sierra, thereby avoiding the terrain-blocking problem, and also find no evidence for recent significant uplift. Cassel and Graham (2011) present a shear-stress-based paleo-slope analysis that also supports tectonic quiescence since at least the early Oligocene. Of course, there are uncertainties associated with the results from these 4 papers, however, it seems unlikely that, by coincidence, they would have all found the same result despite using 4 different techniques. In addition, recent papers describing the paleotopography of the region also merit consideration. Cassel et al (2014) and Cassel et al (2012a) provide evidence that, in the Oligocene, the Sierra Nevada was the western ramp of a high-elevation plateau, the Nevadaplano, that has

been gradually collapsing. Thus, according to these studies, it is not the Sierras that have been recently uplifted, it's the land to the east that's been sinking.

REFERENCES Amos, C. B., Audet, P., Hammond, W. C., Burgmann, R., Johanson, I. A., and Blewitt, G., 2014, Uplift and seismicity driven by groundwater depletion in central California: Nature, v. 509, no. 7501, p. 483-486.

Cassel, E. J., Breecker, D. O., Henry, C. D., Larson, T. E., and Stockli, D. F., 2014, Profile of a paleo-orogen: High topography across the present-day Basin and Range from 40 to 23 Ma: Geology, v. 42, no. 11, p. 1007-1010.

Cassel, E. J., and Graham, S. A., 2011, Paleovalley morphology and fluvial system evolution of Eocene-Oligocene sediments ("auriferous gravels"), northern Sierra Nevada, California: Implications for climate, tectonics, and topography: Geological Society of America Bulletin, v. 123, no. 9/10, p. 1699 - 1719.

Cassel, E. J., Graham, S. A., and Chamberlain, C. P., 2009, Cenozoic tectonic and topographic evolution of the northern Sierra Nevada, California, through stable isotope paleoaltimetry in volcanic glass: Geology, v. 37, no. 6, p. 547 - 550.

Cassel, E. J., Graham, S. A., Chamberlain, C. P., and Henry, C. D., 2012a, Early Cenozoic topography, morphology, and tectonics of the northern Sierra Nevada and western Basin and Range: Geosphere, v. 8, no. 2, p. 229-249.

Cassel, E. J., Grove, M., and Graham, S. A., 2012b, Eocene drainage evolution and erosion of the Sierra Nevada batholith across northern California and Nevada: American Journal of Science, v. 312, p. 117 - 144.

Gabet, E. J., 2014, Late Cenozoic uplift of the Sierra Nevada, California? A critical analysis of the geomorphic evidence: American Journal of Science, v. 314, p. 1224-1257.

Gabet, E. J., 2019, Lithological and structural controls on river profiles and networks in the northern Sierra Nevada: Geological Society of America Bulletin, v. in press.

Gabet, E. J., and Miggins, D., in review, Antiquity of canyons in the Sierra Nevada challenges the hypothesis of late Cenozoic uplift.

Hack, J. T., 1957, Studies of longitudinal profiles in Virginia and Maryland: U.S.G.S. Professional Paper 294-B, p. 45-97.

Hammond, W. C., Blewitt, G., Li, Z., Plag, H., and Kreemer, C., 2012, Contemporary uplift of the Sierra Nevada, western United States, from GPS and InSAR measurements: Geology, v. 40, p. 667 - 670.

Hietanen, A., 1973, Geology of the Pulga and Bucks Lake Quadrangles, Butte and Plumas Counties, California.

Huber, N. K., 1981, Amount and timing of late Cenozoic uplift and tilt of the central Sierra Nevada, California - Evidence from the upper San Joaquin River basin.

Lindgren, W., 1911, The Tertiary gravels of the Sierra Nevada of California: U.S.G.S Professional Paper 73.

Martel, S. J., Stock, G. M., and Ito, G., 2014, Mechanics of relative and absolute displacements across normal faults, and implications for uplift and subsidence along the eastern escarpment of the Sierra Nevada, California: Geosphere, v. 10, no. 2, p. 243-263.

Mix, H. T., Caves Rugenstein, J. K., Reilly, S. P., Ritch, A. J., Winnick, M. J., Kukla, T., and Chamberlain, C. P., 2019, Atmospheric flow deflection in the late Cenozoic Sierra Nevada: Earth and Planetary Science Letters, v. 518, p. 76-85.

Mix, H. T., Ibarra, D. E., Mulch, A., Graham, S. A., and Chamberlain, C. P., 2015, A hot and high Eocene Sierra Nevada: Geological Society of America Bulletin.

Sklar, L. S., and Dietrich, W. E., 2001, Sediment and rock strength controls on river incision into bedrock: Geology, v. 29, no. 12, p. 1087-1090.

Stock, G. M., Anderson, R. S., and Finkel, R. C., 2004, Pace of landscape evolution

in the Sierra Nevada, California, revealed by cosmogenic dating of cave sediments: Geology, v. 32, no. 3, p. 193 - 196.

Whitney, J. D., 1880, The auriferous gravels of the Sierra Nevada of California, Harvard Colln. Museum Comp. Zoology Mem., 659 p.

Please also note the supplement to this comment:
https://www.earth-surf-dynam-discuss.net/esurf-2019-24/esurf-2019-24-SC1-supplement.pdf

**ESurfD**

---

## Short Comment (SC2) · 6 Jun 2019

I apologize for not including this in my earlier comment. This final note is related to the confusion about which river, exactly, the authors were analyzing. Whether they analyzed the Middle Fork or the Rubicon is critical because the upper watershed of the Rubicon was glaciated down to an elevation of ~1000 m during the Last Glacial Maximum whereas glaciers only came down to ~1200 m on the Middle Fork. This means that the upper ~50 km of the river shown in Figure 10a would have been affected by glaciers. I don't think that glaciers significantly affected the long profile of the Rubicon itself, but valley widening by the glaciers would have altered the profiles of its tributaries (Zimmer and Gabet, 2018) which would scramble attempts to extract uplift information from those profiles. For the Rubicon valley specifically, our data show that it is more

[Figure]

U-shaped upstream of the LGM boundary, suggesting that it was widened by glaciers. If the authors are interested, I can send them the ARC layer that we created to explore glacial modification of these valleys.

Zimmer, P. D., and Gabet, E. J., 2018, Assessing glacial modification of bedrock valleys using a novel approach: Geomorphology, v. 318, p. 336-347.

---

## Author Comment (AC1) · 18 Jun 2019

Thank you so much for taking the time to provide such a thorough comment. You raise good points that we will try to address or clarify in our revisions. Below are some initial thoughts organized around the headings in your comment with italicized subheadings added by us. We will respond more formally to each comment when we complete the revisions.

First, we would like to clarify that our main goal in this paper is to illustrate the first-order fluvial geomorphic signatures of the transient response to a punctuated tilting event, not to prove that the Sierra Nevada has experienced a tilt in the late Cenozoic. Our goal in showing data from the Sierra is twofold: 1) to show that the first-order

signatures of a punctuated tilting event can be seen in a real landscape that has been proposed by many to have been tilted (though the timing and magnitude of this tilt are still debated), and 2) to demonstrate how the signatures of tilt that we show with the models can be applied to a real landscape to estimate timing and magnitude. We hope to convey that our estimates presented in this paper of both timing and magnitude of tilt in the Sierra are rough estimates (in part because of how roughly we constrain model parameter values and in part because of how we limited the topographic analysis to just a few rivers in this paper), but that, because these estimates are consistent with many previously published estimates made from multiple independent methods, the signatures could now be applied in a thorough and systematic way in attempts to extract tectonic histories from landscapes that have experienced a punctuated tilting event. We will try to clarify these objectives in a revised version.

**Response to comments on the Northern Sierra Nevada** Yes, we have analyzed the Rubicon River, which is the longest tributary to the Middle Fork American River. When doing the analysis for the entire Sierra, we used a naming convention in which we called each profile by the fork of the major river that the analyzed tributary drained into. In most cases the largest tributary has the same name as the fork it drains into for much of its length, but in this case you're right that it doesn't and that there is another, smaller tributary named the Middle Fork American River. We will put a channel head marker, a blue line for the river on the DEM, and call the river Rubicon/Middle Fork American to clarify which river was analyzed.

*Variable incision depths and Cenozoic volcanics near to the modern river level*

Thanks for bringing up the excellent field observations of extremely variable incision depths in adjacent river basins and Cenozoic deposits not far above modern river elevation on many Sierra rivers. These are the exact observations that initially drew us to studying Sierra rivers and ones we are trying to show can be explained if these river profiles are viewed as in a transient state with nonuniform erosion rates. We show a chi plot of the Rubicon / Middle Fork American and it is far from the characteristic form of

an equilibrium chi plot in which all river channels collapse on a single straight line and there are large changes in channel steepness that do not correspond strictly to rock type. We suggest that because the disequilibrium profile form of the Rubicon/Middle Fork American is consistent with a transient state of adjustment to a short-lived tilting event, only the profile downstream of the mainstem knickpoint should be a robust recorder of rock uplift that occurred in the punctuated episode of tilting. In the 1-D simulation of a transient response to a punctuated tilting event, incision depth varies from near-zero at the mountain front (at the tilt axis), increases linearly up to the mainstem knickpoint (the portion of the profile that has incised through all the rock uplift due to punctuated tilt) and decreases back to near-zero at the channel head (the portion of the profile where incision is less than rock uplift). In the simulations, maximum uplift occurs at the crest, but this is exactly the point where incision has been limited because the small drainage area limits the erosion rate and the main tilting-induced knickpoint is still far downstream. With the additional complexity of heterogeneous lithology in the 1-D model, incision depths can become highly variable upstream of the mainstem knickpoint. Rock with anomalously low erodibility can stall mainstem knickpoints and lead to little incision at that location, but still have significant incision downstream where the mainstem knickpoint has propagated and upstream where another knickpoint has kicked off in more erodible rock (see Fig. 4 and check out the simulations on Figshare at https://doi.org/10.6084/m9.figshare.8111498.v1). We suggest that the transient response to tilting modulated by heterogeneous lithology is a mechanism that can explain how incision histories can vary widely both among neighboring basins and within an individual basin. In our revised version of this paper, we will make sure to highlight this conclusion and to emphasize that, when interpreting tectonic histories from incision depths in real landscapes, it is important to analyze spatial gradients of incision rather than isolated measurements and further, that only incision downstream of the mainstem knickpoint directly records the magnitude of rock uplift that occurred in the punctuated episode of tilting.

*Also, since these canyons have been deep for a long time, knickpoints along the profiles of their tributaries cannot be used in the manner adopted here to provide evidence of recent uplift-driven incision.*

We don't follow the logic of this comment and hence are not sure how to respond. Our results would suggest that if the mainstem had indeed incised long ago, tributary knickzones formed in response to that incision should have propagated much greater distances than are observed.

*Streampower framework begins with the assumption that there is uplift to be detected and thus this technique cannot be used to determine whether or not there was uplift because it already assumes that U is not zero.*

By employing the stream power model of bedrock river incision we make no assumptions regarding the value of rock uplift. In the governing equation (equation 1) rock uplift can be any value, including zero. If rock uplift is zero, equation 1 says the time rate of change of elevation will be equal to the river incision rate, which in this case we assume scales with stream power. If one is interested in equilibrium channel steepness, $k_{sn}$, (that is, the channel steepness at which rock uplift is perfectly balanced by river incision) one can use equation 5. Uplift can be zero in equation 5, it just predicts that at equilibrium, zero channel steepness is needed to balance a rock uplift rate of zero. But yes, in order to have any channel steepness and fluvial relief at equilibrium, uplift (or base level fall) must be actively occurring. One can, however, model a transient declining state in which uplift has occurred in the past but is then reduced to zero.

*Heterogeneous lithology*

Thanks for pointing out that our approach could be made clearer in the methods section – we will revise it in a future version. We intentionally assume uniform lithology

when calculating chi in the Rubicon/Middle Fork American although we know this to be false because this approach reveals whether changes in channel steepness (the slope of the chi plot) correspond to lithologic boundaries or whether changes in steepness occur independent of rock type and might thus be a result of temporal changes in boundary conditions. Figure 10 shows that the biggest change in steepness (at chi of 75) does not occur at a lithologic boundary. The other big change in steepness (at chi of 140) does approximately correspond to lithologic boundaries and is consistent with the transient response to tilting in a band of more erodible rock upstream of the mainstem knickpoint.

*Methods used to obtain model parameter values*

In response to our choice to use the incision rate from the Stanislaus as a proxy for modern uplift rate in our calculation of $K$: Our aim is simply to present a back-of-the-envelope calculation of $K$ for the Sierra in an effort to show that timing can be calculated using these methods *if K can be constrained*, not to provide a robust estimate of tilt timing using the rough estimate of $K$. Given that the Sierra Nevada appears to be in a transient state, it is tricky to use equation 5 to estimate $K$ given that only isolated portions of the river network has equilibrium steepness. Unfortunately, there are no published erosion rates for basins that have reached equilibrium to modern boundary conditions (i.e., near the mountain front) and thus no great proxy for modern uplift rate. We chose to use the incision rate from the Stanislaus River as a proxy for modern uplift rate in the lower Rubicon/Middle Fork American because 1) it is a measurement from a mainstem river (like the Middle Fork American), and 2) it measures incision over a short, recent time period of 1.6 Myr (rather than longer term measurements made from cosmogenic nuclides) and thus is possibly closer to modern boundary conditions. Yes, the cave occurs in metamorphic rock and the lower section of the Rubicon/Middle Fork American (where the rate in question is used as a proxy for uplift rate) runs almost exclusively through metavolcanic and metasedimentary rock. We will justify our choice to use this incision rate in a revised version and also clarify that we do not think $k_{sn}$

from a single basin and an incision rate from a transient river is a robust way to estimate $K$ and hence, our estimates of timing for the Sierra are only presented as an example that, with rough parameterization, results in timing that is consistent with published research.

**Response to comments on the Southern Sierra Nevada:** Thanks for pointing out that we are missing crucial information in this section. The method we present for estimating tilt magnitude from azimuth-gradient relationships in the modern river network seems to overestimate tilt magnitude and thus gives maximum values because gradient continues to increase for millions of years following a punctuated tilting event. We will clarify that our estimate of 2.3 degrees is a maximum and a known overestimate and is thus consistent with the estimate you present of 1.1 degrees. Despite rapidly overestimating tilt magnitude, we will highlight that we think application of this method is still useful in that it demonstrates that a signature of tilt is recorded in a real modern river network in the manner predicted by a 2-D model and the existence of this signature is consistent with a recent tilting event.

**Response to comments on modeling:** We simulate the end-member scenario of instantaneous tilt with zero internal deformation as the first step to investigate first-order signatures of the transient response to rapid tilting. The transient response to this simple scenario is quite complicated on its own and we wanted to fully explore it before adding additional complications such as more realistic nonuniform uplift fields. We are not trying to exactly simulate the Sierra or any particular landscape with the 2-D landscape evolution model and for that reason we use the simplest possible model. We agree that adding more realistic flexural deformation, such as described in Martel et al., 2014, would be a great next step. To show that our end member of instantaneous tilt likely applies to more realistic tilting scenarios in which tilting occurs over a time period that is short relative to knickpoint travel times we will add an additional simulation to the supplement in which the same tilt magnitude is achieved over a finite amount of time such as 5 or 10 million years.

We aim to show that the observed azimuth-gradient relationship in the modern stream network in the Sierra Nevada is consistent with the first-order signatures predicted with a simple 2-D landscape evolution model. We will add discussion points on the timescale of tilt and internal deformation in the discussion on the limitations of the model.

**Response to comments on connections with previous work:** In a revised version we will try to provide a more balanced review of the debate and make it clear that the debate surrounding the timing and magnitude of tilt in the Sierra is active and unresolved. The tectonic history of the Sierra is not the focus of this paper and thus a thorough review of all published evidence for and against late Cenozoic tilting would be out of place. Our primary goal in this paper is to present robust, first-order signatures of the transient fluvial response to a punctuated tilting event and our use of the Sierra is simply to show that, in a landscape that many have argued has been tilted, these signatures exist.

---

## Author Comment (AC2) · 18 Jun 2019

We limited our tributary analysis to tributaries that ran exclusively through unglaciated terrain as mapped by Gillespie and Clark (2011). Thanks for calling our attention to this as we forgot to describe the criteria for picking these tributaries.

---

## Short Comment (SC3) · 21 Jun 2019

I've attached some figures that should help clarify my main comments.
* * *
[Figure]

After re-reading my original comments, I realized that they would have been clearer had I provided figures to illustrate my main points. My main comment regarding the northern Sierra Nevada was that canyons cannot be younger than the sediment found within them. Below is a revision of Figure 15B in my 2014 AJS paper; it shows a new Ar/Ar age for the volcanic deposit deep in the South Fork American River canyon, < 200 m above the bed of the modern river (this new age is 'in press'). Moreover, as I mentioned in my earlier comment, Eocene-Oligocene sediments can also be found < 170 m above the modern river bed in this canyon and near the present bed elevation in the Middle Fork as well. These three deposits are on published maps and reports, and the first two were addressed in my 2014 AJS paper. Therefore, incision of the American River drainage could not have begun 5 Ma (or 1.9 Ma); instead, these canyons are likely much older.

[Figure]

**Fig. 1.** Northern Sierra

[Figure]

My main comment regarding the southern Sierra Nevada was that it could not have experienced 2.3° of recent tilting. As noted by Huber (1981), the upper uneroded surface of a 10 Ma lava flow along the San Joaquin River (which is near where the authors did their analysis) forms a series of table mountains. The source of this flow was the Sierra Nevada (top-right of the map) and it flowed down into the Central Valley (bottom-left of the map). The line in this figure shows the transect plotted on the next page.

The upper surface of the flow is at an angle of 1.37 deg (first figure below). If we subtract the 2.3 deg of recent tilt hypothesized by the authors, the upper surface of the flow is now at -0.9 deg (ie. 1.37 – 2.3 = -0.9; second figure below). This means that the lava would have flowed uphill, a result that refutes the hypothesis that there has been 2.3 deg of tilt over the past 10 Ma.

[Figure]

**Fig. 2.** Southern Sierra

---

## Author Comment (AC3) · 22 Jun 2019

Thank you for the figures and for keeping the discussion going.

*Northern Sierra Nevada canyons cannot be younger than the sediment found within them.*

We completely agree with this point. Lindgren's original maps and more recent papers from Henry et al. and Cassell et al. paint a clear and striking picture of a large former river system that in some places shared the same locations of the modern river canyons. As you have pointed out and shown nicely in this valley cross-section from the South Fork American there are late Cenozoic volcanics quite close to the modern river elevation in many of the northern Sierra rivers. We agree that the basement rocks

must have been incised at least to that elevation before these flows arrived. We will reread the paper to ensure that we do not discount the antiquity of the canyons. That is certainly not our point. What we are suggesting in this paper is that based on our modeling and analogous disequilibrium longitudinal profile forms and patterns of incision found in the Sierra that a younger episode of incision that is still ongoing can explain what is observed in the Sierra.

Attached is an analogous figure to the one in the current paper for the South Fork American. Our guess is that valley cross-section you show comes from one of the minimum incision points around 50 or 75 straight-line kilometers from the mountain front on our plot. As we said in our previous comment, and what we are striving to convey in this paper, is that this unique nonuniform incision pattern, which repeats to first-order across the entire northern Sierra, is entirely consistent with a young transient phase of incision. Incision of just a few hundred meters above the mainstem slope-break knickpoint is consistent with this transient response. If one adds in heterogeneous lithology incision depth can become even more nonuniform. In short, we are completely on board with the canyons being older than the volcanics you describe, and we think these points of lesser incision above what we call the mainstem slope-break knickpoint make up a critical piece of the geomorphic signatures we are proposing.

**Southern Sierra**

We will consider carefully how we present our estimate of tilt magnitude. As we said in the previous reply, we meant to frame this particular estimate as a clear maximum and known overestimate but it was in a paragraph in the discussion that wound up getting cut. We will fix this issue in the revised version. We will add in the discussion you have provided to show that indeed tilt magnitude must be less than 2.3° based on nice geologic evidence to try to bring that point home more clearly. In the end we might even omit the numerical value as to not distract from the more robust result of a measurable azimuth-gradient relationship in the modern stream network.

**ESurfD**

Interactive
comment

[Figure]

[Figure]

**Fig. 1.**

---

## Referee Comment (RC1) · Anonymous Referee #1 · 25 Jun 2019

The authors have tackled an interesting problem in this manuscript in seeking the signature of transient fluvial network response to tectonic tilting. They identify characteristic features in simulated river profiles generated by 1- and 2-D landscape evolution models responding to tilting, propose a means to calculate the timing and magnitude of a tilting event based on river profile geometry, and compare their model results to river profiles from the northern Sierra Nevada, proposed by several studies to have undergone tilting during the late Cenozoic. The paper is well-written and the model setup and results are straightforward and clearly described. Identifying the characteristic topographic signature of tilting is a worthy goal and could be a valuable contribution toward understanding the climate-tectonics-topography system especially in extensional settings. The numerical modeling the authors have applied to the problem is a rea-

sonable approach, and in conjunction with analysis of natural landscapes, could yield useful results. The interpretations that the authors make of their model results seem sound, but there are some significant issues that need to be addressed.

- Why weren't any simulations included in which tilting is imposed gradually, or, even better, with a varying rate? While the instantaneous tilt scenarios are worth including, it doesn't necessarily seem reasonable to interpret the features resulting from an instantaneous tilt in the same way as those resulting from a realistic, gradual tilt. At the very least, if the authors maintain that the simplified model scenarios produce comparable features that can be interpreted in the same way, some explanation as to why should be included.

- In Figure 7b it's really difficult to tell what's going on in the chi plots. In general, I'd love to see a little more from 2-D model runs included, and analysis of the resulting profiles done in more detail. It's important to examine how even the complexity introduced by the tributaries not being perpendicular to the mainstem affects how well timing and magnitude of a tilt can be resolved. I think it would strengthen the case the authors are making by having detailed analysis of a 2-D tilt simulation as an intermediary between the 1-D analysis and the Middle Fork American River. The inclusion of the 1-D simulations of different sorts of perturbations and the features that appear in river profiles as a result is really nice for this sort of paper, but without some sort of intermediate complexity analysis before jumping right into the Sierra Nevada analysis it almost undermines the authors' conclusions by reminding the reader of all the different ways channel profiles can respond to various perturbations even in an highly simplified model. Then, by the time the analysis of the real river comes around, it leaves me wondering how meaningful the results actually are in a system that's so vastly more complex.

- Figure 7c, would we see a similar relationship between channel segment azimuth and ksn along a linear mountain front where uplift was not driven by tilting?

- 6.1, Line 14: Would 5 Ma. be the onset or termination of tilting?

- 6.1, Line 26: I have a hard time with any interpretations of knickzones that were above the glacial limit as containing meaningful information about tectonics.

- I don't quite understand the rationale of choosing this location to test the model results. I don't know much about Sierra Nevada tectonics, so I'll defer to others on whether recent tilt is a valid hypothesis, but even just the combination of extensive Pleistocene glaciation and lithologic heterogeneity seems like it would make it a difficult place to make a comparison to simple, 1-D model results. Even in the unglaciated reaches of these rivers, sediment supply and discharge would have been varying wildly throughout the Quaternary. Not to say the Sierra Nevada stuff should be thrown out, but it might be more convincing to include some analysis of river profiles from a simpler tilted-block range. The model results in general are straightforward, but where K, m, and n seem like they could vary so widely in space and time, I just don't know that I trust interpretation of the knickzones in this river as being tilt-related features.

- In 6.2 and Figure 11, how was it determined whether a knickzone collapsed with the mainstem? Was it just determined visually or were there some other criteria?

- 6.3, Line 23: Shouldn't this degree of tilting be causing pretty rapid migration of the main divide? That could really complicate sorting out knickpoint migration velocities.

- Along with this, it might be really cool to analyze the back-tilted catchments on the opposite side of a tilting range in conjunction with the forward-tilted ones.

- Figure 13 could really benefit from a chi plot or something to better show the capture event. It's hard to tell exactly why it's being interpreted that way.

Overall, this is an interesting and readable manuscript that with additional analysis (a detailed look at a 2-D model river network and/or analysis of river profiles from an area with simpler lithology and a simpler history) could support its conclusions much more strongly. As is, I still have some doubts about how robustly the signature of tilting is

preserved.

**ESurfD**

---

## Short Comment (SC4) · 27 Jun 2019

I thank the authors for their response to my comments, and it is encouraging that we have found common ground on the issue of the antiquity of the canyons. However, the key point that I was trying to convey regarding the age of the canyons has to do with the authors' incision analysis and its underlying assumption. Fortunately, the authors have offered their analysis of the South Fork American River, which will be helpful to clarify my point. On the first figure, I show the mapped location of Eocene-Oligocene gravels within the canyon of the South Fork American River (I've visited this site to confirm their presence). These gravels are ∼250 m below the volcanic rocks that the authors are using to calculate recent bedrock incision depths. I've plotted, on their figure, the approximate position of this deposit. The assumption underlying the analysis shown in

the inset plot is that the difference in elevation between the lowest volcanic rocks and the modern river bed is a measure of recent incision. However, the presence of the Eocene-Oligocene gravels below the volcanic rocks refutes this assumption. Indeed, the fact that these ancient gravels underlie the volcanic deposits was well known by miners during the Gold Rush; they tunneled down through the volcanic deposits to get at the gold in the underlying Eocene-Oligocene gravels. I've included a map showing the gravels (Ng) below the andesitic volcanic deposits (Na) near Placerville (this is from a 1:125000 map; these small deposits are not shown on the 1:250000 map). This map shows that there are Eocene-Oligocene gravels below the volcanic deposits at the location where the authors have added a black arrow on their profile of the river. Thus, in order for their S Fk American River figure to depict the actual distribution of the Cenozoic deposits, they would need to include the Eocene-Oligocene gravels that underlie the volcanic deposits. Because these older and deeper deposits are not shown, this figure is missing critical information.

The present distribution of the remnant volcanic deposits is simply due to the fact that volcanic deposits that were once in the valley and along the steep valley walls have been eroded away; the last figure shows some small remnants of these volcanic rocks along the valley wall near Riverton on the South Fork. So, my point is that the distribution of the volcanic rocks does not define a pre-incision paleosurface and, therefore, cannot provide information on incision depths. The present distribution of the volcanic rocks is an artefact of hillslope erosion (ie. deposits on steep valley walls were eroded away). Finally, please note that the information about the Eocene-Oligocene deposits is presented in my 'in review' paper and, therefore, is embargoed and cannot be used elsewhere.

Finally, my GSAB paper showing that lithology has a first order control on channel steepness is now in press. I've attached the proofs as a supplement.

Please also note the supplement to this comment:

https://www.earth-surf-dynam-discuss.net/esurf-2019-24/esurf-2019-24-SC4-supplement.pdf

**ESurfD**
[Figure]

The position of the Eocene gravels above (Tc in the middle of the map) is shown below.

[Figure]

**Fig. 1.** S Fk AR 1

[Figure]

**Fig. 2.** S Fk AR 2

**Supplement:**

**Lithological and structural controls on river profiles and networks in the northern Sierra Nevada (California, USA)**

**Emmanuel J. Gabet[†]**
*Department of Geology, San Jose State University, San Jose, California 95192, USA*

**ABSTRACT**

In this study, the strong lithological heterogeneity of the northern Sierra Nevada (California, USA) is exploited to elucidate the role of lithology on river profiles and patterns at the mountain-range scale. The analyses indicate that plutonic, metavolcanic, and quartzite bedrock generally host the steepest river reaches, whereas gentle reaches flow across non-quartzite metasedimentary rocks and fault zones. In addition, the largest immobile boulders are often in the steepest reaches, suggesting that wide joint spacing plays a role in creating steep channels, and a positive relationship between boulder size and hillslope angle highlights the coupling of the hillslope and fluvial systems. With respect to river network configurations, dendritic patterns dominate in the plutonic bedrock, with channels aligned down the slope of the range; in contrast, river reaches in the metamorphic belts are mainly longitudinal and parallel to the structural grain. River profiles and patterns in the northern Sierra Nevada, therefore, bear a strong lithological imprint related to differential erosion. These observations indicate that attempts to infer uplift and tilting of the range based on the gradients and orientations of paleochannel remnants should first account for the effect of bedrock erodibility. Indeed, the differences in gradients of Tertiary paleochannel remnants used to argue for late Cenozoic uplift of the range can be wholly explained by differences in lithology.

**INTRODUCTION**

The longitudinal profile of a river is often idealized as a smooth concave form, with steep reaches in the headwaters gradually transitioning to gentler reaches as the channel loses elevation (e.g., Inoue, 1992; Larue, 2008). In the upper reaches, where channels incise into bedrock, deviations from this shape can be attributed to a variety of causes, such as changes in lithology along the river's course (e.g., Duvall et al., 2004; Lecce, 1997; Phillips and Lutz, 2008; Pike et al., 2010), spatial variations in the delivery of coarse sediment to the channel (e.g., Finnegan et al., 2017; Hack, 1973; Hanks and Webb, 2006), and drainage capture (Fan et al., 2018). River profiles may also be modified by crustal deformation, thereby offering the potential for obtaining information about tectonic activity that might otherwise be difficult to acquire (e.g., Pavano et al., 2016).

However, before river profiles can be reliably used for inferring tectonic activity, other factors that can affect their shape must first be considered. Although important advances have been made, the development of mechanistic theories for describing the incision processes that determine a bedrock river's profile is still ongoing (e.g., Chatanantavet and Parker, 2009; Lamb et al., 2015; Sklar and Dietrich, 2006b; Whipple et al., 2000). Indeed, our understanding of the controls on bedrock channel gradients is incomplete and predicting how steep a reach should be, even given complete information regarding discharge, underlying lithology, sediment supply, and tectonic history, remains challenging.

This study presents a large-scale analysis of the influence of lithology and structure on river profiles and network configurations along the western slope of the northern Sierra Nevada (California, USA) (Fig. 1). Because of its history as a subduction zone during the Jurassic and Cretaceous periods, this region is lithologically complex (e.g., Snow and Scherer, 2006). From the western margin of the range to its eastern ridgeline, the pre-Cenozoic basement generally consists of north-south–trending belts of Mesozoic metasedimentary and metavolcanic rocks, parallel belts of Paleozoic metasedimentary and metavolcanic units, and, finally, plutonic rocks (albeit with some isolated remnants of Mesozoic metamorphic rocks) underlying the crest (Saucedo and Wagner, 1992; Wagner et al., 1981). Although plutonic rocks dominate the eastern half of the range, there are numerous smaller plutonic intrusions throughout the metamorphic belts. Because the belts of metamorphic rocks, separated by fault zones, are aligned parallel to the ridge-crest, rivers flowing down the western slope of the range encounter a variety of rock types with differing erodibilities; this region, therefore, provides an opportunity for examining the role of spatially varying bedrock on river profiles and planforms. Moreover, an analysis of the modern rivers is critical for assessing the results of studies that have used the profiles of Sierran paleochannels to infer late Cenozoic uplift of the range (Hudson, 1955; Jones et al., 2004; Lindgren, 1911; Yeend, 1974).

**MATERIALS AND METHODS**

**Longitudinal Profiles and Slope Calculations**

This study focuses on five major river systems draining the northern Sierra Nevada (Fig. 1). The headwaters of many of these rivers are located on a relatively low-relief bedrock surface that dominates much of the high-altitude terrain in the region; from this low-relief surface, the rivers cascade down into deep canyons with steep slopes. The longitudinal profiles of five trunk streams (Fig. 1) were extracted from 30 m digital elevation models (DEMs) in a geographical information system (GIS) using standard techniques (O'Callaghan and Mark, 1984). The upper extents of the South Yuba River, the South Fork American River, and the North Fork Mokelumne River profiles were established where the drainage area exceeds ~100 km[2] to avoid, as much as possible, sections where these channels flow across the low-relief bedrock surface and have failed to incise into it to an appreciable degree. The upper extent of the North Fork Feather River profile was established downstream of a region of faults active in the Quaternary (Saucedo and Wagner, 1992), and the upper extent of the Merced River was established downstream of the alluviated Yosemite Valley. The downstream extent of each profile

[†]manny.gabet@sjsu.edu

*GSA Bulletin*;    https://doi.org/10.1130/B35128.1; 13 figures; 2 tables; Data Repository item 2019274.

*E.J. Gabet*

[Figure]

**Figure 1. Generalized lithological map of study area (Sierra Nevada, California, USA). Labeled rivers were analyzed in this study. Modified from Parrish (2006). N.—North; S.—South; R.—River; Mz—Mesozoic; Pz—Paleozoic; volc.—volcanic; sed.—sedimentary.**

was terminated where it meets one of the large reservoirs built along the Sierran foothills.

The elevations of lithological boundaries along each channel profile were determined from paper maps (Bateman and Krauskopf, 1987; Hietanen, 1973; Peck, 2002; Saucedo and Wagner, 1992; Wagner et al., 1991; Wagner et al., 1981) and a GIS layer (Ludington et al., 2005). The different sources were used to cross-check the precise locations of the boundaries.

The average channel slope across each lithological unit was calculated from the difference in elevation of the reach, from one lithological boundary to the next, and the length of the reach (note, the term "reach" is used throughout to describe a section of river flowing across a single lithological unit). To account for the tendency of river gradients to decrease with increasing drainage area (Hack, 1957), the steepness index ($k_s$), which normalizes river gradient ($S$) according to drainage area ($A$), was calculated with

$$k_s = A^\theta S \qquad (1)$$

where $\theta$ is the concavity of a channel's profile (Flint, 1974). By determining a common concavity between watersheds (i.e., a reference concavity, $\theta_{ref}$), values of $k_s$ for individual reaches of different rivers can be compared (Wobus et al., 2006). A reference concavity of 0.35 for the study area was calculated by analyzing the longitudinal profiles of tributary watersheds (Supplemental Information, Fig. DR1[1]) using the chi analysis algorithm in LSDTopoTools (Mudd et al., 2014; Mudd et al., 2018a; Mudd et al., 2018b; Perron and Royden, 2013). The use of Equation (1) to provide a steepness index is not meant to imply that the region is in topographic steady-state (see later). This approach is adopted because it facilitates comparisons of the gradients of individual reaches, but it is not used to investigate the overall profiles of the channels.
* * *
[1]GSA Data Repository item 2019274, calculations of reference concavity and peak discharges, is available at http://www.geosociety.org/datarepository/2019 or by request to editing@geosociety.org.

Finally, note that the upper sections of three of the rivers studied here (the South Yuba, the North Fork of the Mokelumne, and the Merced rivers) were occupied by glaciers during the Pleistocene ice ages (Gillespie and Clark, 2011). However, as with other glaciated canyons in the Sierra Nevada (Brocklehurst and Whipple, 2002; Dühnforth et al., 2010; Matthes, 1930; Zimmer and Gabet, 2018), vertical incision by the glaciers appears to have been limited (Bateman and Wahrhaftig, 1966), and there is no evidence that the longitudinal profiles of these valleys were substantially affected. In any case, since both rivers and glaciers incise via plucking and abrasion, differences in rock erodibility ought to be similarly reflected in the profiles they create.

**Channel and Valley Characteristics**

Lithological differences may not only be expressed in channel slopes but also in channel widths and the size of the bed sediment (e.g., Duvall et al., 2004; Montgomery and Gran, 2001). Moreover, in mountainous terrain where the valleys are narrow, the fluvial system is tightly coupled to the adjacent hillslopes; fluvial processes, therefore, may be influenced by the surrounding terrain. These additional factors were examined along a subset of the studied channels. Because reaches along these rivers are typically inaccessible due to steep terrain, bankfull channel width was measured at ~500 m intervals from high-resolution (0.5 m) Google Earth imagery using the ruler tool along the North Fork of the Feather River, the South Yuba River, and the Merced River. These three were chosen because, of the five investigated here, their lithology has been mapped in the greatest detail and the available Google Earth imagery for these sites is of high quality (e.g., the water is at low flow and clear, there are few shadows due to a high sun angle, etc.). The photographs were taken during low-flow conditions in the summer of 2017, therefore visual cues, including vegetation lines and color differences along the banks, were used to estimate bankfull conditions. Record rainfall during the winter and spring of 2016–2017 (Lin and St. John, 2017), before the photographs were taken, led to high discharges that left behind clear trimlines along most of the reaches. Moreover, the V-shaped cross-section of many of the valleys implies that channel width increases slowly with discharge, thus constraining errors in these measurements. Nevertheless, I estimate an uncertainty of ±10 m in the measurement of the bankfull channel widths.

The long axis of the largest boulder within a radius of ~25 m around each channel-width-measurement site was also estimated with the

ruler tool in Google Earth for the three afore-mentioned rivers and along an anomalous section of the North Fork Mokelumne River. Because of the low-flow conditions in the images and the clarity of the water, large boulders can be easily seen in the imagery. However, for clasts smaller than 0.5 m, a default value of 0.5 m was assigned. The maximum uncertainty in these measurements is likely on the order of ±2 m for the largest boulders, primarily because the lower parts of these clasts were submerged and their outlines sometimes difficult to identify precisely.

Google Earth does not provide information regarding its orthorectification procedures; this issue, as well as others regarding Google Earth imagery (Fisher et al., 2012), suggests that some measurements could be incorrect because of photographic distortion. To gauge the potential for measurement error, orthorectified aerial photographs from the National Agricultural Imagery Program (NAIP, 2018) for a subsample of the sites were imported into a GIS. The dimensions of stable features such as large boulders and roads were measured from both the NAIP (0.5 m resolution; Fisher et al., 2012) and Google Earth imagery and compared. The average difference in measurements between both sets of photographs was $0.2 \pm 0.2$ m (n = 20), indicating minimal errors due to photographic distortion.

Valley relief along each channel was calculated from the 30 m DEMs by subtracting the river elevation from the highest elevation found within a 1000 m radius, a distance sufficient to include the valley rims. The mean angle of the hillslopes along each river was determined by averaging the slopes of all cells with slopes >15° within a 240 m radius of each river cell; cells with slopes ≤15° were excluded to avoid floodplains, terraces, and fans. The radius was chosen to avoid terrain above the canyon rims.

**Shear Stress Calculations**

A river's boundary shear stress ($\tau_b$) plays an important role in controlling the rates of channel incision (e.g., Lamb et al., 2015). For the three rivers where boulder size and channel width were measured along their entire study length, reach-averaged maximum shear stresses were estimated with

$$\tau_b = \gamma Q^{0.6} n^{0.6} S^{0.7} w^{-0.6} \qquad (2)$$

where $\gamma$ is the unit weight of water, $Q$ is discharge, $n$ is Manning's roughness coefficient, $S$ is the reach-averaged slope, and $w$ is the reach-averaged channel width (Snyder et al., 2003). For each reach, $Q$ was estimated from the contributing area based on discharge records during the January 1997 floods in the northern Sierra Nevada (Supplemental Information; see footnote 1). Manning's n was estimated with $n = 0.034 D_{90}^{1/6}$ where $D_{90}$, the ninetieth percentile of the particle size distribution, was approximated by reach-averaged values of the maximum boulder size (USACE, 1994); although this approach likely overestimated the roughness coefficient and, thus, the shear stress, the error is tempered somewhat by the low value of the exponent.

**River Network Analysis**

To investigate the role of lithology and structure in controlling the shape of the river networks, all the channels within the study region were extracted from a 30 m DEM using an accumulation area threshold of 10 km². The channels were split into ~5 km sections, and the azimuth of each section was calculated in ARCGIS using the EasyCalculate 10 add-in (www.ian-ko.com /free/EC10/EC10_main.htm). In addition, the bedrock underlying each section was extracted from a digital lithological map of California (Ludington et al., 2005). Channel sections in metamorphic bedrock east of the Melones Fault Zone (Fig. 1) were excluded from the analyses because of the dominance of a subduction mélange (the Paleozoic Calaveras Complex) in which the various lithological units have been jumbled together (Snow and Scherer, 2006).

**RESULTS**

**River Gradient and Lithology**

The results from the profile analyses are presented from north to south. Along the North Fork Feather River (henceforth referred to as the NF Feather River), all of the breaks-in-slope are found at or near lithological contacts (although not every contact produces a break-in-slope), and the gradients of individual reaches often vary according to lithology (Fig. 2). The reaches with the steepest median $k_s$ are underlain by metavolcanic rocks ($0.23 \pm 0.04$; Table 1), followed by reaches flowing over plutonic rocks ($0.15 \pm 0.02$) and non-quartzite metasedimentary bedrock (0.02). Interestingly, within the plutonic rocks, the difference in gradients can be substantial. For example, the slope across the lower exposure of the quartz diorite (KJhqd at km-19) is ~3× times steeper than the reach across the pyroxene diorite just upstream (KJpd at km-15). The gentlest reach is across a fault zone (ultramafic/metavolcanic at km-52); where the river crosses similar bedrock upstream (ultramafic/metavolcanic/metasedimentary at km-40), the gradient is 3× times higher.

Along the South Yuba River (Fig. 3A), the plutonic and quartzite rocks have the highest

median $k_s$, $0.20 \pm 0.06$ and $0.20 \pm 0.07$, respectively (Table 1). In contrast to the NF Feather River, the unfaulted metavolcanic bedrock has a low $k_s$ (0.08), similar to the non-quartzite metasedimentary rocks ($0.09 \pm 0.01$). Where the river crosses the fault zone near km-80, the profile abruptly flattens. The overall profile of the South Yuba River is stair-stepped, with steep reaches over plutonic rock connected to gently-sloping reaches across fault zones and non-quartzite metasedimentary rock.

The profile of the South Fork American River (henceforth referred to as the SF American River) also has breaks-in-slope associated with lithological contacts (Fig. 4). This trend is particularly apparent in the section of river from km-32 to km-72, where steep reaches across granitic rocks (median $k_s = 0.16 \pm 0.09$; Table 1) are juxtaposed with gentle reaches across mainly non-quartzite metasedimentary rocks (median $k_s = 0.10 \pm 0.01$). There are exceptions, however, to this general pattern; for example, in the section of river between km-20 and km-34, a plutonic unit has slopes comparable to its non-quartzite metasedimentary neighbors. Similarly, the reach of river flowing over granitic bedrock at km-80–90 has nearly the same $k_s$ as the adjoining downstream reach flowing across non-quartzite metasedimentary rocks. Finally, note the significant break-in-slope in the upper reaches of the SF American River (km-8). Mapping of the plutonic rocks in this watershed has not been as detailed as in others, and the bedrock has only been generically identified as Mesozoic granite (Wagner et al., 1981); this break-in-slope, therefore, may be associated with an unmapped inter-plutonic contact (compare with km-10–28 on the NF Feather River; Fig. 2A).

Along the North Fork Mokelumne River (henceforth referred to as the NF Mokelumne River), the reaches flowing over the plutonic rocks are generally steeper (median $k_s = 0.17 \pm 0.08$; Table 1) than those over the non-quartzite metasedimentary rocks ($0.09 \pm 0.04$). A notable feature of this river is the cascade (km-65–70) straddling the lithological contact between granitic rocks (Mzg; Fig. 5) and the metasedimentary rocks of the Calaveras Complex (Pzcc). The metasedimentary bedrock has been deeply incised relative to the granitic rocks just upstream to form a ~200-m-high knickpoint. At the base of the knickpoint, a large tributary increases the drainage area by 40%, and the abrupt increase in flow may be partially responsible for the deeper incision through the weaker rock.

Of the five rivers analyzed in this study, the reaches of the Merced River (Fig. 6A) have the greatest difference in median $k_s$ between the plutonic rocks ($0.40 \pm 0.05$) and the non-quartzite metasedimentary bedrock ($0.08 \pm 0.02$). This

*E.J. Gabet*

**Figure 2. (A, B) Longitudinal profile, lithology, and other data along the North Fork Feather River (adapted from Johnson, 2015). Gradient and $k_s$ (steepness index; in italics) shown above each reach. Plutonic rocks designated by patterned fills; metavolcanic rocks designated by hatch pattern; metasedimentary rocks designated by solid shading. Pzcc—metachert, argillite; KJhqd—coarse-grained quartz diorite; KJpd—fine- to medium-grained pyroxene diorite; KJmt—coarse-grained monzotonalite; MzPz db—metabasalt; um—ultramafic; mv—metavolcanic; mg—metagabbro; ms—metasedimentary; the four rock types at the end of the profile are of Mesozoic and Paleozoic ages (Hietanen, 1973; Saucedo and Wagner, 1992). Fill patterns represent the distribution of bedrock units at the surface and not the underlying structure. Gaps in $D_{max}$ (maximum particle size) are due to reservoirs.**

[Figure]

sharp difference is perhaps attributable to the segregation of the different rock types into zones that are more distinct than along the other four channels. The quartzite bedrock has a $k_s$ value of 0.17, intermediate between the other two rock types. The differences in gradients between the different plutonic bedrocks are modest, and the two main breaks-in-slope in the profile are within individual reaches rather than at their boundaries. The first, at km-2, leads into a relatively flat section where the river follows a master joint, and the second, at km-11, is discussed later.

**Channel and Hillslope Characteristics**

No relationship between channel widths and lithology could be detected (Fig. 7). The maximum particle size ($D_{max}$), however, appears to be controlled by lithology whereby, in general, $D_{max-plut.} > D_{max-metavol.} > D_{max-qtzt} > D_{max-metased.}$ (Table 2). Considering that many of the boulders are angular and quite large, they likely have not experienced much fluvial transport; indeed, estimated shear stresses during one of the largest floods on record were 10× lower than the critical shear stresses (Fig. 8). The spatial distribution of $D_{max}$, therefore, should reflect the delivery of sediment from the adjacent slopes (Hack, 1957). Indeed, $D_{max}$ is positively correlated with average hillslope angle in plutonic and metavolcanic terrain (Fig. 9). A similar relationship could not be detected in the metasedimentary rocks, perhaps because this rock type sometimes produced clast sizes below the limit of resolution (i.e., smaller than 0.5 m). Within the plutonic rocks,

joint density appears to play an important role in the maximum size of boulders entering the rivers. For example, on the NF Feather River, the source of the 22 m boulder at km-31 is a set of large, unjointed exfoliation sheets forming a cliff above the channel (note the steep hillslopes

at this site; Fig. 2B). Similarly, at km-18, there is an abrupt 3-fold increase in $D_{max}$ from one plutonic rock to the next, presumably a consequence of joint spacing. A peak in boulder size is also coincident with a steep reach along the South Yuba River at km-22–28 (Fig. 3B).

TABLE 1. MEDIAN $k_S$ AND MEDIAN ABSOLUTE DEVIATION ACCORDING TO ROCK TYPE

| | Plutonic | | Metavolcanic | | Quartzite | | Metasedimentary without quartzite | |
|---|---|---|---|---|---|---|---|---|
| NF Feather River | 0.15 ± 0.02 | (5) | 0.23 ± 0.04 | (4) | N.D. | | 0.02 | (1) |
| South Yuba River* | 0.20 ± 0.06 | (7) | 0.08 | (1) | 0.20 ± 0.07 | (2) | 0.09 ± 0.01 | (2) |
| S Fk American River | 0.16 ± 0.09 | (6) | 0.05[†] | (1) | N.D. | | 0.10 ± 0.01 | (6) |
| NF Mokelumne River[§] | 0.17 ± 0.08 | (7) | N.D. | | N.D. | | 0.09 ± 0.04 | (5) |
| Merced River | 0.40 ± 0.05 | (5) | N.D. | | 0.17 | (1) | 0.08 ± 0.02 | (2) |
| All rivers | 0.17 ± 0.07 | (30) | 0.19 ± 0.09 | (6) | 0.17 ± 0.04 | (3) | 0.09 ± 0.02 | (16) |

*Note:* Rock units in fault zones were excluded. Sample sizes in parentheses. N.D.—no data; S Fk—South Fork; NF—North Fork; $k_s$—steepness index.

*Only reaches after km-20 were included to avoid the low-relief region where fluvial incision appears insignificant.

[†]Highly sheared mélange terrane (Wagner et al., 1981).

[§]Reaches at km-60 and km-70 (Fig. 5) were split because of the large differences in gradients within the individual units.

*Lithological and structural controls on Sierran rivers*

[Figure]

Figure 3. (A, B) Longitudinal profile, lithology, and other data along the South Yuba River. Gradient and $k_s$ (steepness index; in italics) shown above each reach. Plutonic rocks designated by patterned fills; metavolcanic rocks designated by hatch pattern; metasedimentary rocks designated by solid shading. KJgr—granite, granodiorite; Jms—metasedimentary; Jmv—metavolcanic; Jdi—diorite; Pzsf ss—quartzite, phyllite; Dbg—granite, granodiorite; Pzp—peridotite; Pzcc—metachert, argillite; Pzcv—metavolcanic; MzPz ms—metasedimentary; Jgr—granite, granodiorite; gb—gabbro, diabase; um—ultramafic; mv—metavolcanic; qd—metadiorite; the three rock types at the end of the profile are of Mesozoic and Paleozoic ages (Hietanen, 1973; Saucedo and Wagner, 1992). Fill patterns represent the distribution of bedrock units at the surface and not the underlying structure. Data gaps are due to a large reservoir. "LGM" points to glacial extent during the last glacial maximum. $D_{max}$—maximum particle size.

[Figure]

Figure 4. Longitudinal profile and lithology along the South Fork American River. Gradient and $k_s$ (steepness index; in italics) shown above each reach. Plutonic rocks designated by patterned fills; metavolcanic rocks designated by hatch pattern; metasedimentary rocks designated by solid shading. Mzg—granitic; Pzu?—undifferentiated; Pzcc—metachert, argillite; Jm—slate, greywacke, conglomerate; ms—metasedimentary (mélange terrane); mv—metavolcanic (mélange terrane) (Wagner et al., 1981). Fill patterns represent the distribution of bedrock units at the surface and not the underlying structure.

*E.J. Gabet*

[Figure]

Figure 5. Longitudinal profile and lithology along the North Fork Mokelumne River. Gradient and $k_s$ (steepness index; in italics) shown above each reach. Plutonic rocks designated by patterned fills; metasedimentary rocks designated by solid shading. Mzg—granitic; Mzd—dioritic; Pzu?—undifferentiated; Pzcc—metachert, argillite (Wagner et al., 1981). Fill patterns represent the distribution of bedrock units at the surface and not the underlying structure. "LGM" points to glacial extent during the last glacial maximum.

[Figure]

Figure 6. (A, B) Longitudinal profile, lithology, and other data along the Merced River. Gradient and $k_s$ (steepness index; in italics) shown above each reach. Plutonic rocks designated by patterned fills; metasedimentary rocks designated by solid shading. Ke—coarse-grained granite, granodiorite; Kbl—medium-grained tonalite, granodiorite, quartz diorite; Ka—medium-grained granodiorite; Pzp qtzt—quartzite; Trh—phyllite, chert; TRb—phyllite (Bateman and Krauskopf, 1987; Peck, 2002). Fill patterns represent the distribution of bedrock units at the surface and not the underlying structure. "LGM" points to glacial extent during the last glacial maximum. $D_{max}$—maximum particle size.

*Lithological and structural controls on Sierran rivers*

[Figure]

**Figure 7. Reach-averaged channel widths increase with drainage area. The overlap of data from different rock types suggests that channel width is insensitive to lithology. The exponent on drainage area is lower than reported values from other regions (e.g., Lague, 2014), perhaps a result of the strong precipitation gradient driven by orographic effects (Mulch et al., 2006). Less precipitation at lower elevations would lead to lower discharges and narrower channels than might otherwise be expected given the drainage areas. Error bars = 1σ. w/o qtzt—without quartzite.**

TABLE 2. REACH-AVERAGED $D_{MAX}$ (m) ± 1σ ACCORDING TO ROCK TYPE

| | Plutonic | | Metavolcanic | | Quartzite | | Non-quartzite metasedimentary | |
|---|---|---|---|---|---|---|---|---|
| North Fork Feather River | 5.7 ± 4.7 | (34) | 5.3 ± 2.6 | (24) | N.D. | | 1.2 ± 0.7 | (8) |
| South Yuba River | 4.7 ± 2.4 | (65) | 2.9 ± 1.8 | (13) | 3.2 ± 1.4 | (25) | 2.7 ± 1.9 | (49) |
| Merced River | 8.2 ± 4.8 | (32) | N.D. | | 2.2 ± 1.0 | (13) | 2.5 ± 1.4 | (28) |

*Note:* Sample sizes in parentheses. N.D.—no data; $D_{MAX}$ (m)—maximum particle size.

[Figure]

**Figure 8. Shear stress increases with maximum boulder size. Regression for solid line is in regular font and regression for dashed line is in italics; both regressions are statistically significant (p < 0.05). Given the noise in the channel width data and the uncertainty in the discharge data, the strength of the correlation is surprising. A dimensionless shear stress for particle size $D$ can be calculated with $\tau^* = \tau_h/(\rho_s - \rho)gD$, where $\rho_s$ and $\rho$ are the density of rock (2600 kg/m³) and water, respectively (e.g., Lamb et al., 2015). Substituting the italicized regression into this equation yields 0.004 for $\tau^*$, a value that is ~10× lower than the shear stress needed to move these large clasts. qtzt metased.—quartzite medasedimentary; $D_{max}$—maximum particle size.**

There are no statistically significant differences in relief and average hillslope angle (typically ranging from 30–32°) between the plutonic, metavolcanic, and metasedimentary rocks. Nevertheless, particularly steep and deep canyons are associated with individual units. For example, two peaks in average hillslope angle are centered over narrow lithological units on the NF Feather River: metabasalt at km-29 and metagabbro at km-43 (Fig. 2B). These locally steep areas are associated with greater relief as well. The relationship between steep slopes, high relief, and individual lithological units finds its greatest expression along the South Yuba River, with peaks in these topographic metrics narrowly centered over units of granite (km-25), metachert (km-44), diorite (km-67), and quartz diorite (km-85) (Fig. 3B) (note that the gentle hillslopes near the headwaters of the South Yuba River reflect the low-relief topography of the Sierra Nevada uplands). A dip in hillslope angle, likely because of rock damage, is associated with a fault zone at km-40–41 along the South Yuba River (the other major dip, at km-24, is due to a junction with a large tributary). On the Merced River, peaks in hillslope angle and relief are centered over the Bass Lake tonalite (Kbl; Fig. 6B). Finally, relief and average hillslope angle were poorly correlated to channel gradient and $k_s$ (data not shown); for example, the steepest hillslopes overlook the gentlest reach of the Merced River (km-30–40; Fig. 6A). The absence of a correlation between the river gradients and hillslope angles suggests that the valley walls are near their threshold slopes (except in the high-elevation, low-relief regions).

**Reach Orientation and Lithology**

There were 170 ~5 km reaches extracted from the metamorphic belts and 468 from the plutonic rocks. Longitudinal reaches dominate in the metamorphic rocks, indicating that their orientations are controlled by the NNW-SSE structural grain of the Sierra Nevada foothills (Fig. 10). In contrast, reaches flowing across plutonic bedrock are generally oriented down the slope of the range and, thus, their azimuths do not appear to be structurally controlled. The increased development of longitudinal channels in the metamorphic rocks is also apparent on regional geological maps (e.g., Wagner et al., 1991; Wagner et al., 1981) which show that many reaches in the metamorphic belts are parallel or subparallel to the structural grain and have 90° junction angles with other streams. In contrast, where these networks extend into plutonic rocks, they become dendritic.

*E.J. Gabet*

[Figure]

**Figure 9. Maximum size of river boulders increases with hillslope angle in plutonic and metavolcanic bedrock. Both regressions are statistically significant. Error bars = 1σ.**

**DISCUSSION**

**River Gradient and Lithology**

Along the five rivers analyzed here, changes in channel gradient are often associated with lithological transitions. The steepest reaches are typically in plutonic, metavolcanic, and quartzite bedrock while the gentlest reaches are generally underlain by non-quartzite metasedimentary bedrock (Table 1); Hack (1973) made similar observations in Appalachian streams. Indeed, for all five rivers analyzed, the median $k_s$ values for the plutonic, metavolcanic, and quartzite rocks are about twice that of the non-quartzite metasedimentary rocks (Table 1). Also, for four of the five rivers, the highest median $k_s$ value is in the plutonic rocks and the lowest is in the non-quartzite metasedimentary rocks. These consistent findings demonstrate that channel gradient is strongly influenced by lithology. If the steep reaches were due to actively migrating knickpoints, they would be randomly distributed across all of the lithologies.

There are exceptions to these general trends, however. For example, a reach of the NF Mokelumne River flows over granite with the same gradient as the adjacent reach flowing over chert and argillite (km-80–90; Fig. 5), and, on the Merced River (km-20; Fig. 6A), a short reach underlain by granitic rock has a similar gentle gradient to the adjacent downstream reach flowing across phyllite and chert. This latter example is striking because the granitic bedrock lining this gentle reach also lines the steepest reach in the profile (km-15–20). The presence of these low-gradient granitic reaches at lower elevations suggest that they might have been exposed more often to subsurface water than their counterparts at higher elevations and, therefore, the bedrock may have been weakened by chemical weathering (Callahan et al., 2019; Wahrhaftig, 1965).

In steady-state landscapes, where the uplift rate is matched by the incision rate, the gradients of bedrock rivers are typically assumed to be adjusted such that they are able to generate shear stresses sufficient to transport the supplied bedload and incise the bed (e.g., Lamb et al., 2015; Sklar and Dietrich, 2006a). In lithologically diverse terrain under steady-state conditions, therefore, the gradient of each reach will be adjusted to mobilize the sediment supplied to it from local and upstream sources, which may include different lithologies, and to incise the bedrock at a rate that matches the uplift rate (Duvall et al., 2004; Finnegan et al., 2017;

Hack, 1957; Hack, 1973). Thus, reaches underlain by resistant rock and/or supplied with large sediment will be steeper than those underlain by weak rock (e.g., Lecce, 1997).

The presence of high-elevation low-relief surfaces separated by knickpoints from high-relief regions at lower elevations and the absence of convincing evidence for sustained late Cenozoic uplift (Gabet, 2014 and references therein), however, indicate that the northern Sierra Nevada is not at steady-state. Nevertheless, reach gradient and resistance to fluvial erosion appear to be correlated in the actively incising portions of the landscape, and this correlation may be related to clast size. For example, the steep reach cut into quartz diorite at km-19 along the NF Feather River is associated with a 15-fold increase in the maximum sediment size relative to the gentler reach upstream cut into the same bedrock (km-10) (Fig. 2B). Similarly, along the South Yuba River, a positive relationship between the largest boulders and some of the steepest reaches can be seen at km-23–27 and km-70–76 (Fig. 3B).

The relationship between $D_{max}$ and estimated shear stresses during the 1997 flood (Fig. 8) suggests that the effect of lithology on gradient may be related to clast size via two processes. First, with the assumption that median particle sizes delivered from the local hillslopes scales with $D_{max}$, the reaches may be adjusted to the caliber of the local sediment supply because incision cannot proceed until the bed is cleared of sediment (e.g., Lenzi et al., 2006). For example, at km-20 on the NF Feather River, the steep first kilometer of the reach underlain by monzotonalite may be a response to large boulders being delivered from the quartz diorite reach immediately upstream (Fig. 2B). In contrast, the second reach

[Figure]

**Figure 10. Rose diagram of azimuths of ~5-km reaches of river in the northern Sierra Nevada. Channel reaches flowing through metamorphic rocks in the foothills are aligned parallel to the structural grain and the range crest (i.e., approximately NNW-SSE). Reaches flowing across plutonic rocks are generally oriented down the slope of the range.**

*Lithological and structural controls on Sierran rivers*

of monzotonalite (km-30–38) is not preceded by a bedrock unit delivering large clasts and lacks a steep initial section. This observation suggests that the caliber of the sediment delivered by a particular lithology may be affecting the gradients of local reaches, as well as those farther downstream (Brocard and van der Beek, 2006; Duvall et al., 2004; Finnegan et al., 2017; Hack, 1957; Hack, 1973). Further evidence for the role of sediment supply on channel steepness may be seen at km-68 along the NF Mokelumne River (Fig. 11) where a very steep reach in granitic rock (8.9%) transitions to a reach in non-quartzite metasedimentary rock which is initially also steep (3.0%) but then grades into a much gentler slope (0.7%). The cliffs above the steep granitic reach are composed of exfoliation sheets which have few joints and, thus, break into large clasts as they fall into the river below. Some of these granitic boulders (albeit not the largest ones) are then likely transported downstream where they armor the relatively weak metasedimentary bedrock. Moreover, the failure of unjointed exfoliation sheets provides a strong negative feedback to river incision at this spot (Shobe et al., 2016). As the channel cuts down, it removes the toe of the exposed sheet and triggers a slope failure which delivers 10–20 m boulders to the riverbed. At this point, incision is likely inhibited until these boulders are sufficiently comminuted that they can be cleared from the bed by fluvial transport processes; this process is likely so slow that the knickpoint is essentially stationary.

Whereas the role of local sediment supply appears to contribute to the lithological controls on reach gradient, this may only be a partial explanation. Indeed, many of the lithological contacts are associated with sharp breaks-in-slope. If the transport of material delivered from upstream reaches was consistently limiting local incision, then the breaks-in-slope at the lithological transitions would be more diffuse (e.g., km-68 on the NF Mokelumne River). Another process, then, that might link lithology, $D_{max}$, and gradient is incision of the bed by the plucking of individual blocks. Because of differences in stress history and strength, different rock units will have different joint densities (Hancock and Engelder, 1989). Although joint spacing was not measured in this study, $D_{max}$ is assumed here to be a proxy for the maximum distance between joints (Hack, 1957), and the slope of an incising channel should increase with joint spacing where plucking dominates (Chatanantavet and Parker, 2009; Lamb et al., 2015). Indeed, differences in joint spacing could explain the significant differences in erodibility and gradient among the plutonic rocks. The effect of joint density could also explain the low gradients of reaches pass-

[Figure]

Figure 11. $D_{max}$ (maximum particle size) along a section of the North Fork Mokelumne River. The steep knickpoint in granite is associated with an abrupt increase in boulder size, a consequence of exfoliation sheets with low joint density lining the canyon walls. Large clasts delivered from these granitic slopes appear to be transported downstream to the reach in metasedimentary rock, resulting in an initially steep gradient in a bedrock unit that typically only supports gentle gradients and has $D_{max}$ values in the 2–4 m range. Gap in $D_{max}$ at ~60 km due to poor quality of the Google Earth imagery. Mzg—granitic; Pzcc—metasedimentary.

ing through fault zones where the fracturing of bedrock by tectonic activity may break the rock into pieces sufficiently small to be easily eroded by plucking (Molnar et al., 2007).

Whipple et al. (2000) estimated that incision by plucking in the Sierra Nevada was at least 10× faster than by abrasion, and a limited set observations in this study suggest that, indeed, abrasion does not appear to be a dominant process. Finer textures and higher quartz contents increase the tensile strength of crystalline rocks (Tuğrul and Zarif, 1999) and, thus, their resistance to abrasion (Sklar and Dietrich, 2001). Therefore, because the rate of abrasion increases with channel gradient (Johnson and Whipple, 2007; Lamb et al., 2008; Wohl and Ikeda, 1997), the slopes of individual reaches through plutonic rocks with different mineralogies should vary if abrasion is an important process. However, on the NF Feather River, alternating sequences of plutonic rocks with varying levels of quartz content (e.g., quartz diorite versus pyroxene diorite) or grain sizes (e.g., fine- to medium-grained pyroxene diorite versus coarse-grained monzotonalite) do not lead to consistent differences in reach gradient (Fig. 2). Low abrasion rates may be related to the high tensile strengths of plutonic, metavolcanic, and quartzite bedrock that dominate the northern Sierra Nevada (Sklar and Dietrich, 2001). In addition, the presence of Eocene–Oligocene deposits throughout the region (Lindgren, 1911) attest to slow rates of erosion that might not yield a sufficient quantity of material to efficiently abrade the bed.

Finally, the role of time is important in the relationship between channel slope and lithology. Jansen et al. (2010) concluded that lithological controls on topography are more pronounced in post-orogenic landscapes because differential erosion has had more time to erode the weaker rocks and leave the stronger rocks behind; thus, differential erosion will amplify the topographic expression of variations in rock strength. For example, at the lower end of the NF Feather River profile (Fig. 2), a 100-m-high step has developed where a reach across a fault zone, which presumably weakened the rock, appears to have eroded faster than its upstream neighbors.

**Drainage Patterns and Lithology**

The reach-azimuth data (Fig. 10) and observations from the large-scale regional geological maps (Saucedo and Wagner, 1992; Strand, 1967; Wagner et al., 1991; Wagner et al., 1981) demonstrate that lithology has an important control on network structure in the northern Sierra Nevada. At the lower elevations, where the rivers are passing through the metamorphic belts, long reaches parallel to the structural grain are cut into weaker rock. Indeed, the strong bias for longitudinal valleys in the Sierra Nevada foothills misled early miners into concluding that gold-bearing gravels had been deposited by rivers with outlets to the north (Bateman and Wahrhaftig, 1966). Ninety-degree junction angles between trunk streams and their tributaries are also evidence for structural control on the spatial organization of the river networks (Howard, 1976).

*E.J. Gabet*

In contrast, in the upper reaches of the watersheds where plutonic rocks dominate, the rivers are organized into dendritic networks. In the absence of any large-scale structural control, the rivers have a strong affinity for being oriented down the path of steepest descent (Fig. 10). Note, however, that these observations do not imply a complete lack of structural control on channels in the plutonic rocks. For example, the Merced River follows a large joint as it debouches from Yosemite Valley and, at the km-scale, there is a strong association between the orientations of fractures and streams (Ericson et al., 2005). Nevertheless, at the watershed-scale, there is little evidence for structural control on network configurations in the plutonic bedrock.

**Implications for Interpreting the Geologic History of the Sierra Nevada**

The age of the Sierra Nevada has been a contentious issue (e.g., Cassel et al., 2009; Gabet, 2014; Wakabayashi, 2013). While paleoelevation studies using stable isotopes have provided evidence that the Sierra Nevada has been a high range for much of the Cenozoic (Cassel et al., 2009; Crowley et al., 2008; Hren et al., 2010; Mix et al., 2016; Mulch et al., 2006; Poage and Chamberlain, 2002), others have relied on geomorphic analyses to argue that the range is much younger, having risen to its present heights only during the late Cenozoic (e.g., Jones et al., 2004; Lindgren, 1911; Wakabayashi, 2013; Yeend, 1974). The primary line of evidence used to support recent uplift is the difference in gradients of the remnants of Eocene and Oligocene paleochannels found throughout the northern half of the range. Lindgren (1911) linked these isolated deposits together to attempt to restore the course of the ancient rivers and noted that the reconstructed segments that flowed normal to the rangecrest (A and C in Fig. 12) were steeper than those flowing parallel to the rangecrest (B in Fig. 12). Lindgren concluded that the 0.7° difference in slopes was due to post-depositional westward-tilting and uplift of the northern Sierra. Since then, others have undertaken similar analyses of Lindgren's reconstructed channels and have also concluded that the Sierran block has been tilted by 0.7–1.0° in the late Cenozoic, resulting in significant uplift at the crest (Hudson, 1955; Jones et al., 2004; Yeend, 1974).

The validity of these analyses rest on several assumptions. First, they must assume that Lindgren's hypothetical reconstructions were accurate. However, some of Lindgren's paleochannels defy gravity and imply that water can flow uphill over high ridges (Gabet, 2014). This problem is highlighted in Hudson (1955), which applied a trigonometric analysis to "untilt" different sections of Lindgren's Tertiary Yuba River (Fig. 13) according to their azimuth to estimate the amount of late Cenozoic uplift. However, Hudson conceded that this approach "often produce(s) absurd results, such as negative gradients." Furthermore, Cassel et al. (2012b) and Durrell (1966) disproved two of Lindgren's paleochannel reconstructions by demonstrating that he had connected gravel patches that were receiving sediment from different source areas and, thus, could not have been part of the same river system. Second, studies that rely on Lindgren's paleochannels to investigate post-depositional tilt of the range must assume that these reconstructions represent contemporaneous rivers that were in steady-state. Cassel et al.

[Figure]

**Figure 12. Hypothesis, proposed by others (Hudson, 1955; Jones et al., 2004; Lindgren, 1892; Yeend, 1974), explaining how the different gradients of paleochannel remnants support recent uplift of the Sierra Nevada. Map shows Lindgren's (1911) reconstruction of the ancient South Yuba River (solid line) superimposed over modern rivers (dotted lines). In the lower illustrations, dashed lines are the abandoned paleochannel remnants; arrow width represents relative gradients. At Time 1, the gradients of the transverse paleochannels (A, C) and the longitudinal reach (B) are similar. Hypothesized tilting and uplift (Time 2) increase the gradients of A and C but left the gradient of B unchanged.**

(2012a) and Cassel et al. (2012b) concluded, however, that gravel deposition progressed eastward up the range and that this general trend of aggradation was interrupted by periods of incision such that the isolated paleochannel remnants represent different generations of rivers. Finally, these studies assume that the Tertiary channels originally had smooth profiles such that any steps in the reconstructed profiles must be due to tilting (after accounting for reach orientation). This assumption, therefore, dismisses the role of lithology and structure in controlling channel gradient.

The analysis of the profiles of the modern rivers presented here can shed light on the relationship between gradient and orientation that has been used as evidence for recent tilting of the northern Sierra Nevada. For example, the range-parallel, low gradient segment in Lindgren's reconstruction of the ancient South Yuba River (B in Figs. 12 and 13) would have flowed across weak bedrock and a fault zone, conditions that yield low gradients in the modern South Yuba River. In contrast, the steep, range-normal segments in the putative ancient channel (A, C in Figs. 12 and 13) are in rock types that yield steep gradients in the modern rivers (i.e., plutonic, metavolcanic, and quartzite bedrocks). Assigning slopes for each lithological unit in segments A and C based on measurements from the modern rivers yields a difference in gradient of 1.3–1.9% (0.7–1.1°) between the range-parallel and range-normal segments (Fig. 13). In other words, lithology can account for 0.7–1.1° of the difference in the slopes of these segments whereas previous studies attributed this difference to uplift and tilting (Hudson, 1955; Jones et al., 2004; Lindgren, 1911; Yeend, 1974). Therefore, even if the first two assumptions were plausible, the dominant influence of lithology on river gradients cannot be dismissed when using the slopes of the paleochannels to infer uplift.

**Lithological versus Tectonic Knickpoints**

Several studies have concluded that alleged significant tilting of the Sierra Nevada in the late Cenozoic created large knickpoints that quickly migrated up through the drainage networks and cut deep canyons (e.g., Clark et al., 2005; Figueroa and Knott, 2010; Matthes, 1930). For example, Wakabayashi (2013) proposed that an uplift-generated knickpoint swept up the American River system 3 m.y., incising to depths of 900 m. Similarly, Clark et al. (2005) analyzed knickpoints in tributaries to the Kern River, in the southern Sierra Nevada, and concluded that it had begun incising its canyon 3.5 m.y. as a result of contemporaneous uplift. Volcanic

*Lithological and structural controls on Sierran rivers*

[Figure]

**Figure 13. Profile of Lindgren's (1911) Tertiary Yuba River using data from Hudson (1955). The reconstructed profile, presented as a continuous feature, is based on isolated paleochannel remnants. Segments A, B, and C correspond to Figure 12; their gradients were calculated three different ways. The first value was calculated using each segment's distance as measured from the reconstructed profile and the second was calculated using each segment's "straight-line" distance to represent the regional slope; previous studies have attributed the differences in slopes between segments to tilting. The third value (in bold) is the distance-weighted average of the gradients of each lithological unit as determined from the modern South Yuba and South Fork American rivers (i.e., it is the slope of the segment that accounts for lithology). Since segment B would not have been tilted, no adjustment for lithology is needed for its gradient. Note, the similarities in the slopes of the reconstructed channel and the slopes adjusted for lithology. Figures 3 and 4 present the bedrock unit descriptions and slopes. Fill patterns represent the distribution of bedrock units at the surface and not the underlying structure. Lithology in reaches with (\*) inferred from surrounding outcrops because Cenozoic volcanic deposits cover the basement rock.**

rocks older than the alleged uplift deep within the American and Kern canyons, however, refute these two studies (Gabet, 2014) and highlight the importance of distinguishing migrating knickpoints generated by uplift from those forming in situ as a result of differential erosion (e.g., Cyr et al., 2014; Larue, 2008).

Multiple lines of evidence can be used to distinguish between tectonic and lithological knickpoints. For example, because uplift is a regional event, associated knickpoints ought to be found in similar topographic positions in neighboring drainages (Crosby and Whipple, 2006). In the Sierra Nevada, however, knickpoints are essentially randomly distributed with respect to topographic position, and knickpoints on adjacent streams typically do not project to a common baselevel (Wahrhaftig, 1965). In addition, there are no strath terraces in the northern Sierra Nevada, another distinguishing characteristic

of migrating knickpoints, that could be used as evidence for the passage of knickpoints 100s of meters high (e.g., Crosby and Whipple, 2006). Because erosion rates in the northern Sierras are slow, as demonstrated by the presence of Eocene-age gravel deposits throughout the region (Lindgren, 1911), strath terraces formed 3 m.y. would have been preserved.

In addition to fluvial features, the surrounding topography can also be used to distinguish between tectonic and lithological knickpoints. A break-in-slope along the South Yuba River at km-25–35 might appear to be a migrating knickpoint sweeping up the river system (Fig. 3B); if this were the case, steep slopes and high relief would be expected immediately downstream of the feature because of the delay in hillslope response times (Hurst et al., 2013). Instead, mean angle and relief drop to their lowest values, suggesting that this knickpoint was formed through differential erosion. Observations of the surrounding topography are particularly salient in the cases of knickpoints contained within a single unit, where there are no obvious lithological contrasts and a tectonic explanation would seem likely. For example, the knickpoint at km-11 on the Merced River is in the middle of a plutonic unit and, therefore, could be interpreted as a migrating knickpoint (Fig. 6B). If it were a migrating knickpoint, steep slopes and high relief would be expected immediately downstream of the feature (Hurst et al., 2013); instead, again, mean angle and relief drop to their lowest values. Tellingly, relative to the reach immediately downstream, $D_{max}$ increases 3-fold at the knickpoint face, suggesting an abrupt change in joint density. Indeed, Google Earth imagery confirms that the valley walls adjacent to the knickpoint are composed of large granitic sheets with few joints. Therefore, the rapid increase in channel slope, accompanied by peaks in hillslope angle, valley relief, and boulder size, most likely reflects a dip in joint density, rather than a migrating knickpoint

**CONCLUSION**

The lithological heterogeneity of the northern Sierra Nevada provides an opportunity for exploring the lithological controls on river profiles and patterns. The channels studied here have steep reaches over lithologies generally considered to be resistant: plutonic, metavolcanic, and quartzite bedrock. Reaches with gentle gradients are generally found in weaker non-quartzite metasedimentary rock and across fault zones where rock damage is likely. Because of differences in erodibility, alternating sequences of weak and strong lithologies create river profiles with steps as high as ~600 m (e.g., the Merced and South Yuba rivers).

In addition to the river profiles, there is a strong lithological influence on the channel network patterns. In the metamorphic belts of the Sierra Nevada foothills, the river networks are trellis-like, with long reaches parallel to the structural grain. In the plutonic bedrock, the drainage networks are dendritic with most of the channel reaches oriented down the path of steepest descent.

Finally, the influence of rock type on river gradients and planforms demonstrate that caution should be used when using fluvial features to infer tectonic activity, particularly in lithologically complex terrain. In the northern Sierra Nevada, attempts to extract tilt information from paleochannels must account for the role of rock erodibility and structure in controlling channel steepness. Indeed, the different gradients of paleochannel remnants used to argue for

E.J. Gabet

late Cenozoic tilting and uplift of the northern Sierra Nevada can be wholly attributed to differences in lithology. Considering that the alleged tilt of paleochannels forms the foundation for the paradigm of late Cenozoic uplift of the northern Sierra Nevada and that the other lines of evidence are also disputed (Gabet, 2014), significant recent uplift of the region should be considered an unproven hypothesis.

**ACKNOWLEDGMENTS**

I am grateful to S. Mudd for his help in determining $\theta_{ref}$ and N. Finnegan and M. Lamb for their insights. I thank J. Jansen, E. Baynes, and an anonymous reviewer for their rigorous and stimulating comments.

**REFERENCES CITED**

Bateman, P.C., and Krauskopf, K.B., 1987, Geologic map of the El Portal Quadrangle, west-central Sierra Nevada, California: U.S. Geological Survey, Miscellaneous Field Studies Map 1998, scale 1:62,500, https://doi.org/10.3133/mf1998.

Bateman, P.C., and Wahrhaftig, C., 1966, Geology of the Sierra Nevada, in Bailey, E.H., ed., Geology of Northern California: California Division of Mines and Geology Bulletin 190, p. 107–172.

Brocard, G.Y., and van der Beek, P.A., 2006, Influence of incision rate, rock strength, and bedload supply on bedrock river gradients and valley-flat widths: Field-based evidence and calibrations from western Alpine rivers (southeast France), in Willet, S.D., Hovius, N., Brandon, M.T., and Fisher, D.M., eds., Penrose Conference Series: Tectonics, Climate, and Landscape Evolution: Geological Society of America Special Paper 398, p. 101–126, https://doi.org/10.1130/2006.2398(07).

Brocklehurst, S.H., and Whipple, K.X., 2002, Glacial erosion and relief production in the Eastern Sierra Nevada, California: Geomorphology, v. 42, p. 1–24, https://doi.org/10.1016/S0169-555X(01)00069-1.

Callahan, R.P., Ferrier, K.L., Dixon, J., Dosseto, A., Hahm, W.J., Jessup, B.S., Miller, S.N., Hunsaker, C.T., Johnson, D.W., Sklar, L.S., and Riebe, C.S., 2019, Arrested development: Erosional equilibrium in the southern Sierra Nevada, California, maintained by feedbacks between channel incision and hillslope sediment production: Geological Society of America Bulletin, v. 131, no. 7-8, https://doi.org/10.1130/B35006.1.

Cassel, E.J., Graham, S.A., and Chamberlain, C.P., 2009, Cenozoic tectonic and topographic evolution of the northern Sierra Nevada, California, through stable isotope paleoaltimetry in volcanic glass: Geology, v. 37, no. 6, p. 547–550, https://doi.org/10.1130/G25572A.1.

Cassel, E.J., Graham, S.A., Chamberlain, C.P., and Henry, C.D., 2012a, Early Cenozoic topography, morphology, and tectonics of the northern Sierra Nevada and western Basin and Range: Geosphere, v. 8, no. 2, p. 229–249, https://doi.org/10.1130/GES00671.1.

Cassel, E.J., Grove, M., and Graham, S.A., 2012b, Eocene drainage evolution and erosion of the Sierra Nevada batholith across northern California and Nevada: American Journal of Science, v. 312, p. 117–144, https://doi.org/10.2475/02.2012.03.

Chatanantavet, P., and Parker, G., 2009, Physically based modeling of bedrock incision by abrasion, plucking, and macroabrasion: Journal of Geophysical Research. Earth Surface, v. 114, no. F4, https://doi.org/10.1029/2008JF001044.

Clark, M.K., Maheo, G., Saleeby, J., and Farley, K.A., 2005, The non-equilibrium landscape of the southern Sierra Nevada, California: GSA Today, v. 15, no. 9, p. 4–10, https://doi.org/10.1130/1052-5173(2005)015[4:TNLOTS]2.0.CO;2.

Crosby, B.T., and Whipple, K.X., 2006, Knickpoint initiation and distribution within fluvial networks: 236 waterfalls in the Waipoa River, North Island, New Zealand: Geomorphology, v. 82, no. 1, p. 16–38, https://doi.org/10.1016/j.geomorph.2005.08.023.

Crowley, B.E., Koch, P.L., and Davis, E.B., 2008, Stable isotope constraints on the elevation history of the Sierra Nevada Mountains, California: Geological Society of America Bulletin, v. 120, p. 588–598, https://doi.org/10.1130/B26254.1.

Cyr, A.J., Granger, D.E., Olivetti, V., and Molin, P., 2014, Distinguishing between tectonic and lithologic controls on bedrock channel longitudinal profiles using cosmogenic 10Be erosion rates and channel steepness index: Geomorphology, v. 209, p. 27–38, https://doi.org/10.1016/j.geomorph.2013.12.010.

Dühnforth, M., Anderson, R.S., Ward, D., and Stock, G.M., 2010, Bedrock fracture control of glacial erosion processes and rates: Geology, v. 38, no. 5, p. 423–426, https://doi.org/10.1130/G30576.1.

Durrell, C., 1966, Tertiary and quaternary geology of the northern Sierra Nevada, in Bailey, E.H., ed., Geology of Northern California: California Division of Mines and Geology Bulletin 190, p. 185–197.

Duvall, A.R., Kirby, E., and Burbank, D.W., 2004, Tectonic and lithologic controls on bedrock channel profiles and processes in coastal California: Journal of Geophysical Research. Solid Earth, v. 109, no. F3, https://doi.org/10.1029/2003JF000086.

Ericson, K., Migon, P., and Olvmo, M., 2005, Fractures and drainage in the granite mountainous area: A study from Sierra Nevada, USA: Geomorphology, v. 64, no. 1, p. 97–116.

Fan, N., Chu, Z., Jiang, L., Hassan, M.A., Lamb, M.P., and Liu, X., 2018, Abrupt drainage basin reorganization following a Pleistocene river capture: Nature Communications, v. 9, no. 1, no. 3756, https://doi.org/10.1038/s41467-018-06238-6.

Figueroa, A.M., and Knott, J.R., 2010, Tectonic geomorphology of the southern Sierra Nevada Mountains (California): Evidence for uplift and basin formation: Geomorphology, v. 123, p. 34–45, https://doi.org/10.1016/j.geomorph.2010.06.009.

Finnegan, N.J., Klier, R.A., Johnstone, S., Pfeiffer, A.M., and Johnson, K., 2017, Field evidence for the control of grain size and sediment supply on steady-state bedrock river channel slopes in a tectonically active setting: Earth Surface Processes and Landforms, v. 42, p. 2338–2349, https://doi.org/10.1002/esp.4187.

Fisher, G.B., Amos, C.B., Bookhagen, B., Burbank, D.W., and Godard, V., 2012, Channel widths, landslides, faults, and beyond: The new world order of high-spatial resolution Google Earth imagery in the study of earth surface processes, in Whitmeyer, S.J., Bailey, J. E., DePaor, D.G., and Ornduff, T., eds., Google Earth and Virtual Visualizations in Geoscience Education and Research: Geological Society of America Special Paper 492, p. 1–22, https://doi.org/10.1130/2012.2492(01).

Flint, J.J., 1974, Stream gradient as a function of order, magnitude, and discharge: Water Resources Research, v. 10, no. 5, p. 969–973, https://doi.org/10.1029/WR010i005p00969.

Gabet, E.J., 2014, Late Cenozoic uplift of the Sierra Nevada, California? A critical analysis of the geomorphic evidence: American Journal of Science, v. 314, p. 1224–1257, https://doi.org/10.2475/08.2014.03.

Gillespie, A.R., and Clark, D.H., 2011, Glaciations of the Sierra Nevada, California, USA, in Ehlers, J., Gibbard, P. L., and Hughes, P. D., eds., Quaternary Glaciations: Extent and Chronology: A Closer Look: Amsterdam, The Netherlands, Elsevier, v. 15, p. 447–462, https://doi.org/10.1016/B978-0-444-53447-7.00034-9.

Hack, J.T., 1957, Studies of longitudinal profiles in Virginia and Maryland: U.S. Geological Survey Professional Paper 294-B, p. 45–97, https://doi.org/10.3133/pp294B.

Hack, J.T., 1973, Stream-profile analysis and stream-gradient index: Journal of Research of the U.S. Geological Survey, v. 1, no. 4, p. 421–429.

Hancock, P.L., and Englelder, T., 1989, Neotectonic joints: Geological Society of America Bulletin, v. 101, no. 10, p. 1197–1208, https://doi.org/10.1016/0016-7606(1989)101<1197:NJ>2.3.CO;2.

Hanks, T.C., and Webb, R.H., 2006, Effects of tributary debris on the longitudinal profile of the Colorado River in Grand Canyon: Journal of Geophysical Research. Earth Surface, v. 111, no. F2, https://doi.org/10.1029/2004JF000257.

Hietanen, A., 1973, Geology of the Pulga and Bucks Lake quadrangles, Butte and Plumas counties, California: U.S. Geological Survey Professional Paper 731, scale 1:48,000, https://doi.org/10.3133/pp731.

Howard, A.D., 1976, Drainage analysis in geological interpretation: A summation: The American Association of Petroleum Geologists Bulletin, v. 51, no. 11, p. 2246–2259.

Hren, M.T., Pagani, M., Erwin, D.M., and Brandon, M.T., 2010, Biomarker reconstruction of the early Eocene paleotopography and paleoclimate of the northern Sierra Nevada: Geology, v. 38, no. 1, p. 7–10, https://doi.org/10.1130/G30215.1.

Hudson, F.S., 1955, Measurement of the deformation of the Sierra Nevada, since middle Eocene: Geological Society of America Bulletin, v. 66, p. 835–870, https://doi.org/10.1130/0016-7606(1955)66[835:MOTDOT]2.0.CO;2.

Hurst, M.D., Mudd, S.M., Attal, M., and Hilley, G.E., 2013, Hillslopes record the growth and decay of landscapes: Science, v. 341, p. 868–871, https://doi.org/10.1126/science.1241791.

Inoue, K., 1992, Downstream change in grain size of river bed sediments and its geomorphological implications in the Kanto Plain, central Japan: Geographical Review of Japan, v. 65, no. 2, p. 75–89, https://doi.org/10.4157/grj1984b.65.75.

Jansen, J.D., Codilean, A.T., Bishop, P., and Hoey, T.B., 2010, Scale dependence of lithological control on topography: Bedrock channel geometry and catchment morphometry in western Scotland: The Journal of Geology, v. 118, no. 3, p. 223–246, https://doi.org/10.1086/651273.

Johnson, B.D., 2015, Lithologic controls on knickpoint formation in Sierra Nevada bedrock channels [M.S. thesis]: San Jose, California, USA, San José State University, 62 p., https://doi.org/10.31979/etd.36yj-4qzk.

Johnson, J.P.L., and Whipple, K.X., 2007, Feedbacks between erosion and sediment transport in experimental bedrock channels: Earth Surface Processes and Landforms, v. 32, no. 7, p. 1048–1062, https://doi.org/10.1002/esp.1471.

Jones, C.H., Farmer, G.L., and Unruh, J.R., 2004, Tectonics of Pliocene removal of lithosphere of the Sierra Nevada, California: Geological Society of America Bulletin, v. 116, no. 11-12, p. 1408–1422, https://doi.org/10.1130/B25397.1.

Lague, D., 2014, The stream power river incision model: Evidence, theory and beyond: Earth Surface Processes and Landforms, v. 39, no. 1, p. 38–61, https://doi.org/10.1002/esp.3462.

Lamb, M.P., Dietrich, W.E., and Sklar, L.S., 2008, A model for fluvial bedrock incision by impacting suspended and bed load sediment. Earth Surface, v. 113, no. F3, https://doi.org/10.1029/2007JF000915.

Lamb, M.P., Finnegan, N.J., Scheingross, J.S., and Sklar, L.S., 2015, New insights into the mechanics of fluvial bedrock erosion through flume experiments and theory: Geomorphology, v. 244, p. 33–55, https://doi.org/10.1016/j.geomorph.2015.03.003, (corrigendum: http://dx.doi.org/10.1016/j.geomorph.2018.05.028).

Larue, J.-P., 2008, Effects of tectonics and lithology on long profiles of 16 rivers of the south Central Massif border between the Aude and the Orb (France): Geomorphology, v. 93, p. 343–367, https://doi.org/10.1016/j.geomorph.2007.03.003.

Lecce, S.A., 1997, Nonlinear downstream changes in stream power on Wisconsin's Blue River: Annals of the Association of American Geographers, v. 87, no. 3, p. 471–486, https://doi.org/10.1111/1467-8306.00064.

Lenzi, M.A., Mao, L., and Comiti, F., 2006, When does bedload transport begin in steep boulder-bed streams?: Hydrological Processes, v. 20, no. 16, p. 3517–3533, https://doi.org/10.1002/hyp.6168.

Lin, R.-G., and St. John, P., 2017, From extreme drought to record rain: Why California's drought-to-deluge cycle is getting worse: Los Angeles, California, USA, Los

*Lithological and structural controls on Sierran rivers*

Angeles Times, https://www.latimes.com/local/lanow/la-me-record-rains-20170410-story.html.

Lindgren, W., 1892, Two Neocene rivers of California: Geological Society of America Bulletin, v. 4, p. 257–298, https://doi.org/10.1130/GSAB-4-257.

Lindgren, W., 1911, The Tertiary gravels of the Sierra Nevada of California: U.S. Geological Survey Professional Paper 73, 226 p., https://doi.org/10.3133/pp73.

Ludington, S., Moring, B.C., Miller, R.J., Flynn, K.S., Stone, P.A., and Bedford, D.R., 2005, Preliminary integrated databases for the United States-western states: California, Nevada, Arizona, and Washington: U.S. Geological Survey Open File Report 2005-1305, https://doi.org/10.3133/ofr20051305.

Matthes, F.E., 1930, Geologic history of the Yosemite Valley: U.S. Geological Survey Professional Paper 160, 137 p., https://doi.org/10.3133/pp160.

Mix, H.T., Ibarra, D.E., Mulch, A., Graham, S.A., and Chamberlain, C.P., 2016, A hot and high Eocene Sierra Nevada: Geological Society of America Bulletin, v. 128, p. 531–542, https://doi.org/10.1130/B31294.1.

Molnar, P., Anderson, R. S., and Anderson, S. P., 2007, Tectonics, fracturing of rock, and erosion: Journal of Geophysical Research. Earth Surface, v. 112, no. F3, https://doi.org/10.1029/2005JF000433.

Montgomery, D.R., and Gran, K.B., 2001, Downstream variations in the width of bedrock channels: Water Resources Research, v. 37, no. 6, p. 1841–1846, https://doi.org/10.1029/2000WR900393.

Mudd, S.M., Attal, M., Milodowski, D.T., Grieve, S.W., and Valters, D.A., 2014, A statistical framework to quantify spatial variation in channel gradients using the integral method of channel profile analysis: Journal of Geophysical Research. Earth Surface, v. 119, p. 138–152, https://doi.org/10.1002/2013JF002981.

Mudd, S.M., Clubb, F.J., Gailleton, B., and Hurst, M.D., 2018a, How concave are river channels?: Earth Surface Dynamics, v. 6, p. 505–523, https://doi.org/10.5194/esurf-6-505-2018.

Mudd, S.M., Clubb, F.J., Gailleton, B., Hurst, M.D., Milodowski, D.T., and Valters, D.A., 2018b, The LSDTopoTools Chi Mapping Package (Version 1.11): Zenodo, https://doi.org/10.5281/zenodo.1291889.

Mulch, A., Graham, S.A., and Chamberlain, C.P., 2006, Hydrogen isotopes in Eocene river gravels and paleo-elevation of the Sierra Nevada: Science, v. 313, p. 87–89, https://doi.org/10.1126/science.1125986.

National Agricultural Imagery Program (NAIP), 2018, NAIP Imagery: U.S. Department of Agriculture Farm Service Agency, https://www.fsa.usda.gov/programs-and-services/aerial-photography/imagery-programs/naip-imagery/.

O'Callaghan, J.F., and Mark, D.M., 1984, The extraction of drainage networks from digital elevation data: Computer Vision Graphics and Image Processing, v. 28, p. 323–344, https://doi.org/10.1016/S0734-189X(84)80011-0.

Parrish, J.G., 2006, Simplified geological map of California: California Geological Survey, scale 1:2,250,000, sheet 57.

Pavano, F., Pazzaglia, F.J., and Catalano, S., 2016, Knickpoints as geomorphic markers of active tectonics: A case study from northeastern Sicily (southern Italy): Lithosphere, v. 8, no. 6, p. 633–648, https://doi.org/10.1130/L577.1.

Peck, D.L., 2002, Geologic map of the Yosemite quadrangle, central Sierra Nevada, California: U.S. Geological Survey, IMAP 2751, scale 1:62,500, 1 sheet, https://doi.org/10.3133/i2751.

Perron, J.T., and Royden, L., 2013, An integral approach to bedrock river profile analysis: Earth Surface Processes and Landforms, v. 38, no. 6, p. 570–576, https://doi.org/10.1002/esp.3302.

Phillips, J.D., and Lutz, J.D., 2008, Profile convexities in bedrock and alluvial streams: Geomorphology, v. 102, p. 554–566, https://doi.org/10.1016/j.geomorph.2008.05.042.

Pike, A.S., Scatena, F.N., and Wohl, E.E., 2010, Lithological and fluvial controls on the geomorphology of tropical montane stream channels in Puerto Rico: Earth Surface Processes and Landforms, v. 35, p. 1402–1417, https://doi.org/10.1002/esp.1978.

Poage, M.A., and Chamberlain, C.P., 2002, Stable isotopic evidence for a Pre-Middle Miocene rain shadow in the western Basin and Range: Implications for the paleotopography of the Sierra Nevada: Tectonics, v. 21, no. 4, p. 16-1–16-10.

Saucedo, G.J., and Wagner, D.L., 1992, Geologic map of the Chico quadrangle, California: U.S. Geological Survey, California Division of Mines and Geology, Regional Geologic Map 7A, scale 1:250,000, https://ngmdb.usgs.gov/Prodesc/proddesc_63087.htm.

Shobe, C.M., Tucker, G.E., and Anderson, R.S., 2016, Hillslope-derived blocks retard river incision: Geophysical Research Letters, v. 43, no. 10, p. 5070–5078, https://doi.org/10.1002/2016GL069262.

Sklar, L.S., and Dietrich, W.E., 2001, Sediment and rock strength controls on river incision into bedrock: Geology, v. 29, no. 12, p. 1087–1090, https://doi.org/10.1130/0091-7613(2001)029<1087:SARSCO>2.0.CO;2.

Sklar, L.S., and Dietrich, W.E., 2006a, The role of sediment in controlling steady-state bedrock channel slope: Implications of the saltation-abrasion incision model: Geomorphology, v. 82, no. 1-2, p. 58–83, https://doi.org/10.1016/j.geomorph.2005.08.019.

Sklar, L.S., and Dietrich, W.E., 2006b, The role of sediment in controlling steady-state bedrock channel slope: Implications of the saltation-abrasion incision model: Geomorphology, v. 82, no. 1-2, p. 58–83, https://doi.org/10.1016/j.geomorph.2005.08.019.

Snow, C.A., and Scherer, H., 2006, Terranes of the western Sierra Nevada foothills metamorphic belt, California: A critical review: International Geology Review, v. 48, no. 1, p. 46–62, https://doi.org/10.2747/0020-6814.48.1.46.

Snyder, N.P., Whipple, K.X., Tucker, G.E., and Merritts, D.J., 2003, Importance of a stochastic distribution of floods and erosion thresholds in the bedrock river incision problem: Journal of Geophysical Research. Solid Earth, v. 108, no. B2, p. 1–14, https://doi.org/10.1029/2001JB001655.

Strand, R.G., 1967, Geologic map of California, Mariposa sheet: U.S. Geological Survey, California Division of Mines and Geology, scale 1:250,000, https://ngmdb.usgs.gov/Prodesc/proddesc_491.htm.

Tuğrul, A., and Zarif, I.H., 1999, The correlation of mineralogical and textural characteristics with engineering properties of selected granitic rocks from Turkey: Engineering Geology, v. 51, p. 303–317, https://doi.org/10.1016/S0013-7952(98)00071-4.

U.S. Army Corps of Engineers (USACE), 1994, Hydraulic Design of Flood Control Channels: Washington, D.C., U.S. Army Corps of Engineers, 183 p.

Wagner, D.L., Jennings, C.W., Bedrossian, T.L., and Bortugno, E.J., 1981, Geologic map of the Sacramento quadrangle, California: U.S. Geological Survey, California Division of Mines and Geology, Regional Geologic Map 1A, scale 1:250,000, https://ngmdb.usgs.gov/Prodesc/proddesc_520.htm.

Wagner, D.L., Bortugno, E.J., and McJunkin, R.D., 1991, Geological map of the San Francisco-San Jose quadrangle, California: U.S. Geological Survey, California Division of Mines and Geology, Regional Geologic Map 5A, scale 1:250,000, https://ngmdb.usgs.gov/Prodesc/proddesc_519.htm.

Wahrhaftig, C., 1965, Stepped topography of the southern Sierra Nevada, California: Geological Society of America Bulletin, v. 76, p. 1165–1190, https://doi.org/10.1130/0016-7606(1965)76[1165:STOTSS]2.0.CO;2.

Wakabayashi, J., 2013, Paleochannels, stream incision, erosion, topographic evolution, and alternative explanations of paleoaltimetry, Sierra Nevada, California: Geosphere, v. 9, no. 2, p. 191–215, https://doi.org/10.1130/GES00814.1.

Whipple, K.X., Hancock, G.S., and Anderson, R.S., 2000, River incision into bedrock: Mechanics and relative efficacy of plucking, abrasion, and cavitation: Geological Society of America Bulletin, v. 112, no. 3, p. 490–503, https://doi.org/10.1130/0016-7606(2000)112<490:RIIBMA>2.0.CO;2.

Wobus, C.W., Whipple, K.X., Kirby, E., Snyder, N., Johnson, J., Spyropolou, K., Crosby, B.T., and Sheehan, D., 2006, Tectonics from topography: Procedures, promises, and pitfalls: Geological Society of America Bulletin, v. 398, p. 55–74.

Wohl, E.E., and Ikeda, H., 1997, Experimental simulation of channel incision into a cohesive substrate at varying gradients: Geology, v. 25, no. 4, p. 295–298, https://doi.org/10.1130/0091-7613(1997)025<0295:ESOCII>2.3.CO;2.

Yeend, W. E., 1974, Gold-bearing gravel of the ancestral Yuba River, Sierra Nevada, California: U.S.G.S Professional Paper 772, p. 1–44.

Zimmer, P.D., and Gabet, E.J., 2018, Assessing glacial modification of bedrock valleys using a novel approach: Geomorphology, v. 318, p. 336–347, https://doi.org/10.1016/j.geomorph.2018.06.021.

SCIENCE EDITOR: ROB STRACHAN
ASSOCIATE EDITOR: JOHN JANSEN

MANUSCRIPT RECEIVED 31 AUGUST 2018
REVISED MANUSCRIPT RECEIVED 5 APRIL 2019
MANUSCRIPT ACCEPTED 28 MAY 2019

Printed in the USA

---

## Short Comment (SC5) · 27 Jun 2019

I've added another Eocene-Oligocene deposit elevation to the authors' S FK American River figure. This shows that, near the black arrow on the authors' figure, maximum incision depths in this area since the Eocene-Oligocene cannot exceed ~225 m. The elevation for this comes from mining tunnel data presented in Lindgren (1911) - it can be found on page 180. Note, these data suggest a possible avulsion event; however, the important point to recognize is that the S Fk American River was already within 225 m of its modern depth 40 Ma. Moreover, once these old and deep deposits are recognized, it is difficult to see how recent tilt information can be extracted from the inset graph.

[Figure]

**ESurfD**

Interactive
comment

[Figure]

[Figure]

**Fig. 1.**

---

## Short Comment (SC6) · 28 Jun 2019

Here's another comment regarding using the volcanic deposits as indicative of recent incision. I apologize for all of these short notes, but I find this Discussion very interesting and it is helping me guide my ideas on the evolution of the range. In my 2014 paper, I plotted local relief (calculated over a 5 km window) along a transect in the northern Sierra (see the first figure). You'll notice that relief increases gradually from the Central Valley, peaks where there is a band of resistant rock, dips slightly, and then becomes approximately constant. The patterns of incision shown in the two plots generated by the authors show this same trend. This similarity is not coincidental since the remnants of the volcanic rocks are predominantly found in the interfluves (the volcanic deposits on the valley walls having been mostly eroded away). It appears, then, that the plots

of incision are really plots of landscape relief. It should be reasonable to expect that relief increases gradually as one goes from the Central Valley into the range but that, because of rock strength limitations, relief reaches a maximum and then remains constant. Therefore, the pattern seen in the incision plots can be explained on the basis of how relief changes in a mountain range and there is no need to appeal to tilting.

[Figure]

Rubicon/Middle Fork American River

[Figure]

South Fork American River

[Figure]

**Fig. 1.** Fig 1

---

## Short Comment (SC7) · 28 Jun 2019

This is a brief comment explaining the important implications of the gravel deposit along the South Fork American River that I noted previously. The first figure shows the topographic position of these Eocene-Oligocene gravels relative to the valley floor and the younger volcanic deposits. As far as I can tell, there is only one way to explain this stratigraphic sequence, and I show this in the second figure. At the top of this second figure, I've drawn a sketch of the cross-section shown in the first figure: the Miocene volcanics cap the interfluve and the gravels are lower down, about 150 m from the valley floor. Below this sketch, I show the general sequence of events that created this cross-section. The canyon was cut before the Eocene and then was filled with the auriferous gravels. This was then followed by some incision into the gravel but then the

valley was filled once again, this time by a series of rhyolitic and andesitic eruptions (of course, there would have been incisional episodes in between the eruptions). After the end of these large eruptions in the Pliocene, nearly all of this fill has been eroded away; only the deposits on the low-gradient interfluves and some scattered deposits within the canyons remain. This, then, is a classic cut-and-fill stratigraphic sequence and highlights why the elevation of the lowest volcanic deposits cannot be used to calculate basement incision as the authors have done. Although this example comes from the South Fork American River, the general principal applies to the Middle Fork American/Rubicon as well (and, in fact, to all northern Sierran rivers).

[Figure]

The pin shows the location of the Eocene-Oligocene gravels. The pink is granitic bedrock
and the orange is Miocene volcanics.

**Fig. 1.** Fig 1

[Figure]

**Fig. 2.** Fig 2

---

## Referee Comment (RC2) · Anonymous Referee #2 · 10 Jul 2019

Beeson & McCoy present a set of numerical modeling experiments and topographic analyses of select rivers draining the western slopes of the Sierra Nevada aimed at identifying characteristic signatures of tilting in bedrock river morphology. The authors primarily use linked 1-D river profile models to demonstrate that rigid block tilting generates patterns of incision, channel steepness, and knickpoint type and distribution dissimilar from those generated by a step change in uniform rock uplift rate or rock erodibility or a major truncation of headwater drainage area. They suggest that the magnitude and time since tilting can be recovered from bedrock river morphology, including tilt magnitude in landscapes with heterogeneous lithology — based on modeling tests with mid-channel bands of more and less erodible bedrock. The authors also use a 2-D landscape evolution model to show that channels flowing in tilt direction have elevated gradients during transient adjustment and that tilting induces stream captures in the direction of tilt. Finally, the authors analyze the bedrock river profile morphology of select rivers draining the western Sierra Nevada, a landscape thought to have experienced late Cenozoic tilting, to show that the landscape indeed displays the signatures of forward tilting identified in this study.

As the authors nicely articulate, quantitative characterizations of the transient response of bedrock rivers to tilting are lacking, even though tilting is widely documented in mountain ranges. This study therefore addresses an important outstanding question in tectonic geomorphology. Notwithstanding the many simplifications and sources of uncertainty in the modeling and topographic analyses, I find that theses limitations are reasonably explained to the reader (though I point out a few places where I think more qualification would be useful below). The study therefore provides valuable insight and advancement (even if only an important first step) towards identifying decipherable topographic signatures of tilting. The paper is well written and the authors are quite thorough in their analyses, considering various tilt orientations and magnitudes and stream power incision slope exponents in their modeling experiments - in places, making the main text a bit cumbersome (I suggest a few places where I think the main text reference to supplementary material could be shortened).

The study certainly inspires a number of follow-up questions (What about different tilt histories and/or lithologic patterns? How do thresholds for river incision or sediment flux dynamics impact the topographic signatures of tilting?), but I think this is a good thing, demonstrating the unique contribution of the work to an interesting and outstanding research question. In my view, the authors have made judicious choices in what specific questions to address and what assumptions to make in their modeling, and I don't think the authors should significantly expand the scope of the work.

That said, my main critique of the manuscript is that the 2-D landscape evolution modeling and application to the Sierra Nevada feel a bit underdeveloped relative to the 1-D river profile modeling. These are both critical litmus tests for evaluating if and when

topography may record a decipherable signature of tilting. It's great that the authors included these components in the first place, but I think the manuscript would be significantly strengthened if one or both or these analyses dug a little deeper into the results and discussion of the modeling experiments and topographic analyses already performed.

In particular, for the landscape evolution modeling, I wonder about the frequency and rates of divide migration associated with both gradual divide migration and discrete capture events and the river network response times. How does discrete vs. gradual drainage reorganization impact channel morphology (and how does this relate back to the 1-D profile modeling scenarios of tilting and major truncation/beheading of mainstem rivers)? How does tributary junction angle modify the morphology of tributary knickzones or steepness patterns, and what affect does this have on river profile analyses? I also suspect that drainage reorganization in the landscape evolution model is sensitive to the model discretization and flow routing algorithm you use (as mentioned on pg. 22, line 5), and it would be nice to see some test/statement of how robust the results are to these factors. It feels a little like a missed opportunity to perform the 2-D landscape evolution modeling and not take a closer look at the diversity of river profile morphologies that develop in the evolving river network.

For the Sierra Nevada analyses, at pg. 25, Line 9, why do you chose to use a concavity theta = 0.45? Does this linearize and/or collapse mainstem and tributary rivers in the region on chi plots? How do the knickpoint travel times tau (and inferred timing of the tilting event) vary for n = 2/3 or n = 5/3? What about tilting associated with postglacial rebound? Granger & Stock (2004) measure tilt angles up to ~0.08 deg in the past ~15 kyr from cave deposits in the Southern Sierra Nevada near the San Joaquin and Kings Rivers. How would this secondary tilting event affect your results, considering even the case in which 90-95% of tilting occurred ~5 Ma and the remaining 5-10% occurred since the LGM? Is it reasonable to analyze channels above the Pleistocene glacial limit? To what extent does this topography retain its glacial form (are valleys

v-shaped above 1000 m?) and how, even downstream of the glacial limit, could Quaternary changes in discharge and sediment supply affect river profile morphology? Are there any knickpoints you can definitively associate with tilting rather than a transient response to deglaciation or heterogenous lithology? In Figure 11, if you exclude all tributaries above the glacial limit, does your tilt magnitude estimate change? With its glacial history and lithologic variability, the Sierra Nevada seems like a pretty complex test case to evaluate the real-world applicability of fluvial metrics of tilting identified in the modeling. The relative convergence of the results and consistency with previous interpretations is compelling, so perhaps the complexity proves the robustness and widespread potential for extracting tilt histories from river profile and network morphology. However, at present, I'm not sure the authors have fully addressed these confounding factors nor sufficiently quantified the sensitivity and associated uncertainty of the Sierra Nevada results to their assumptions and interpretations.

Other comments: (1) Pg. 2, Line 30 - What about documented response to tilting associated with dynamic topography? There are a number of papers (e.g., Braun et al. 2013, Ruetenik et al. 2016) that have addressed fluvial response to long wavelength dynamic uplift and subsidence, including continental-scale tilting. It seems like this should be mentioned or at least the distinction in underlying mechanism and/or the scale of tilting should be clarified.

(2) One major assumption underlying all model tests is the assumption of uniform concavity, which could very well be violated if the mechanisms of river incision change with channel slope or in different lithologies. There are multiple places (e.g, in presenting the first results in Figure 2 with vertical knickpoints and in presenting the results in Figure 4 with heterogeneous lithology) where this assumption should be restated and its limitation clarified. It's a little hard to tell in Figure 10 since lithology is not included in (b) (I suspect since what would ideally need to be displayed along the chi axis is integrated lithology), but it appears to me, looking at the elevation ranges, that river profile concavity could vary between the different lithologies.

(3) The assumption of a negligibly small/frequently exceeded threshold for bedrock river incision should be stated and the limitations of this assumption should be addressed, particularly since I imagine tilting could have a significantly enhanced impact of bedrock river incision if it increased channel gradients enough to change the frequency and magnitude of events exceeding a threshold for bedrock river incision.

(4) Pg. 9, Lines 10-14 and Line 33; Pg. 11, Lines 19-21; Pg. 13, Lines 7-10 - It's great that the authors performed these sensitivity tests for the slope exponent. I find the references to these tests at the end of each result subsection a bit disruptive to my understanding of the main results, particularly since many results in the n=2/3 and n=5/3 cases are similar to the n=1 case, per the authors' own assessment. I'd prefer to see these all aggregated into a supplementary material section with a single main text reference.

(5) Pg. 11, Line 20 - Should be noted that such 1-D modeling neglects area-loss feedback and full 2-D/network dynamics of drainage reorganization

(6) In the 1-D profile modeling, I suspect some of your quantitative results (e.g., estimated tilt from geometry of local rock-type related knickzones) may be somewhat sensitive to your forward differencing scheme and discretization. I suggest clarifying which forward differencing scheme you use. In the same vein, as mentioned above, in the 2-D LEM, the results are likely sensitive to the flow routing algorithm, which is not specified in the text.

(7) Pg. 17, Line 30 - The heterogeneous lithology tested is quite idealized, and I wonder if this is a really a reasonable proof-of-concept test in considering if one could apply this to a real landscape. What happens if you add noise (even much lower magnitude than the heterogeneous lithology K contrast) to your spatially variable K, as you might expect if there's spatial heterogeneity in rock erodibility even within the same lithology? Is it still possible to recover tilt timing and/or magnitude? Also, many landscapes where tilting is suspected to have modified the topography are landscapes with tilted sedimentary

rock units. Tilting of such tilted strata would effectively result in spatially and temporally variable erodibility within bedrock channels, as rivers incise vertically into a tilted layer, causing the exposure of that layer to migrate up/down stream depending on the river orientation and tilt magnitude and direction. It seems like this at least warrants mention.

(8) Pg. 19, Line 5 - Specify what type of curvature

(9) Figure 10 - I don't think the rock types should be extended all the way down to the x-axis, since this seems to imply that the rock units have vertical dips extending to depth. I suggest fading away the lithology shading just below the river profile.

(10) Pg. 27, Line 5 - Would be helpful, for reproducibility, to specify the threshold you use to determine if tributary knickzones collapse with the mainstem on the chi plot or not. Here and elsewhere, ditto for how exactly you define a knickzone (e.g., what threshold change in steepness do you use to identify the top and bottom?)

References: Braun, J., Robert, X., & Simon‐Labric, T. (2013). Eroding dynamic topography. Geophysical Research Letters, 40(8), 1494-1499.

Granger, D. E., & Stock, G. M. (2004). Using cave deposits as geologic tiltmeters: Application to postglacial rebound of the Sierra Nevada, California. Geophysical Research Letters, 31(22).

Ruetenik, G. A., Moucha, R., & Hoke, G. D. (2016). Landscape response to changes in dynamic topography. Terra Nova, 28(4), 289-296.

---

## Short Comment (SC8) · 10 Jul 2019

**Northern Sierra Nevada: Canyon Incision**

To calculate their tilt estimate, the authors have assumed that the canyons in the northern Sierra Nevada were deepened beneath the lowest level of the volcanic rocks only in the past 5 Ma. In other words, they assume that the bedrock river bed was at the same elevation as the lowest volcanic outcrop 5 Ma and, since then, has incised through bedrock down to its present elevation. This assumption is critical to their tilt estimate, as represented in the inset to their Figure 10a:

[Figure]

However, as shown in the figure below (right), much older sediment can be found deep within these canyons.

[Figure]

[Figure]

This is a view of the South Fork American River. The Miocene volcanic Mehrten Formation is shown in orange; the small patch of Eocene – early Oligocene auriferous gravels are shown with the yellow pin on the map (the small tan oval); the granitic bedrock is pink. To the left, I show the stratigraphic relationship of these sediments. This means that the fundamental assumption used by the authors for their tilt estimate is unequivocally refuted. The elevations of the volcanic rocks do not represent late Cenozoic bedrock channel elevations and, therefore, the points shown in the inset of Figure 10a are not bedrock incision depths; they are simply reflecting the fact that

valley relief increases as you go into the mountains. The canyon's present topography and its distribution of sediments is explained by the sketch below.

[Figure]

The authors have assumed, however, that all the bedrock below the Late Miocene volcanics in the first sketch (ie. "Present S Fork") was incised since 5 Ma. Here is a cartoon illustrating their conclusion regarding the timing of canyon incision.

[Figure]

Clearly, significant post-Miocene bedrock incision is contradicted by the presence of the Eocene gravels near the bottom of the canyon and this represents a fundamental problem in their analyses. (I should emphasize that nothing that I'm presenting here is new: these stratigraphic relationships and Eocene deposits have been known since at least 1880.) This problem impairs two of their approaches regarding uplift of the northern Sierras. In Sections 4.1.1 and 6.1, tributary knickzones are used to estimate the timing of tilt; since these large knickzones date back to at least the Eocene, they cannot provide much information regarding recent uplift. In Section 4.4.2 and 6.2, tributary knickzone drop heights and incision depths are used to estimate tilt magnitude but, these incision depths are a product of a much older period of incision and won't provide information regarding recent incision. Since the Eocene, there has been only a maximum of 100-200 m of net bedrock channel incision, all of which can be attributed to the response to uplift during the Mesozoic. As I mentioned elsewhere, the northern Sierra was buried by gravel and volcanic rocks during much of the Cenozoic and the rivers have only recently have had access again to their bedrock beds.

Northern Sierra Nevada: Model
I've had a bit more time to examine the model and its assumptions and there are a few issues here as well.

1) One of these is the assumption of uniform uplift superimposed on to uplift by tilting. Uniform uplift would leave an obvious scarp at the range-front, but no scarp like this has been mapped nor has anyone suggested that uniform uplift has occurred during the Cenozoic. Therefore, there doesn't seem to be a strong basis for comparing the model results (Fig. 3b) to an actual profile (Fig. 10b) in Section 6.1. My concern is that uniform uplift is being added to make the model work right (I've written many models so I understand how this happens), but this boundary condition may be affecting the results in some significant way.

2) In Section 6.1, knickzone geometry is used to estimate tilt; however, this geometry will be sensitive to rock erodibility (K). The authors did not explain how they *independently* determined the erodibility of these rocks. It is important that K be determined without appealing to assumptions regarding uplift, incision, etc (eg, Eqn 5) because then things may get a bit circular.

3) In addition to assuming that rock erodibility is uniform (which I discussed in an earlier comment), the streampower formulation used here assumes that rainfall is uniform as well. However, this assumption is violated by the strong orographic effect whereby annual precipitation at the Sierran crest is 4-5 times greater than in the foothills (see map below). Because of the nature of storm tracks in the region, this precipitation gradient has existed since at least the Eocene (Chamberlain et al., 2012). As shown by several published papers (eg, Roe et al, 2003), properly accounting for precipitation gradient is necessary for investigating the spatial and temporal distribution of channel slopes and this could be easily done by rewriting Hack's Law such that drainage area is a function of both distance and elevation.

[Figure]

**Average Annual Precipitation**

**California**

Copyright 2000 by Spatial Climate Analysis Service, Oregon State University

Legend (in inches)

| | | | |
|---|---|---|---|
| Under 5 | | 30 to 40 | |
| 5 to 10 | | 40 to 60 | |
| 10 to 15 | | 60 to 80 | |
| 15 to 20 | | 80 to 120 | |
| 20 to 30 | | Above 120 | |

Chamberlain, C. P., Mix, H. T., Mulch, A., Hren, M. T., Kent-Corson, M. L., Davis, S. J., Horton, T. W., and Graham, S. A., 2012, The Cenozoic climatic and topographic evolution of the western North American Cordillera: American Journal of Science, v. 312, no. 2, p. 213-262.

---

## Author Comment (AC4) · 10 Jul 2019

We have updated our figure to show the locations of auriferous gravel deposits as mapped on the 1:250,000 quad maps (geologic maps at the 1:125k scale are not available in digital form) and included an analogous plot from the North Fork American to help clarify some points discussed below. I was unable to locate any data on the Spring Valley/Soldier Creek auriferous gravel deposit you mention, and thus could not verify its age or elevation. I have plotted its approximate location given that you said it was 250 m below the nearest volcanic deposit. It is unclear which elevation you chose from Lindgren's data, but all of the tunnels appear to be south and east of Placerville. If they are under the volcanics, the elevation can't be much lower than the volcanics and adjacent auriferous gravels we have plotted now. Note that in the original figure, we used

a 10 km wide swath profile centered on the mainstem to pull the range of elevations of the volcanics within the swath and project these to the location of the mainstem, but in these figures we used a 15 km wide swath in order to capture the auriferous gravel deposits that you describe near Placerville.

The addition of the auriferous gravel deposits does not change our results as they are most commonly found directly subjacent to the volcanics. Deposits upstream of the mainstem slope-break knickpoint that are near to river elevation and their associated low incision values do not refute the idea that incision downstream of the knickpoint reflects surface uplift in the form of a punctuated tilting event. Rather, these low deposits upstream of the knickpoint are consistent with a transient response to a punctuated tilting event, which we show can generate highly variable incision depths upstream of the slope-break knickpoint. Note that the deposits in the South Fork American River basin near Placerville that are close to river elevation plot near the top of the first knickzone in granite and that the Spring Valley deposit plots at the top of the second knickzone in granite (a knickzone that is consistent with the western metamorphic belt acting as a band of more erodible rock). Based on our model we would expect these locations to exhibit low incision values. A similar trend is observed in the North Fork American, in which low incision values occur in a reach that is just upstream of the slope-break knickpoint and just downstream of a deeply incised section that corresponds with the western metamorphic belt.

*So, my point is that the distribution of the volcanic rocks does not define a pre-incision paleosurface and, therefore, cannot provide information on incision depths.*

We agree that the auriferous gravels would be a superior pre-incision paleosurface than the Cenozoic volcanics in that they define river valleys, whereas the volcanics covered the entire landscape and thus the basal contact inherited the paleo relief of the buried landscape. Unfortunately, the auriferous gravels are sparse and thus it is difficult to establish longitudinal trends using them. As Wakabayashi (2013) and Cassell et al., (2011) have described, the gravel deposits are usually thin (order 10s of meters

with the thickest deposits described by Cassell et al., (2011) approximately 140 m) and, as is clear in our plots, they are commonly directly subjacent to the volcanics. Therefore, it seems reasonable to assume that the bottom of the Cenozoic volcanic deposits approximates a pre-incision surface even where gravels do not exist. We interpret local deviations from these longitudinal trends caused by the gravels being significantly (order 100 m) below nearby volcanics as the effects of paleo relief (i.e., volcanics on a paleo interfluve would be above the gravels in a paleo valley) or due to faulting after gravel deposition, which Lindgren showed could be 10-100 m.

*Finally, my GSAB paper showing that lithology has a first order control on channel steepness is now in press. I've attached the proofs as a supplement.*

The variability in steepness that you document in your attached paper is consistent with our interpretation of the Sierra as being in a transient state in response to a punctuated tilting event, with the tilting event heavily modulated by lithology. Our interpretation is that most of the metamorphics are more erodible and have thus responded faster, resulting in more consistent and lower channel steepness. In contrast, granodiorite is less erodible and thus all knickzones related to tilt (both the mainstem slope-break knickpoint and the rock-type slope-break knickpoints) are propagating at a slower speed, resulting in the high variability in steepness in streams in granite that you document and that is shown in our profiles. This can explain why the steep reaches are not randomly distributed across all lithologies, rather than dismissing the idea of actively migrating knickpoints.

**North Fork American**

R$^2$ = 0.92
$\theta$ = 0.8
1261

700 m

Incision (m)
Elevation (m)
Eucl. dist. from MF (km)

▲ Mainstem slope-break knickpoint
●— Auriferous gravel deposit
●— Cenozoic volcanic deposit

**Middle Fork American / Rubicon**

R$^2$ = 0.92
$\theta$ = 0.8
1184

Incision (m)
Elevation (m)
Eucl. dist. from MF (km)

**South Fork American**

R$^2$ = 0.9
$\theta$ = 0.6
949

Deposits near to and
east of Placerville

Gabet's Spring Valley / Soldier Creek deposit?

Incision (m)
Elevation (m)
Eucl. dist. from MF (km)

Distance upstream from mountain front (km)

**Fig. 1.** American River longitudinal profiles with surface geology, Cenozoic volcanics and Eocene auriferous gravels. Insets show depth of canyon in basement below volcanics and gravels.

---

## Author Comment (AC5) · 10 Jul 2019

*It appears, then, that the plots of incision are really plots of landscape relief.*

Yes, our measure of incision is similar to that of relief, only our measure is closer to the relief of steep tributaries draining canyon walls in that it is measured in a 15 km wide swath centered along the mainstem river.

*It should be reasonable to expect that relief increases gradually as one goes from the Central Valley into the range but that, because of rock strength limitations, relief reaches a maximum and then remains constant. Therefore, the pattern seen in the incision plots can be explained on the basis of how relief changes in a mountain range and there is no need to appeal to tilting.*

We agree that, in a steady-state landscape or one that is strength limited, relief is likely to increase moving inward of the mountain front and be relatively uniform within the range. However, nowhere in the Sierra is relief uniform. In contrast, relief is variable both within and among basins. For example, in the American River, the South Fork reaches a maximum relief of 700 m in its upper reaches about 70 km from the mountain front, the Middle Fork reaches 700 m of relief not far from the outlet at about 40 km from the mountain front, and the North Fork reaches 1,100 m of relief again at 70 km from the mountain front. If one looks to southern Sierra the same rock types support relief of up to 2000 m. Furthermore, in steady-state landscapes, relief should be greater in more resistant rock. In contrast to this, some of the least incised reaches in the Sierra are those that flow through granodiorite, as illustrated in all forks of the American.

**North Fork American**

Incision (m): $R^2 = 0.92$, $\theta = 0.8$, 1261
Eucl. dist. from MF (km)

700 m

**Middle Fork American / Rubicon**

Incision (m): $R^2 = 0.92$, $\theta = 0.8$, 1184
Eucl. dist. from MF (km)

**South Fork American**

Incision (m): $R^2 = 0.9$, $\theta = 0.6$, 949
Eucl. dist. from MF (km)

Deposits near to and
east of Placerville

Gabet's Spring Valley / Soldier Creek deposit?

Distance upstream from mountain front (km)

▲ Mainstem slope-break knickpoint
●■ Auriferous gravel deposit
●■ Cenozoic volcanic deposit

**Fig. 1.** American River longitudinal profiles with surface geology, Cenozoic volcanics and Eocene auriferous gravels. Insets show depth of canyon in basement below volcanics and gravels.

---

## Author Comment (AC6) · 10 Jul 2019

We agree that the stratigraphic record of valley cut-and-fill (in which valleys were filled with gravels and then evacuated) could look identical to that of bedrock incision in response to recent tilt (in which incision below gravels is interpreted as a response to surface uplift) and thus based only on the stratigraphic record it is difficult to discriminate the two possible histories. In the valley cut-and-fill interpretation, gravel deposits must have been at least as thick as their current height above the valley bottom. However, taking the upper North Fork American as an example, this implies that the gravels were at least 700 m thick, filling the canyon beneath the minimum elevation of deposits preserved on the canyon edge. Given that the maximum measured thickness of auriferous gravel deposits is 140 m (Cassell et al., 2011), this scenario seems less likely than

late Cenozoic tilting that does not require deposits thicker than observed elsewhere in the range.

[Figure]

**Fig. 1.** American River longitudinal profiles with surface geology, Cenozoic volcanics and Eocene auriferous gravels. Insets show depth of canyon in basement below volcanics and gravels.

---

## Editor Comment (EC1) · Jens Turowski (Editor) · 16 Jul 2019

Dear authors,

we have received two solicited reviews of the paper and a host of unsolicited comments (making the manuscript, in fact, the one that has been most commented on in the history of ESurf!). The solicited referees agree largely in their assessment: the paper is interesting, well written and deals with an important topic. The modelling approach is appropriate and the theoretical results are convincing. There are some questions for technical details and requests for clarification. The main concerns are related to the choice of the field area and the development of the comparison between field example and model results. Both reviewers suggest that the Sierra Nevada is not an ideal choice

for a field test due to its complex history with regards to tectonics and glaciation and the variable geology. The unsolicited comments by Gabet make a similar point. In fact, Gabet thinks there is field evidence contradicting your interpretation. I take these comments seriously and I hope you make all necessary efforts to address them. In any case, it is great to see so much discussion and that you seriously engage with it.

For a revision, please take all the comments into account and provide a detailed rebuttal. In light of the comments on the choice of field area, you may want to think about choosing a different area or adding a second case study.

I am looking forward to seeing your revised paper and thanks for your efforts.

With best wishes, Jens Turowski

---

## Author Response (AR1)

**Response to the Editor**
We made many changes to the manuscript to address both the solicited and unsolicited reviews. The main changes were as follows: 1) additional 1-D simulations exploring tilting over longer timescales and non-uniform uplift that deviates from perfect rigid-block tilting such as that observed when bending an elastic plate or with more pronounced internal deformation; 2) further analysis of 2-D model results to demonstrate that to first-order they are similar to the 1-D model results and to make a more concrete link with profile analysis of field sites; 3) the addition of another field example in Baja California that lacks Quaternary glaciation and for which independent evidence exists that constrains the timing and magnitude of rigid-block tilting; and 4) an expanded discussion section that hits on many of the points brought up by each reviewer and more clearly specifies limitations of the current analysis. Despite these changes, the bulk of the manuscript and the main conclusions remained unchanged. We still find that the Sierra Nevada is a great example of these signatures expressed in a well-studied landscape.

Below is our detailed response to reviewers. Comments by reviewers are reproduced in italics, while our response is in regular text. We responded in detail to comments by Gabet in the open discussion forum and hence do not reproduce those responses here.

*The unsolicited comments by Gabet make a similar point. In fact, Gabet thinks there is field evidence contradicting your interpretation.*
Although Dr. Gabet disagrees with our conclusion regarding the timing of tilt in the Sierra, we point out that the only evidence he discusses that would contradict our results is currently unpublished or embargoed. All published data he referred to actually supports our interpretation that the Sierra experienced rapid tilting in the late Cenozoic. We tried our best in the open discussion to highlight that much of the evidence he uses to support a high Sierra since the Cretaceous can be easily interpreted as evidence for an ongoing transient response to late Cenozoic tilting.

**Response to Reviewers**
Beeson and McCoy, MS No. esurf-2019-24

Response to Anonymous Referee #1

*- Why weren't any simulations included in which tilting is imposed gradually, or, even better, with a varying rate? While the instantaneous tilt scenarios are worth including, it doesn't necessarily seem reasonable to interpret the features resulting from an instantaneous tilt in the same way as those resulting from a realistic, gradual tilt. At the very least, if the authors maintain that the simplified model scenarios produce comparable features that can be interpreted in the same way, some explanation as to why should be included.*
This is a good point. We included four additional simulations in which tilting occurs over 1, 3, 5, and 10 Myr along with detailed analysis of how the signatures of tilting change as the duration over which tilting occurs increases. Now it is clear that the signatures of tilt we define are robust so long as the duration over which tilting occurs is short relative to the channel response time. See section **4.7 Deviations away from instantaneous end-member scenario**.

*- In Figure 7b it's really difficult to tell what's going on in the chi plots. In general, I'd love to see a little more from 2-D model runs included, and analysis of the resulting profiles done in more detail. It's important to examine how even the complexity introduced by the tributaries not being perpendicular to the mainstem affects how well timing and magnitude of a tilt can be resolved. I think it would strengthen the case the authors are making by having detailed analysis of a 2-D tilt simulation as an intermediary between the 1-D analysis and the Middle Fork American River. The inclusion of the 1-D simulations of different sorts of perturbations and the features that appear in river profiles as a result is really nice for*

*this sort of paper, but without some sort of intermediate complexity analysis before jumping right into the Sierra Nevada analysis it almost undermines the authors' conclusions by reminding the reader of all the different ways channel profiles can respond to various perturbations even in an highly simplified model. Then, by the time the analysis of the real river comes around, it leaves me wondering how meaningful the results actually are in a system that's so vastly more complex.*

We included detailed river profile analysis of a mainstem and its tributaries at *t*=2 Ma in the 2-D simulation of instantaneous tilt. We identified mainstem and tributary knickzones and used these to recover tilt timing, as well as using both tributary knickzone drop height and mainstem incision to estimate magnitude. See section **5.2 River profile analysis of 2-D simulated river network.**

*- Figure 7c, would we see a similar relationship between channel segment azimuth and ksn along a linear mountain front where uplift was not driven by tilting?*

No, this is shown by the pre-tilt initial condition in Fig. 9 c and d (formerly Fig. 7) for which no azimuth-gradient relationship is observed.

*- 6.1, Line 14: Would 5 Ma. be the onset or termination of tilting?*

We changed the wording to clarify that 5 Ma is when rapid tilting ceased: "The mainstem slope-break knickpoint occurs at chi=5, thus we estimate that rapid tilting in the northern Sierra ceased ca. 5 Ma."

*- 6.1, Line 26: I have a hard time with any interpretations of knickzones that were above the glacial limit as containing meaningful information about tectonics.*

We added the following text to clarify that we only analyzed knickzones below the Pleistocene glacial limit: "We analyzed all major tributaries to the Middle Fork American River that traverse terrain downstream of the Pleistocene glacial limit as mapped by Gillespie and Clark (2011)."

*- I don't quite understand the rationale of choosing this location to test the model results. I don't know much about Sierra Nevada tectonics, so I'll defer to others on whether recent tilt is a valid hypothesis, but even just the combination of extensive Pleistocene glaciation and lithologic heterogeneity seems like it would make it a difficult place to make a comparison to simple, 1-D model results. Even in the unglaciated reaches of these rivers, sediment supply and discharge would have been varying wildly throughout the Quaternary. Not to say the Sierra Nevada stuff should be thrown out, but it might be more convincing to include some analysis of river profiles from a simpler tilted-block range. The model results in general are straightforward, but where K, m, and n seem like they could vary so widely in space and time, I just don't know that I trust interpretation of the knickzones in this river as being tilt-related features.*

As referenced in the text, there is a significant body of literature documenting evidence for late Cenozoic tilting. However, we included analysis of an additional field site: the Sierra San Pedro Mártir, Baja California, Mexico, a range that is proposed to have experienced onset of slip on range-bounding normal faults in the mid-Miocene and an increase in slip rate in the late Pliocene and that has no evidence of glaciation. Surface geology of the San Pedro Mártir, however, is approximately as heterogeneous as the Sierra Nevada. The fact that we see similar signatures in Baja indicates that these signatures are not a result of glaciation. Although both ranges have heterogeneous lithology, the presence of these signatures despite heterogeneous lithology suggests they are robust signatures of tectonics.

*- In 6.2 and Figure 11, how was it determined whether a knickzone collapsed with the mainstem? Was it just determined visually or were there some other criteria?*

We changed the text to clarify: "We categorized tributary knickzones as those that appear visually to collapse with the mainstem on the chi plot…"

*- 6.3, Line 23: Shouldn't this degree of tilting be causing pretty rapid migration of the main divide? That could really complicate sorting out knickpoint migration velocities.*

We added the following paragraph to the discussion that includes how drainage area increases would impact estimates of tilt timing: "Estimates of timing made using methods outlined herein may also be sensitive to heterogeneity in rock properties, processes, and the dynamics of real landscapes, even in landscapes where signatures of tilt are strongly expressed. Estimating timing requires constraining K, m, and n, with standard methods utilizing equilibrium rivers - a difficult requirement in transient landscapes. Even if these can be constrained, any perturbation to knickpoint travel times will affect estimates of tilt timing. For example, nonuniformity in K can result in either faster or slower knickpoint travel times and result in either overestimates or underestimates of tilt timing, respectively. Furthermore, calculations of travel times are generally based on the modern river network, but major topologic change in river networks (e.g., Willett et al., 2018) following tilting has the potential to impact estimates of timing. Drainage area gain that occurs after knickpoints have propagated for some time would result in underestimation of time since tilt as area gain moves profiles to the left of equilibrium in chi plots (Willett et al., 2014), thus resulting in lower tau values."

*- Along with this, it might be really cool to analyze the back-tilted catchments on the opposite side of a tilting range in conjunction with the forward-tilted ones.*

Rivers draining both the eastern side of the Sierra Nevada and the Sierra San Pedro Mártir are not so much back tilted, but rather drain large escarpments formed by active normal faulting. The drainages on the eastern side of the Sierra Nevada were heavily glaciated and are thus are bit complicated to analyze, but the eastern draining rivers in the Sierra San Pedro Mártir have been extensively studied by Rossi et al. (2017). This study provides a nice complement to ours and indeed similar histories seem to be recorded on opposite sides of the range. On the west side, our new analysis shows that the tilt signatures constrain the recent accelerations in tilt rate that Rossi et al. propose generated a suit of knickzones on the east side.

*Figure 13 could really benefit from a chi plot or something to better show the capture event. It's hard to tell exactly why it's being interpreted that way.*
We added an arrow to show the pre-capture flow path. We left the chi plot off because the chi plot of this capture (and of many captures in the Sierra) are too impacted by tilting to show the characteristic signature of capture.
Response to Anonymous Referee #2

*That said, my main critique of the manuscript is that the 2-D landscape evolution modeling and application to the Sierra Nevada feel a bit underdeveloped relative to the 1-D river profile modeling.*
We added additional analysis of the 2-D landscape evolution model (river profile analysis of mainstem and tributaries at t=3 Ma) and an additional field example of signatures of tilt expressed in river profiles and networks in the Sierra San Pedro Mártir, Baja California.

*I wonder about the frequency and rates of divide migration associated with both gradual divide migration and discrete capture events and the river network response times. How does discrete vs. gradual drainage reorganization impact channel morphology (and how does this relate back to the 1-D profile modeling scenarios of tilting and major truncation/beheading of mainstem rivers)?*
We agree that these are interesting questions but answering them would require significantly more research and they are thus out of the scope of this paper. We did add discussion points to highlight how changing drainage area could affect estimates of timing of tilt.

*How does tributary junction angle modify the morphology of tributary knickzones or steepness patterns, and what affect does this have on river profile analyses?*
We conducted river profile analysis on mainstem and tributaries at t=2 Ma in the 2-D model of instantaneous tilt and within that section include the following text:

"Tributary junction angle does not impact the magnitude of tributary knickzone drop height nor the degree to which tributary knickzones collapse with the mainstem."

*I also suspect that drainage reorganization in the landscape evolution model is sensitive to the model discretization and flow routing algorithm you use (as mentioned on pg. 22, line 5), and it would be nice to see some test/statement of how robust the results are to these factors. It feels a little like a missed opportunity to perform the 2-D landscape evolution modeling and not take a closer look at the diversity of river profile morphologies that develop in the evolving river network.*
Again, while we think questions relating to river basin dynamics are valuable questions, given their complexity full exploration of them is out of the scope of this paper. We did add discussion points to highlight how changing drainage area could affect estimates of timing of tilt and profile shape.

*For the Sierra Nevada analyses, at pg. 25, Line 9, why do you chose to use a concavity theta = 0.45? Does this linearize and/or collapse mainstem and tributary rivers in the region on chi plots?*
We added the following text to address this: "Standard techniques for finding theta cannot be applied given the disequilibrium state of Sierra rivers. Thus, we chose to use theta =0.45 as it is a common value and allows for comparison to modeled profiles."

*How do the knickpoint travel times tau (and inferred timing of the tilting event) vary for n = 2/3 or n = 5/3?*
We added the following text: "With values of n≠1, tau cannot be calculated directly. However, based on 1-D model results, we can see that if n=2/3 but n=1 is assumed when analyzing river profiles, response times increase and thus 5 Ma would be an underestimate (Fig. 4S), whereas if n=5/3 but n=1 is assumed when analyzing river profiles, response times decrease and thus 5 Ma would be an overestimate."

*What about tilting associated with postglacial rebound? Granger & Stock (2004) measure tilt angles up to ~0.08 deg in the past ~15 kyr from cave deposits in the Southern Sierra Nevada near the San Joaquin and Kings Rivers. How would this secondary tilting event affect your results, considering even the case in which 90-95% of tilting occurred ~5 Ma and the remaining 5-10% occurred since the LGM?*
The tilt angle associated with postglacial rebound is insignificant compared to that estimated from tectonics. However, we had already included how secondary tilting would impact signatures of tilt – see Fig. S11. In short, a sequence of punctuated tilting events can be distinguished in modeled river profiles, but would likely be difficult to distinguish in natural profiles with more heterogeneity.

*Is it reasonable to analyze channels above the Pleistocene glacial limit? To what extent does this topography retain its glacial form (are valleys C3 ESurfD Interactive comment Printer-friendly version Discussion paper v-shaped above 1000 m?) and how, even downstream of the glacial limit, could Quaternary changes in discharge and sediment supply affect river profile morphology?*
We did not analyze channels above the Pleistocene glacial limit. The mainstem slope-break knickpoint is far below the glacial limit and we added the following text to clarify that all tributaries that we analyzed are also below the glacial limit: "We analyzed all major tributaries to the Middle Fork American River that traverse terrain downstream of the Pleistocene glacial limit as mapped by Gillespie and Clark (2011)"

*Are there any knickpoints you can definitively associate with tilting rather than a transient response to deglaciation or heterogenous lithology?*
Yes, the positive-curvature slope-break knickpoint in the mainstem. This is one of our main points, so hopefully it is clear. We made minor changes to try to help emphasize this point throughout.

*In Figure 11, if you exclude all tributaries above the glacial limit, does your tilt magnitude estimate change?*

We added the following text: "We analyzed all major tributaries to the Middle Fork American River that traverse terrain downstream of the Pleistocene glacial limit as mapped by Gillespie and Clark (2011)". The tributaries analyzed did not change, but in the initial submission our criteria for picking tributaries was accidentally omitted.

*With its glacial history and lithologic variability, the Sierra Nevada seems like a pretty complex test case to evaluate the real-world applicability of fluvial metrics of tilting identified in the modeling. The relative convergence of the results and consistency with previous interpretations is compelling, so perhaps the complexity proves the robustness and widespread potential for extracting tilt histories from river profile and network morphology. However, at present, I'm not sure the authors have fully addressed these confounding factors nor sufficiently quantified the sensitivity and associated uncertainty of the Sierra Nevada results to their assumptions and interpretations.*

To address this comment we added three new pieces to the paper:

1) A second field example (Sierra San Pedro Mártir, Baja California) in which similar signatures of rapid tilting are observed and for which independent evidence of tilting has already been published. This is a nice comparison to the Sierra example in that the length scales and tilt magnitude are similar, yet the Sierra San Pedro Mártir do not have a Quaternary glacial history.

2) More rigorous estimates of theta and K in the Sierra Nevada that provide quantification of uncertainty in this key parameter: "Standard techniques for finding theta cannot be applied given the disequilibrium state of Sierra rivers. We used slope-area plots of mainstem rivers downstream of slope-break knickpoints to estimate that theta=0.41-0.48, and thus we use theta=0.45 for all analyses in the Sierra. We calculated K for the Sierra batholith using a selection of published [10]Be-derived denudation rates and their associated $k_{sn}$ values. From a compilation of erosion rates for basins on an unglaciated, low-relief upland surface in the Sierra (Callahan et al., 2019), we selected basins with $R^2$>0.9 for linear regression of chi-elevation data above the sampling point for catchment-average erosion rates such that erosion rates should be in approximate equilibrium with uplift rates (Fig. S13). This yielded 21 basins from which we calculated a mean K of $1 \times 10^{-6} \pm 0.03 \times 10^{-7}$ m$^{0.1}$yr$^{-1}$."

3) A paragraph to the discussion addressing the factors that may impact estimates of timing in real landscapes. "Estimates of timing made using methods outlined herein may also be sensitive to heterogeneity in rock properties, processes, and the dynamics of real landscapes, even in landscapes where signatures of tilt are strongly expressed. Estimating timing requires constraining K, m, and n, with standard methods utilizing equilibrium rivers - a difficult requirement in transient landscapes. Even if these can be constrained, any perturbation to knickpoint travel times will affect estimates of tilt timing. For example, nonuniformity in K can result in either faster or slower knickpoint travel times and result in either overestimates or underestimates of tilt timing, respectively. Furthermore, calculations of travel times are generally based on the modern river network, but major topologic change in river networks (e.g., Willett et al., 2018) following tilting has the potential to impact estimates of timing. Drainage area gain that occurs after knickpoints have propagated for some time would result in underestimation of time since tilt as area gain moves profiles to the left of equilibrium in chi plots (Willett et al., 2014), thus resulting in lower tau values."

*Pg. 2, Line 30 - What about documented response to tilting associated with dynamic topography? There are a number of papers (e.g., Braun et al. 2013, Ruetenik et al. 2016) that have addressed fluvial response to long wavelength dynamic uplift and subsidence, including continental-scale tilting. It seems like this should be mentioned or at least the distinction in underlying mechanism and/or the scale of tilting should be clarified.*

We added the following text to the discussion section: "We focused on moderate magnitude tilting (0.5-1 degree) at 100 km scales over which fluvial networks are well developed. Signatures of the fluvial

response to high magnitude tilting over short lengths scales as is observed in the Basin and Range (e.g., Stewart, 1980) may be quite different as river networks are much smaller and hillslope processes may dominate over fluvial. Similarly, fluvial signatures of very low magnitude tilting over large wavelengths as can occur through dynamic topography (e.g., Liu and Gurnis, 2010) may differ substantially from those presented herein as near-rigid-block behavior is less likely over long length scales (e.g., Martel et al., 2014) as are the required assumptions of uniform concavity and uniform erosional processes."

*One major assumption underlying all model tests is the assumption of uniform concavity, which could very well be violated if the mechanisms of river incision change with channel slope or in different lithologies.*
We address this by including the following text in the Discussion: "Deviations away from n=1, which may be associated with a shift in erosional mechanisms (Whipple et al., 2000), would also impact transient profile forms and would additionally affect estimates of timing made from knickpoint locations. In particular, a shift in m and n or a transition from detachment-limited to transport-limited between tributaries and mainstem rivers might generate similar signatures in a river network that simply reflect a change in process rather than tectonics."

*There are multiple places (e.g, in presenting the first results in Figure 2 with vertical knickpoints and in presenting the results in Figure 4 with heterogeneous lithology) where this assumption should be restated and its limitation clarified.*
We chose to address the limitations of applying these signatures to real landscapes in the discussion rather than throughout the model results (see above).

*It's a little hard to tell in Figure 10 since lithology is not included in (b) (I suspect since what would ideally need to be displayed along the chi axis is integrated lithology), but it appears to me, looking at the elevation ranges, that river profile concavity could vary between the different lithologies.*
It is not possible given the disequilibrium state of Sierra rivers to determine whether equilibrium river profile concavity differs between lithologies. We added the following text to clarify: "Standard techniques for finding theta cannot be applied given the disequilibrium state of Sierra rivers. Thus, we chose to use theta=0.45 as it is a common value and allows for comparison to modeled profiles."

*The assumption of a negligibly small/frequently exceeded threshold for bedrock river incision should be stated and the limitations of this assumption should be addressed, particularly since I imagine tilting could have a significantly enhanced impact of bedrock river incision if it increased channel gradients enough to change the frequency and magnitude of events exceeding a threshold for bedrock river incision.*
We added the text in bold to this sentence in the discussion: "Similarly, processes not explored here, such as sediment flux dependent erosion (Lague, 2014; Sklar and Dietrich, 1998; 2004), **thresholds in shear stress or stream power in controlling bedrock incision (e.g., Lague et al., 2005; Snyder et al., 2003a; Snyder et al., 2003b; Tucker, 2005),** or changes in channel width with uplift rate (Attal et al., 2011; Finnegan et al., 2005; Lague, 2014; Turowski et al., 2009; Whittaker et al., 2007; Wobus et al., 2006; Yanites and Tucker, 2010; Yanites et al., 2010), may also have the potential to confound signatures of tilt presented here."

*Pg. 11, Line 20 - Should be noted that such 1-D modeling neglects area-loss feedback and full 2-D/network dynamics of drainage reorganization*
We added the following text to that section: "It should be noted that all truncation simulations are limited in that they do not include the positive feedback often associated with drainage area loss (Willett et al., 2014) that may limit the ability of the river basin to achieve equilibrium steepness following truncation."

*In the 1-D profile modeling, I suspect some of your quantitative results (e.g., estimated tilt from geometry of local rock-type related knickzones) may be somewhat sensitive to your forward differencing scheme and discretization. I suggest clarifying which forward differencing scheme you use.*
We added "upwind-space" to the following sentence in the methods: "To simulate the evolution of the land surface, we numerically solved the following governing equation using a forward-time upwind-space finite-difference solver"

*In the same vein, as mentioned above, in the 2-D LEM, the results are likely sensitive to the flow routing algorithm, which is not specified in the text.*
It is specified in the Methods: "To map both real and simulated river networks, we calculated flow direction and accumulation using a steepest descent flow algorithm."

*What happens if you add noise (even much lower magnitude than the heterogeneous lithology K contrast) to your spatially variable K, as you might expect if there's spatial heterogeneity in rock erodibility even within the same lithology? Is it still possible to recover tilt timing and/or magnitude?*
We had already answered this question in Fig. S8.

*Also, many landscapes where tilting is suspected to have modified the topography are landscapes with tilted sedimentary rock units. Tilting of such tilted strata would effectively result in spatially and temporally variable erodibility within bedrock channels, as rivers incise vertically into a tilted layer, causing the exposure of that layer to migrate up/down stream depending on the river orientation and tilt magnitude and direction. It seems like this at least warrants mention.*
We added the following text to the discussion: "Furthermore, we did not attempt to simulate the transient response to tilting in sub-horizontally layered stratigraphy, the dynamics of which are likely to be considerably more complex (e.g., Forte et al., 2016)."

*Pg. 19, Line 5 - Specify what type of curvature*
We changed the following sentence to read: "We identified the upstream and downstream ends of the rock-type slope-break knickpoint defining the knickzone base by identifying the appropriate peak and trough in profile curvature **on the chi plot** and stopped the simulation when the knickpoint was at the upstream end of the band and the quasi-equilibrium reach stretched only the full length of the band"

*Figure 10 - I don't think the rock types should be extended all the way down to the x-axis, since this seems to imply that the rock units have vertical dips extending to depth. I suggest fading away the lithology shading just below the river profile.*
While we agree this would look nice, it proved to be quite time consuming to make it look nice. Instead, we added the following sentence to the caption: "Surface geology is shown with vertical contacts to schematically show the generally steep dips of contacts found in the Sierra."

*Pg. 27, Line 5 - Would be helpful, for reproducibility, to specify the threshold you use to determine if tributary knickzones collapse with the mainstem on the chi plot or not. Here and elsewhere, ditto for how exactly you define a knickzone (e.g., what threshold change in steepness do you use to identify the top and bottom?)*

[revised manuscript text omitted]

---

## Referee Report (RR1)

The authors have obviously done a lot of work to address reviewer concerns, and I think the paper is much improved for their revisions. Their addition of more 1 and 2-D model results under a wider variety of conditions, especially for non-instantaneous tilt strengthens their conclusions quite a bit. I also appreciate the addition of the Sierra San Pedro Mártir case study, which is nice to see given the discussion around Sierra Nevada tectonics. Since the conditions that can produce a particular river profile are nonunique, additional data showing river response to tilting under a wider variety of circumstances make the authors' conclusions much more convincing. I saw just a few minor errors/points of confusion.

Page 21, lines 6-7: Should this say "the vertical bed of less erodible rock"?

Page 25, line 26: mainstem

Page 26, lines 3-5: To what part of Figure 8 is this sentence referring? The river flowing toward the upper left that gets captured?

Figure 16: I'm a little confused by the pre-capture flow path label, it looks like it was originally flowing in the opposite direction?

Page 44, lines 23-24: This 3 Myr figure is dependent on parameter values, right? Maybe should specify that 3 Myr was specific to your model runs?

---

## Referee Report (RR2)

*Review of Beeson and McCoy*

By E. Gabet

**Summary**

This manuscript consists of 3 parts: modeling, application of the model results to the northern Sierra Nevada, and application of the model to a site in Mexico. With respect to the first part, it's a standard streampower-based exploration of different scenarios, albeit with some odd initial conditions (eg, 1000 m of instantaneous uniform uplift). The novelty is that the authors are exploring the consequences of tilting and there is value in this exercise.

Problems arise in the second part and most of my comments below are focused on this section. The history of the Sierra Nevada is a topic with important consequences regarding our understanding of the geologic evolution of western North America. Because the stakes are high, published results have to be robust. However, many of the assumptions, interpretations, and results presented here are contradicted by the field evidence. For example, the authors base some analyses on the location of a knickpoint that they claim is a tectonically-driven migrating feature; however, all of the evidence indicates that it is a lithological/structural knickpoint. In another example, their reconstruction of the geological history of the Middle Fork American River canyon is refuted by the stratigraphic evidence.

With respect to the analysis of the Mexican site, I am not familiar with the region and was unable to procure a geologic map of the area. However, I am concerned that, although the analyses yield results with respect to tilt magnitude and timing, there doesn't appear to be any data with which to validate them. Usually, when new approaches are developed, their results are tested against known data – this provides confidence in the new approach so that it can be applied to new areas, and it gives us an idea of how accurate it is. In this case, a *new* approach is being applied to a *new* area and so an important step is being skipped. I would recommend applying these techniques first to a site where the tectonic history is well-known. Until that is done, we don't have any way of scientifically assessing the validity of this approach or the robustness of the assumptions (eg, uniform erodibility).

**Comments according to section heading**

*Introduction (p. 3)*

To motivate the analyses done in the Sierra, the authors cite several papers that they claim support the hypothesis of Late Cenozoic tilting; however, these studies have all been debunked. Below is a brief synopsis of the fatal flaws in each of them; more detailed explanations can be found in Gabet (2014). If the authors would like to cite these papers, they will need to explain why the analyses of these fundamental flaws are incorrect; otherwise, there's not much value in referring to discredited studies.

Lindgren (1911) based his tilt estimate with the assumption that he correctly reconstructed the Tertiary paleochannels. Both Gabet (2014) and Cassel (2012) demonstrated that his reconstructions were fundamentally flawed, either because they imply that water flows uphill or because he was linking channel segments that were unrelated in time and space. Moreover, Gabet (2019) demonstrates that, even if Lindgren had correctly reconstructed the channels, the differences in their gradients that he attributed to tilting can be wholly explained by differences in bedrock erodibility.

Jones et al (2004) based their analyses on Lindgren's reconstructions. Since those reconstructions are faulty, their analyses are meaningless. In fact, the reconstruction of the South Yuba River in Jones et al has reaches where water flows uphill (an absurd result). Moreover, like Lindgren, Jones et al ignores the role of erodibility in controlling channel slope.

Wakabayashi (2013) calculated bedrock incision depths and rates from Pliocene volcanic deposits along the rims of canyons without recognizing that older deposits can be found farther down the canyon walls.

Unruh (1991) based his tilt estimate of 1.4° on the gradient of Central Valley sediments. By using simple geometry, one can demonstrate that a consequence of this result is the prediction that, at some point in the mid-Cenozoic, Tertiary gravels at an elevation of ~700 m along the Yuba river were once ~500 m below sea level. In other words, untilting the northern Sierra by 1.4° places sections of the mountain range deep underwater sometime in the past 30-50 my. This study is refuted, therefore, by the absence of deep Cenozoic marine sediments in the Sierra.

**4 Modeling fluvial longitudinal profile response to perturbations (p. 7)**

In this section, the authors describe the profiles from a series of numerical simulations that are then compared to the Middle Fork American River. Their model, based on the simple streampower formulation, is used to show that tilting leads to knickpoints that migrate up from the range-front. However, this result is obtained by making the extraordinary claim that rock erodibility is uniform throughout the area. In fact, rock erodibility is extremely *non-uniform* throughout the range and is the primary control on channel steepness (Gabet, 2019). Shown below is the profile of the North Fork Feather River demonstrating the important role of lithology and erodibility on profile shape. Note how nearly every knickpoint is associated with a lithological boundary and how the steepness index (the second number above each reach) varies according to rock unit, even from one granitic unit to the next.

[Figure]

*4.9.2 Estimating tilt magnitude from rock-type knickzone geometry*

In this section, the authors estimate tilt based on the geometry of a river profile that flows across weak rock that is sandwiched between strong rock. There were a number of issues here.

(1) To develop their model (Fig. 7), the authors are assuming *uniform* uplift in the northern Sierra Nevada (p. 23, l. 7,8). There is, however, *no* evidence for uniform uplift in the northern Sierra Nevada during the Cenozoic. The authors are the first to ever make such a claim and they do so without providing any evidence. Uniform uplift would create an obvious fault scarp hundreds of meters high along the entire western range-front, a feature which has not been observed.

(2) This technique is dependent on the assumption that the reach formed during the transient response to tilt is similar to the initial steady-state river slope. The authors, however, have not provided any evidence to support the claim that the pre-tilt river was at steady-state. Given the multiple episodes of aggradation, incision, and drainage reorganization experienced by these rivers due to repeated volcanic eruptions, the odds that this assumption is correct are vanishingly small.

(3) The authors are assuming specific erodibility values for the rock units at this site along the Rubicon River without any evidence that these values are accurate. They are using a *K* value determined from a different granitic bedrock unit in the southern Sierra Nevada and applying that to the Rubicon site without accounting for the fact that erodibility can vary greatly in granitic rocks (Gabet, 2019). For the Jurassic marine rocks, which the authors take as the "erodible"unit, they are assuming that it is 10x more erodible than the granitic rocks but, again, without any evidence. In fact, the Jurassic marine rocks include quartzite, which is as strong as granitic rocks, and greywacke, which is also very strong and certainly not 10x weaker than granitic bedrock (Gabet, 2019).

(4) Finally, the field site does not conform to their model. Below (left) is their figure illustrating their model and (right) the actual profile. Note how, in the model, the dashed line extends smoothly from the profile at the top of the knickzone down to the lower extent of the "erodible" rock. In the real river, extending the profile above the knickzone in a similar manner yields a completely different geometry.

[Figure]

(5) To summarize, this tilt estimate is wholly dependent on the claim that the Sierras have experienced uniform uplift, which is demonstrably incorrect, as well as several assumptions which are unlikely to be all correct, and a geometric analysis that

does not apply to the field site. It would be difficult to conclude, then, that the tilt estimate from this analysis is scientifically sound.

*6.1 Disequilibrium form of the mainstem Middle Fork American (p. 31)*

In this section, the authors identify a knickpoint in the Middle Fork American River and assume that it is a migrating knickpoint generated by uplift (black triangle in the profile below). This knickpoint, however, is associated with both a lithological and structural boundary.

[Figure]

Although the authors have combined the metamorphic rocks (in green) directly above and below the knickpoint into a single unit, this obscures the fact that they are different formations with different lithologies (note that the pink band of granitic rocks that they show immediately above the knickpoint does not exist on any map that they cite). I have labelled these formations as CC and SF on their figure; I have also added a red dotted line to show the location of the Volcano Canyon Fault that forms the contact between the two (ignoring, as the authors have, smaller units at that site). The metamorphic unit on the upstream side of the knickpoint is the Paleozoic Shoe Fly Formation (SF), which is composed of resistant quartzite and metavolcanic rocks. The unit on the downstream side of the knickpoint is the Paleozoic Calaveras Complex (CC), a highly sheared subduction melange that includes weaker argillite and chert. From Gabet (2019), the steepness index (a measure of erodibility) of the Shoe Fly Formation is 0.13 while, for the Calaveras Complex, its average is 0.07 (note, these values are from other rivers and are, therefore, independent of the particular situation on the Middle Fork). The difference in the steepness index between the two units is strong evidence that this is a lithological knickpoint and, moreover, that the assumption of uniform erodibility is violated.

In addition, their profile does not show that the area downstream of the knickpoint is within the Foothills Fault System where many faults have made the bedrock more erodible (Gabet, 2019) – I have added a label and a bracket to their figure to show the extent of this fault zone. The figure below shows the lithological map of this area where the faults and the different units can be seen (a yellow line across the Middle Fork marks the knickpoint).

[Figure]

Another piece of evidence indicating that this knickpoint is not a migrating feature comes from the surrounding hillslopes. As demonstrated in Hurst et al. (2013), hillslopes just downstream of a knickpoint will be steeper than those upstream. However, in the slope map below, where the black line shows the location of the knickpoint and the river flows from right to left, the hillslopes are actually *steeper* upstream of the knickpoint (red is steep, blue is gentle). This slope map, therefore, not only contradicts the assumption that this is a migrating knickpoint but, instead, it supports the conclusion that the reach of river upstream of the knickpoint is steeper because the rocks are stronger.

[Figure]

Thus, not only have the authors not provided any evidence that this is a migrating knickpoint, all of the available evidence indicates that it is a lithological knickpoint. As a result, the analyses based on the claim that this is a migrating knickpoint are fundamentally flawed unless the authors can demonstrate that this is, in fact, a migrating feature (ie. via field observations rather than model results); this includes the results regarding the magnitude and timing of uplift which I discuss below.

Based on the identification of the knickpoint as a migrating feature, the authors use its location to make estimates about the timing of uplift. To make this estimate, the authors assume that *erodibility is uniform* along the lower part of the profile that the knickpoint travelled across; moreover, they assume that *the erodibility of those rocks is the same as granitic rock* in the southern Sierra Nevada (p. 31, l. 25). Both of these assumptions are contradicted by the field evidence. In the map below, the section of the Middle Fork below the knickpoint begins at the top left corner and then flows diagonally down towards the northern part of Folsom Lake, through the lake and into the Central

[Figure]

Valley. Along this path, it crosses a dozen different geological units, and only one of these is granitic while the rest are various types of metasedimentary and metavolcanic rocks, including a highly sheared subduction melange. As demonstrated in Gabet (2019), the erodibility of these rock types vary greatly; for example the steepness index of the subduction melange is about half that of granitic rock. Furthermore, the Middle Fork crosses six faults, which means that the rocks at those locations will be much weaker than at other spots (Gabet, 2019). Therefore, in addition to problem of assuming that the knickpoint is a migrating feature, the two other assumptions necessary for calculating the timing of uplift are violated

Finally, in this section, the authors estimate the magnitude of recent tilt based on alleged incision depths beneath Mio-Pliocene volcanic deposits (in brown below) and Eocene fluvial gravels (in red). The estimates of incision depths, however, are contradicted by the field evidence.

[Figure]

For example, at the site just above the black triangle on the plot above, the authors are claiming that there has been ~600 m of incision since the Miocene-Pliocene (see the

regression line) and that this has been driven by recent uplift. However, the elevation of the Eocene deposits *proves* that there could *not* have been more than ~450 m of incision since the Eocene. In the illustration to the left, I show the present day profile across the Middle Fork at the top, Beeson and McCoy's interpretation of this profile (left column), and the standard interpretation to the right. The critical point is that the authors are claiming that the Mio-Pliocene volcanics represent a bedrock paleosurface and that all of the relief below the volcanics is due to Pliocene incision into basement rock. However, their interpretation cannot explain the presence of Eocene deposits *below* the Mio-Pliocene deposits. The only interpretation consistent with the field evidence is that the Eocene gravels represent the bedrock surface during the Eocene when this canyon was already at least 150 m deep. This landscape was then buried by volcanic deposits that are known to have been hundreds of meters thick. To put it bluntly, because 450 < 600 and because the Eocene is older than the Mio-Pliocene, the estimate for recent tilt based on the regression line in the incision plot is invalid. The only accurate statement that can be made is that there's been a maximum of ~450 m of incision since the Eocene – Early Oligocene.

The use of the Mio-Pliocene volcanic deposits on the interfluves as an indicator of a paleosurface was promoted in papers by Wakabayashi (Wakabayashi and Sawyer, 2001; Wakabayashi, 2013) and this method has since been debunked as older sediments can be found far below them (Gabet, 2014). Because the volcanic rocks are remnant patches left on the ridges, the plot above is simply showing that relief increases from the range-front and then decreases as the crest is approached, as shown in (Gabet, 2014); compare the incision pattern of the volcanic rocks in the plot to the left with the plot of relief below (solid line).

[Figure]

[Figure]

*6.3 Tilt magnitude recorded in the stream network* (p. 35)

In this section, the authors use a relationship between reach azimuth and gradient to look for signatures of tilt. Based on their analyses, the authors conclude that
 there has been 2.3° of recent tilt. Their technique can be tested using field observations. As noted by Huber (1981), the upper uneroded surface of a 10 Myo lava flow along the San Joaquin River (which is only 5-10 km north of where the authors did one of their analyses) forms a series of table mountains. The source of this flow was the Sierra Nevada (top-right of the map shown to the left) and it flowed down into the Central Valley (bottom-left of the map). The line in the figure to the left shows the transect plotted on the next figure.

The upper surface of the flow is at an angle of 1.37° (first figure below). If we subtract the 2.3° of recent tilt hypothesized by the authors, the upper surface of the flow is now at -0.9° (ie. 1.37 – 2.3 = -0.9; second figure below). This means that, if there had been 2.3° of recent tilting (as the authors claim), this lava would have flowed uphill 10 Mya. This result challenges the approach presented here.

[Figure]

The authors state that their results are a "maximum possible tilt magnitude (p. 36, l. 7)". However, the table mountains data show that the maximum is actually ~1.4°, indicating that their technique is, in fact, unable to constrain the maximum amount of tilt. Moreover, without providing any bounds on the *minimum* possible tilt magnitude, their results are consistent with 0° of tilt. Therefore, this analysis isn't providing any new information. At a minimum, a few other nearby sites should be analyzed because a sample size of 1 is too small to provide confidence in this technique. This would have the benefit of providing some measure of the potential error.

*References cited*

Gabet EJ. 2014. Late Cenozoic uplift of the Sierra Nevada, California? A critical analysis of the geomorphic evidence. American Journal of Science **314**: 1224-1257. DOI: 10.2475/08.2014.03.

Gabet EJ. 2019. Lithological and structural controls on river profiles and networks in the northern Sierra Nevada. Geological Society of America Bulletin **in press**.

Huber NK. 1981. Amount and timing of late Cenozoic uplift and tilt of the central Sierra Nevada, California - Evidence from the upper San Joaquin River basin. In *U.S. Geological Survey Professional Paper 1197*; 29.

Hurst MD, Mudd SM, Attal M, Hilley GE. 2013. Hillslopes record the growth and decay of landscapes. Science **341**: 868-871.

Wakabayashi J. 2013. Paleochannels, stream incision, erosion, topographic evolution, and alternative explanations of paleoaltimetry, Sierra Nevada, California. Geosphere **9**: 1 - 25.

Wakabayashi J, Sawyer TL. 2001. Stream incision, tectonics, uplift, and evolution of topography of the Sierra Nevada, California. The Journal of Geology **109**: 539 - 562.

---

## Referee Report (RR3)

**Review of 'Geomorphic signatures of the transient fluvial response to tilting' by Beeson and McCoy**

The paper under review is a revision of a previous manuscript looking at geomorphic response to tilting in fluvially dominated landscapes (I was not a reviewer of the original manuscript). I think this is an interesting study which sets out nicely different analytical and numerical models of tilting, and that the authors did a good job in the first revision of adding these scenarios and expanding the discussion of the 2D numerical modelling. I also think it is good to add in the second real landscape test site due to the ongoing controversy about Sierra Nevada uplift. However, I have some general concerns detailed below, specifically about whether the suggested signatures of tilting from $\chi$ plots are robust, followed by some more minor comments. The line numbers below refer to line numbers in the tracked changes document.

My main issue with this study is that of equifinality: many of the diagnostic features that the authors attribute to tilting could be the result of different processes, or simply in the choice of how you extract the channel networks and calculate $\chi$. For example, in real landscapes the location of the channel head will have a big impact on the resulting $\chi$ plots. Performing $\chi$ transformation above the channel head will lead to non-linearity of the chi plot which may result in a very similar looking plot to the scenario of higher uplift at the outlet shown in Fig 1d (see Clubb et al. (2014), Fig 8). For the model runs this isn't a big deal, but if you chose an unrealistically low area threshold in a real landscape this could look the same as a spatially variable uplift field. This is not to say that this invalidates the study, but I think the authors could more carefully consider and elaborate on other reasons for non-linearity within $\chi$ plots. Other issues could be the influence of glaciation or mass-wasting processes, although I'm not aware of work at present which explores how these processes might influence $\chi$ plots.

A related, and more important problem, is that varying the concavity value used to calculate the $\chi$ plots can lead to very different curvatures: changing $\theta$ from 0.2 - 0.8 will completely change the curvature of the tributaries from convex to concave. I ran the chi analysis quickly on the 90 m SRTM data from the Middle Fork American River site in the Sierra Nevadas (Figure 1) which demonstrates this effect. The authors state in their discussion (Page 47, Lines 26 - 30) that changes in $m$ or $n$ might violate their assumption of uniform concavity. However, even if there is uniform concavity, if you have selected an incorrect $m/n$ value for the basin, this could lead to an identical signature in the $\chi$ plot as that of a rigid block tilting scenario. I am concerned that the conclusions of the paper might lead other studies to assume that negative curvature in $\chi$ elevation space is always a result of tectonic processes, when in fact it could be a simple artefact of how $\chi$ was calculated.

There are a lot of different scenarios presented in the paper which were not related to tilting (e.g. change in elevation of the channel profile, drainage capture, uniform change in uplift rate). A lot of these results, and the discussion of knickpoint celerity, are not new to this paper and are similar to the work of other authors (Royden and Perron, 2013; Willett et al., 2014). I would suggest to either remove these analyses to focus on the novel points of this paper (the tilting scenarios), or to put these results more in context of the work that has already been done.

For the real landscapes, the authors state that they used slope-area plots to estimate $\theta$. I found this odd, as the noise present in SA plots often makes the calculation of concavity challenging, and is one of the reasons that $\chi$ plots were initially developed. Why did the authors not use a $\chi$ based method of calculating concavity? I think the statement that 'Standard techniques for finding $\theta$ cannot be applied given the disequilibrium state of Sierra rivers' (Page 33, Line 8) is unsupported - there have been many techniques that have developed ways of estimating concavity in transient systems which are not cited in the study (e.g. Mudd et al., 2018; Hergarten et al., 2016).

Page 2, Line 10: I would suggest to also cite Morisawa (1962) here, as it reports slope versus drainage area prior to that of Flint and is often missed out in subsequent literature for some reason.

Page 3, Line 1: change to 'documented across many tectonic settings'

Page 3, Lines 18 - 22: were all these scenarios with the same tilt angle? It would be good to clarify this here.

Page 3, Line 30: very long sentence, split up.

Page 5, Line 10: typo in subheading - repetition of $\chi$ symbol and misspelling of 'transience'

Page 5, Line 21: Replace 'curved' with 'concave-up'

Page 5, Lines 24 - 26: This is only the case if the correct $m/n$ ratio is chosen for the channel. I think it is worth mentioning this caveat.

Page 6, Line 25: more recent studies have since demonstrated that concavities outside 0.4 - 0.7 are common and that $n$ is commonly greater than 1 (Lague, 2014; Harel et al., 2016).

Page 9, Line 7: remove word 'very'

Page 9, Lines 7 - 17: I found myself getting confused by all these scenarios. Would it be possible to put this in a table to make it clearer?

Page 10, Line 1: why equal to five? This could be explained more clearly.

Page 10, Line 5: repetition of 'in the'

Section 4.2: Which scenario was actually ran? Increase in K or decrease in U?

Figure 2: what timescale does the dashed blue line represent? Is this the same as the response time $\tau$?

**References**

Clubb, F. J., Mudd, S. M., Milodowski, D. T., Hurst, M. D., and Slater, L. J. (2014). Objective extraction of channel heads from high-resolution topographic data. *Water Resources Research*, 50(5):4283–4304.

Harel, M.-A., Mudd, S., and Attal, M. (2016). Global analysis of the stream power law parameters based on worldwide 10be denudation rates. *Geomorphology*, 268:184–196.

Hergarten, S., Robl, J., and Stüwe, K. (2016). Tectonic geomorphology at small catchment sizes–extensions of the stream-power approach and the $\chi$ method. *Earth Surface Dynamics*, 4(1):1–9.

Lague, D. (2014). The stream power river incision model: evidence, theory and beyond. *Earth Surface Processes and Landforms*, 39(1):38–61.

Morisawa, M. E. (1962). Quantitative geomorphology of some watersheds in the appalachian plateau. *Geological Society of America Bulletin*, 73(9):1025–1046.

Mudd, S. M., Clubb, F. J., Gailleton, B., and Hurst, M. D. (2018). How concave are river channels? *Earth Surface Dynamics*, 6(2):505–523.

Royden, L. and Perron, T. J. (2013). Solutions of the stream power equation and application to the evolution of river longitudinal profiles. *Journal of Geophysical Research: Earth Surface*, 118(2):497–518.

Willett, S. D., McCoy, S. W., Perron, J. T., Goren, L., and Chen, C.-Y. (2014). Dynamic reorganization of river basins. *Science*, 343(6175):1248765.

[Figure]

Figure 1: χ plots for the Middle Fork American River showing the change in curvature with the selected value of θ from 0.2 to 0.8. This may be incorrectly identified as a signature of tilting. MLE = most likely estimator, colours represent similarity of each tributary compared to the main stem.

---

## Author Response (AR2)

**Response to the Editor**

As suggested by the associate editor, we have made only minor changes in response to the three reviews. We thank the associate editor for his gracious handling of this paper, which has elicited significant debate, largely surrounding the tectonic history of the Sierra Nevada. As requested by the associate editor, we added a paragraph to the discussion that summarizes this debate.

**Response to Reviewers**

Beeson and McCoy, MS No. esurf-2019-24

Response to Anonymous Referee #1

*Page 21, lines 6-7: Should this say "the vertical bed of less erodible rock"?*
Yes. Corrected.

*Page 25, line 26: mainstem*
Corrected.

*Page 26, lines 3-5: To what part of Figure 8 is this sentence referring? The river flowing toward the upper left that gets captured?*
Corrected to (Fig. 8 c and d)

*Figure 16: I'm a little confused by the pre-capture flow path label, it looks like it was originally flowing in the opposite direction?*
We added the following sentence to the capture of Fig. 16: "The proposed pre-capture flow path is along the tributary with a noted wind gap, suggesting flow reversal along this short length."

*Page 44, lines 23-24: This 3 Myr figure is dependent on parameter values, right? Maybe should specify that 3 Myr was specific to your model runs?*
We added the bold text to the following sentence: "Following continuous rapid tilting, the magnitude of tilt is reflected in the river network only briefly (at t=3 Myr) and only with a higher-magnitude tilting rate 1 degree vs. 0.5 degree), **however, this response time is dependent on the chosen parameter values in the model.**"

Response to Anonymous Referee #4

*My main issue with this study is that of equifinality: many of the diagnostic features that the authors attribute to tilting could be the result of different processes, or simply in the choice of how you extract the channel net- works and calculate χ. For example, in real landscapes the location of the channel head will have a big impact on the resulting χ plots. Performing χ transformation above the channel head will lead to non-linearity of the chi plot which may result in a very similar looking plot to the scenario of higher uplift at the outlet shown in Fig 1d (see Clubb et al. (2014), Fig 8). For the model runs this isn't a big deal, but if you chose an unrealistically low area threshold in a real landscape this could look the same as a spatially variable uplift field. This is not to say that this invalidates the study, but I think the authors could more carefully consider and elaborate on other reasons for non-linearity within χ plots. Other issues could be the influence of glaciation or mass-wasting processes, although I'm not aware of work at present which explores how these processes might influence χ plots.*

*A related, and more important problem, is that varying the concavity value used to calculate the χ plots can lead to very different curvatures: changing θ from 0.2 - 0.8 will completely change the curvature of the tributaries from convex to concave. I ran the chi analysis quickly on the 90 m SRTM data from the*

*Middle Fork American River site in the Sierra Nevadas (Figure 1) which demonstrates this effect. The authors state in their discussion (Page 47, Lines 26 - 30) that changes in m or n might violate their assumption of uniform concavity. However, even if there is uniform concavity, if you have selected an incorrect m/n value for the basin, this could lead to an identical signature in the χ plot as that of a rigid block tilting scenario. I am concerned that the conclusions of the paper might lead other studies to assume that negative curvature in χ elevation space is always a result of tectonic processes, when in fact it could be a simple artefact of how χ was calculated.*

These are good points. We changed various parts of the paper to try to stress that it is the suite of geomorphic signatures that is unique tilting, rather than any signature alone. To emphasize this point, we added the bold text to each section noted below:

To the abstract:
"Using a model river network composed of linked 1-D river longitudinal profile evolution models, we show that the transient response to a punctuated rigid-block tilting event creates **a suite of** characteristic forms or geomorphic signatures in mainstem and tributary profiles that **collectively** are distinct from those generated by other perturbations such as a step change in uniform rock uplift rate or major truncation of headwater drainage area that push a river network away from equilibrium."

To the beginning of the discussion:
"In the analysis presented above, we show that the fluvial response to the simple case of a punctuated rigid-block tilting event does inscribe **a suite of identifiable geomorphic forms throughout the river network during the transient evolution back towards an equilibrium state that collectively comprise a robust signature of tilt.**

We expanded a paragraph of the discussion to explicitly state this problem and to include the two scenarios you bring up that would create negative curvature:
"**Certain systematic changes in process or mistakes in data analysis can also create chi plots that could be mis-interpreted as reflecting a signature of tilt.** A shift in $m$ and/or $n$ owing to a change in erosional mechanism (Whipple et al., 2000) or a transition from detachment-limited to transport-limited between tributaries and mainstem rivers or along mainstem rivers would violate our assumption of uniform concavity and **could generate negative curvature and/or tributaries with lower steepness than mainstems** that would simply reflect a change in process rather than tectonic forcing. **Similarly, extending chi analysis above the channel heads results in chi profiles with negative curvature that would also only reflect a change in process (e.g., Clubb et al., 2014). Choosing too high of a reference concavity would result in negative curvature in chi plots as well. This highlights that care must be taken when interpreting tectonic histories from landscape form.** Tilting generates a suite of signatures that collectively are robust and unique, but individual signatures share characteristics with those produced by other perturbations described herein. Furthermore, constraining initial conditions or teasing apart multiple perturbations presents a major challenge. Thus, it could be difficult to recognize signatures of tilt in river profiles and network patterns if tilting is not the predominant perturbation recorded in landscape form or if erosional mechanisms are nonuniform."

Although we agree this is a good point to make in the paper, we note that in the figure you presented of chi analysis performed with a range of reference concavities for the Middle Fork American River, many of the first order signatures of tilt are retained at all concavities. Specifically, the positive-curvature slope-break knickpoint in the mainstem occurs at all concavities and a negative curvature knickpoint occurs at the upper end of the tributary knickzones at all concavities (or, in other words, the chi plots for the tribs roll over). Furthermore, upstream of the mainstem knickpoint, the mainstem chi plot is either relatively straight or with negative curvature - at no concavity does the upper profile have positive curvature.

*There are a lot of different scenarios presented in the paper which were not related to tilting (e.g. change in elevation of the channel profile, drainage capture, uniform change in uplift rate). A lot of these results, and the discussion of knickpoint celerity, are not new to this paper and are similar to the work of other authors (Royden and Perron, 2013; Willett et al., 2014). I would suggest to either remove these analyses to focus on the novel points of this paper (the tilting scenarios), or to put these results more in context of the work that has already been done.*

We added the following two sentences to the section **Modeling fluvial longitudinal profile response to perturbations:** "Although the transient response to both uniform rock uplift/base level fall and step changes in uplift rate or erodibility have been extensively researched (e.g., Baldwin et al., 2003; Bonnet and Crave, 2003; Grimaud et al., 2016; Howard, 1994; Rosenbloom and Anderson, 1994; Royden and Perron, 2013; Tucker and Whipple, 2002; Whipple and Tucker, 1999), we include these perturbations herein to provide comparisons between well-known transient responses and those induced by tilt or truncation. We hope that this comparison highlights the fact that many perturbations disrupt co-linearity of mainstems and tributaries and that examination of the relationship between mainstems and tributaries can facilitate reconstruction of tectonic histories."

*For the real landscapes, the authors state that they used slope-area plots to estimate θ. I found this odd, as the noise present in SA plots often makes the calculation of concavity challenging, and is one of the reasons that χ plots were initially developed. Why did the authors not use a χ based method of calculating concavity? I think the statement that 'Standard techniques for finding θ cannot be applied given the disequilibrium state of Sierra rivers' (Page 33, Line 8) is unsupported - there have been many techniques that have developed ways of estimating concavity in transient systems which are not cited in the study (e.g. Mudd et al., 2018; Hergarten et al., 2016).*

We changed the relevant paragraph under the subsection *Disequilibrium form of the mainstem Middle Fork American consistent with late Cenozoic tilting* to read:

**"Estimating the regional equilibrium river concavity or the reference concavity, $\theta_{ref}$ is challenging in landscapes with transient river networks as both standard techniques using slope-area plots (e.g., Wobus et al., 2006) or chi plots (Perron and Royden, 2013) of entire basins cannot be applied given the disequilibrium state of Sierra rivers. Although more advanced techniques of estimating $\theta_{ref}$ from chi plots have been developed (e.g., Mudd et al., 2018), these methods still require the assumption of co-linearity, which as we show above, is not retained in all transient responses. Therefore,** we used slope-area plots of mainstem rivers downstream of slope-break knickpoints to estimate that $\theta$=0.41-0.48, and thus we use $\theta_{ref}$ =0.45 for all analyses in the Sierra. In doing so we assume that equilibrium river concavity is uniform throughout the river network in the Sierra Nevada, but not that modern river concavity is uniform."

*Page 2, Line 10: I would suggest to also cite Morisawa (1962) here, as it reports slope versus drainage area prior to that of Flint and is often missed out in subsequent literature for some reason.*
Good to know! We added a citation to Morisawa (1962).

*Page 3, Line 1: change to 'documented across many tectonic settings'*
We changed the line accordingly.

*Page 3, Lines 18 - 22: were all these scenarios with the same tilt angle? It would be good to clarify this*
We added the following sentences to the third paragraph of the section **Modeling fluvial longitudinal profile response to perturbations**
"We used different tilt angles for each of the simulations of instantaneous tilt in different directions (forward, back, and lateral tilting) because using the same high tilt angle for back tilting as we used for forward tilting resulted in river reversal but did not result in significant rock uplift at the channel head in

lateral tilting owing to the much shorter length of tributaries compared with the mainstem river. Therefore, we used a lower tilt angle for back tilting and a higher tilt angle for lateral tilting."

*Page 3, Line 30: very long sentence, split up.*
We split it up as follows:
"Lastly, we document the expression of geomorphic signatures of a punctuated rigid-block tilting event proposed to have occurred in the Sierra Nevada of California, USA (Huber 1981; Jones, 2004; Lindgren, 1911; Unruh, 1991; Wakabayashi, 2013) and onset of rapid continuous tilting in the Sierra San Pedro M\'artir of Baja California (Rossi et al., 2017). We use these field examples to demonstrate how signatures of tilt in river profiles and river networks can be applied to estimate the timing and magnitude of tilt in both these regions, but we stress that in neither case do we consider our analysis to be a robust reconstruction of the regional tectonic histories owing to the analysis of only a single river basin."

*Page 5, Line 10: typo in subheading - repetition of χ symbol and misspelling of 'transience'*
We changed the heading to: **χ transformed river profiles for identifying equilibrium and transient forms in river profiles**

*Page 5, Line 21: Replace 'curved' with 'concave-up'*
Replaced.

*Page 5, Lines 24 - 26: This is only the case if the correct m/n ratio is chosen for the channel. I think it is worth mentioning this caveat.*
We added in the text in bold:
"Tributaries in equilibrium with the same uplift rate and with the same erodibility as the mainstem will be co-linear with each other as well as with the mainstem such that all portions of an equilibrium river network collapse towards a single straight line on a chi plot, **provided the correct reference concavity ($m/n$) has been chosen and the analysis has been limited to the fluvial portions of the network (Clubb et al., 2014; Perron and Royden, 2013).**"

*Page 6, Line 25: more recent studies have since demonstrated that concavities outside 0.4 - 0.7 are common and that n is commonly greater than 1 (Lague, 2014; Harel et al., 2016).*
We modified the sentence in the "Parameter values" section to say:
"Although the slope exponent, *n*, has been shown to be commonly greater than unity from relationships of channel steepness with erosion rate (DiBiase et al., 2011; Harel et al., 2016; Lague, 2014; Ouimet et al., 2009), data on knickpoint propagation is best explained with *n* =1 (Lague, 2014), and mechanistic approaches yield estimates ranging between 2/3 and 5/3 (Whipple et al., 2000; Larimer et al., 2019). Given the uncertainty in the value of n and the simplicity of the *n* =1 case, we assume *n* =1 for all simulations and analyses. However, we also ran 1-D simulations with both n=2/3 and n=5/ and present these results in the supplement."

Lague, 2014 shows that theory predicts that equilibrium concavities should fall in the range of 0.4-0.6. The Harel, 2016 study does not attempt to determine whether the basins in question were in steady state and thus the concavities outside this range cannot be interpreted as equilibrium (reference) concavities towards which profiles are evolving. Therefore, we left this discussion point out of the paper.

*Page 9, Line 7: remove word 'very'*
Removed

*Page 9, Lines 7 - 17: I found myself getting confused by all these scenarios. Would it be possible to put this in a table to make it clearer?*
We added a table listing all the scenarios, their background uplift fields, and the associated figures.

*Page 10, Line 1: why equal to five? This could be explained more clearly.*
We expanded the explanation to read:
"With $K=1x10^{-6}$ and uniform, chi values of the vertical-step knickpoint can be interpreted as knickpoint travel times, $\tau$, in millions of years. For example, at $t=5$ Myr, the vertical-step knickpoint is located at a chi value equal to five, which with $K=1x10^{-6}$ and uniform corresponds to $\tau=5$ Myr"

*Page 10, Line 5: repetition of 'in the'*
Removed

*Section 4.2: Which scenario was actually ran? Increase in K or decrease in U?*
We added the following sentence to the beginning of the section to clarify: "A step decrease in fluvial relief can be accomplished by increasing bedrock erodibility or decreasing uniform rock uplift rate. We ran both simulations using different parameters to generate equilibrium initial conditions but the same parameters following each step change such that, following perturbation, the profiles in both simulations were evolving towards the same equilibrium profile and as such the transient response was identical between the two simulations."

*Figure 2: what timescale does the dashed blue line represent? Is this the same as the response time $\tau$?*
It represents the full profile at any time step greater than the channel response time at the channel head, $\tau$. But, it also represents any portion of the profile that has equilibrated to the boundary conditions and as such we simply refer to it as the equilibrium line rather than the profile at a specific time.

Response to review by E. Gabet

The arguments presented in this review are exclusively about our interpretation of the Sierra Nevada field site, which Gabet takes issue with. In this paper we presented analysis for a single river basin, but in a forthcoming paper we have presented the same analyses for all major rivers draining the western slope of the Sierra. Therefore, this is not the best location for this debate given there is limited data to point to.

Below we provide a brief summary of the debate followed by his specific comments (italicized) and our responses (regular text).

**We seek a mechanism that can explain the following measurements:**
The topography of the northern Sierra Nevada consists of deeply incised mainstem rivers separated by broad, low relief upland surfaces. Incision (measured either below Cenozoic deposits or as elevation below upland surface) varies greatly both along individual mainstem rivers and between adjacent rivers at similar distances upstream. Catchment-averaged erosion rates from upland catchments are on the order of 0.01 mm/yr (Callahan et al., 2019; Hurst et al., 2012; Riebe et al., 2001) and erosion rates calculated from bare bedrock surfaces are on the order of 0.001-0.01 mm/yr (Stock et al., 2005). In contrast, stream incision rates into basement rock measured below Cenozoic deposits are on the order of 0.1 mm/yr (Wakabayashi, 2013) and catchment-averaged erosion rates of small sub-basins within the incised portions of mainstems are also on the order of 0.1 (up to 0.25) mm/yr (Hurst et al., 2012).

**Summary of Gabet's arguments**
Gabet's point of view comes from his observations that channel steepness varies with rock type in the Sierra, which he uses to argue that lithology is the primary control of river profile form in the Sierra Nevada and not late Cenozoic tectonics (Gabet, 2019). In his comments on our manuscript he argues that

the disequilibrium forms we identify in the profiles of the Middle Fork American are all related to lithology and that the knickpoint is a static knickpoint.

**Summary of our rebuttal**

We agree that lithology plays an important role in landscape evolution in the Sierra Nevada, but while there are many locations where channel steepness does vary at lithologic boundaries, there are also many locations where it varies little. In addition, there is large variability in channel steepness of any given formation, particularly if drainage areas <100 km$^2$ are not excluded, as Gabet (2019) has done. This suggests that what is going on in the Sierra is more complex than simply reaches with different equilibrium steepness. Gabet acknowledges these observations in the following section in his 2019 paper:

"There are exceptions to these general trends, however. For example, a reach of the NF Mokelumne River flows over granite with the same gradient as the adjacent reach flowing over chert and argillite (km-80–90; Fig. 5), and, on the Merced River (km-20; Fig. 6A), a short reach underlain by granitic rock has a similar gentle gradient to the adjacent downstream reach flowing across phyllite and chert. This latter example is striking because the granitic bedrock lining this gentle reach also lines the steepest reach in the profile (km-15–20). The presence of these low-gradient granitic reaches at lower elevations suggest that they might have been exposed more often to subsurface water than their counterparts at higher elevations and, therefore, the bedrock may have been weakened by chemical weathering (Callahan et al., 2019; Wahrhaftig, 1965)."

In addition to providing an incomplete explanation for the spatial patterns in channel steepness, Gabet's theory also cannot explain the order of magnitude difference in erosion rates between mainstem canyons and upland surfaces, the observed variability in canyon incision, or the disequilibrium forms in river profiles (negative curvature in the mainstem and tributary knickzones that collapse with the mainstem).

**Response to individual comments:**

*With respect to the analysis of the Mexican site, I am not familiar with the region and was unable to procure a geologic map of the area. However, I am concerned that, although the analyses yield results with respect to tilt magnitude and timing, there doesn't appear to be any data with which to validate them. Usually, when new approaches are developed, their results are tested against known data – this provides confidence in the new approach so that it can be applied to new areas, and it gives us an idea of how accurate it is. In this case, a new approach is being applied to a new area and so an important step is being skipped. I would recommend applying these techniques first to a site where the tectonic history is well-known. Until that is done, we don't have any way of scientifically assessing the validity of this approach or the robustness of the assumptions (eg, uniform erodibility).*

Timing and magnitude are constrained for the eastern side of the range, which is where the active normal fault system is that would provide uplift for the crest and hence tilt of the western slope. Below is copy and pasted from our paper:

"Rossi et al. (2017) used topographic analysis combined with 10Be-derived denudation rates in river basins on the eastern side of the range to show that the fault system that borders the eastern boundary of the range initiated during the mid-Miocene and increased in slip rate in the late Pliocene up to a mean of 130 mMyr$^{-1}$."

*To motivate the analyses done in the Sierra, the authors cite several papers that they claim support the hypothesis of Late Cenozoic tilting; however, these studies have all been debunked. Below is a brief synopsis of the fatal flaws in each of them; more detailed explanations can be found in Gabet (2014). If the authors would like to cite these papers, they will need to explain why the analyses of these fundamental flaws are incorrect; otherwise, there's not much value in referring to discredited studies.*

I'm not aware of any code of ethics that one needs to defend a paper against those who have different interpretations of it in order to cite them. These are all good papers with valid observations and hence we cite them.

*Lindgren (1911) based his tilt estimate with the assumption that he correctly reconstructed the Tertiary paleochannels. Both Gabet (2014) and Cassel (2012) demonstrated that his reconstructions were fundamentally flawed, either because they imply that water flows uphill or because he was linking channel segments that were unrelated in time and space. Moreover, Gabet (2019) demonstrates that, even if Lindgren had correctly reconstructed the channels, the differences in their gradients that he attributed to tilting can be wholly explained by differences in bedrock erodibility. Jones et al (2004) based their analyses on Lindgren's reconstructions. Since those reconstructions are faulty, their analyses are meaningless. In fact, the reconstruction of the South Yuba River in Jones et al has reaches where water flows uphill (an absurd result). Moreover, like Lindgren, Jones et al ignores the role of erodibility in controlling channel slope.*

There may be some errors in the reconstructions, but Gabet (2019) did not provided an alternative explanation for why there is a strong azimuth dependency that is well fit by a sine function. He simply showed that rock have different susceptibility to erosion.

*Wakabayashi (2013) calculated bedrock incision depths and rates from Pliocene volcanic deposits along the rims of canyons without recognizing that older deposits can be found farther down the canyon walls.* Wakabayashi (2013) also showed that in the vast majority of cases the older Eocene-Oligocene deposits are found below, yet quite close in elevation, to the base of the Pliocene volcanic deposits and hence using these two datums comparable incision depths into basement rock are found. In Gabet (2014), you point to an older deposit that is near to river elevation in the S. Yuba, but this deposit is in a fault zone and is hard to interpret. The one Pliocene volcanic deposit very near the river elevation in the upper reaches of the South Fork American in Gabet (2014) is well up stream of the slope-break knickpoint where total incision in expected to be less and does not dismiss the observations made by Wakabayashi (2013) that there has been significant incision into basement rock in the majority of rivers.

*Unruh (1991) based his tilt estimate of 1.4° on the gradient of Central Valley sediments. By using simple geometry, one can demonstrate that a consequence of this result is the prediction that, at some point in the mid-Cenozoic, Tertiary gravels at an elevation of ~700 m along the Yuba river were once ~500 m below sea level. In other words, untilting the northern Sierra by 1.4° places sections of the mountain range deep underwater sometime in the past 30-50 my. This study is refuted, therefore, by the absence of deep Cenozoic marine sediments in the Sierra.*

His observations that Central Valley sediments are tilted (with younger deposits progressively more tilted) is still valid and a convincing argument for late Cenozoic tilt, regardless of whether his method resulted in overestimates and/or these estimates should not apply range wide.

*In this section, the authors describe the profiles from a series of numerical simulations that are then compared to the Middle Fork American River. Their model, based on the simple streampower formulation, is used to show that tilting leads to knickpoints that migrate up from the range-front. However, this result is obtained by making the extraordinary claim that rock erodibility is uniform throughout the area. In fact, rock erodibility is extremely non-uniform throughout the range and is the primary control on channel steepness (Gabet, 2019). Shown below is the profile of the North Fork Feather River demonstrating the important role of lithology and erodibility on profile shape. Note how nearly every knickpoint is associated with a lithological boundary and how the steepness index (the second number above each reach) varies according to rock unit, even from one granitic unit to the next.*

We do not make a claim that rock erodibility is uniform throughout the area. In contrast, we show how nonuniform rock type might modulate knickpoint migration and incision. This mechanism can easily explain the observed variability in incision both along individual basins and between them. Importantly

the transient nature of the river profiles can explain why there are large variations in steepness in the transient reaches upstream of our slope-break knick points, while downstream in the near-equilibrium reaches variation in steepness across rock types is much more muted.

*In this section, the authors estimate tilt based on the geometry of a river profile that flows across weak rock that is sandwiched between strong rock. There were a number of issues here.*
*(1) To develop their model (Fig. 7), the authors are assuming uniform uplift in the northern Sierra Nevada (p. 23, l. 7,8). There is, however, no evidence for uniform uplift in the northern Sierra Nevada during the Cenozoic. The authors are the first to ever make such a claim and they do so without providing any evidence. Uniform uplift would create an obvious fault scarp hundreds of meters high along the entire western range- front, a feature which has not been observed.*
Never do we make a claim that the northern Sierra is experiencing uniform uplift. In fact we show four tilting simulations in which the background uplift is nonuniform and the same knickpoint migration and spatial patterns in incision are observed. The transient response to tilt is the same with initial conditions equilibrated to nonuniform or uniform uplift, so there is no reason to believe that the transient response to tilt with heterogeneous lithology would be any different with initial conditions equilibrated to a low magnitude tilt.

*(2) This technique is dependent on the assumption that the reach formed during the transient response to tilt is similar to the initial steady-state river slope. The authors, however, have not provided any evidence to support the claim that the pre-tilt river was at steady-state. Given the multiple episodes of aggradation, incision, and drainage reorganization experienced by these rivers due to repeated volcanic eruptions, the odds that this assumption is correct are vanishingly small.*
That is the assumption stated for the model, but this method does not require that the pre-tilt river was at steady-state, but only that the quasi-equilibrium reach is similar to the initial river slope. There are many examples in our other paper that are focused exclusively on these signatures in the Sierra that demonstrate how this mechanism can explain the observed variability in mainstem incision both along individual rivers and between basins.

*(3) The authors are assuming specific erodibility values for the rock units at this site along the Rubicon River without any evidence that these values are accurate. They are using a K value determined from a different granitic bedrock unit in the southern Sierra Nevada and applying that to the Rubicon site without accounting for the fact that erodibility can vary greatly in granitic rocks (Gabet, 2019). For the Jurassic marine rocks, which the authors take as the "erodible" unit, they are assuming that it is 10x more erodible than the granitic rocks but, again, without any evidence. In fact, the Jurassic marine rocks include quartzite, which is as strong as granitic rocks, and greywacke, which is also very strong and certainly not 10x weaker than granitic bedrock (Gabet, 2019).*
We do not assume any specific erodibility values for the rock units at the site along the Rubicon River. Furthermore, in the relevant modeling section we show that the method works when the more erodible rocks are only 2x more erodible than the flanking rocks, but yields an underestimate.

*(4) Finally, the field site does not conform to their model. Below (left) is their figure illustrating their model and (right) the actual profile. Note how, in the model, the dashed line extends smoothly from the profile at the top of the knickzone down to the lower extent of the "erodible" rock. In the real river, extending the profile above the knickzone in a similar manner yields a completely different geometry.*
No part of this method relies on extending the river profile above the more erodible rock to portions downstream, as you suggest that it does, but rather on the geometry of the incised region.

*6.1 Disequilibrium form of the mainstem Middle Fork American (p. 31)*

*In this section, the authors identify a knickpoint in the Middle Fork American River and assume that it is a migrating knickpoint generated by uplift (black triangle in the profile below). This knickpoint, however, is associated with both a lithological and structural boundary.*

*Although the authors have combined the metamorphic rocks (in green) directly above and below the knickpoint into a single unit, this obscures the fact that they are different formations with different lithologies (note that the pink band of granitic rocks that they show immediately above the knickpoint does not exist on any map that they cite). I have labelled these formations as CC and SF on their figure; I have also added a red dotted line to show the location of the Volcano Canyon Fault that forms the contact between the two (ignoring, as the authors have, smaller units at that site). The metamorphic unit on the upstream side of the knickpoint is the Paleozoic Shoe Fly Formation (SF), which is composed of resistant quartzite and metavolcanic rocks. The unit on the downstream side of the knickpoint is the Paleozoic Calaveras Complex (CC), a highly sheared subduction melange that includes weaker argillite and chert. From Gabet (2019), the steepness index (a measure of erodibility) of the Shoe Fly Formation is 0.13 while, for the Calaveras Complex, its average is 0.07 (note, these values are from other rivers and are, therefore, independent of the particular situation on the Middle Fork). The difference in the steepness index between the two units is strong evidence that this is a lithological knickpoint and, moreover, that the assumption of uniform erodibility is violated.*

[Figure]

*In addition, their profile does not show that the area downstream of the knickpoint is within the Foothills Fault System where many faults have made the bedrock more erodible (Gabet, 2019) – I have added a label and a bracket to their figure to show the extent of this fault zone. The figure below shows the lithological map of this area where the faults and the different units can be seen (a yellow line across the Middle Fork marks the knickpoint).*

Actually, the knickpoint on the MFA occurs upstream of mapped faults and is not at a lithologic boundary. Here is the knickpoint plotted on the USGS 1:250k Sacramento Quad Map. I have also labeled the granodiorite that you claim *"does not exist on any map that they cite"*. It can also be found on the 1:500k that was used for the profiles at: https://mrdata.usgs.gov/geology/state/state.php?state=CA

[Figure]

*Another piece of evidence indicating that this knickpoint is not a migrating feature comes from the surrounding hillslopes. As demonstrated in Hurst et al. (2013), hillslopes just downstream of a knickpoint will be steeper than those upstream. However, in the slope map below, where the black line shows the location of the knickpoint and the river flows from right to left, the hillslopes are actually steeper upstream of the knickpoint (red is steep, blue is gentle). This slope map, therefore, not only contradicts the assumption that this is a migrating knickpoint but, instead, it supports the conclusion that the reach of river upstream of the knickpoint is steeper because the rocks are stronger.*

[Figure]

The slope map very clearly demonstrates a transient response along the entire part of the river canyon that is shown – steep slopes in the canyon and very gentle slopes outside of the canyon. This is exactly the response expected in a tilted landscape. The approach in Hurst et al. (2013) requires that the change in tectonic or climatic conditions is transmitted solely by a knickpoint propagating upstream. In the case of tilting, channel steepness is everywhere affected by the perturbation and thus there is no reason to assume there would be a hysteresis in landscape response.

The steep hillslopes shown in the figure seem to reflect canyon incision along the entire profile, with slightly less steep slopes downstream of the knickpoint, exactly as expected from the predicted lower stream incision rates below the knickpoint as compared to above. Hence, this would actually be consistent with the occurrence of the positive-curvature slope-break knickpoint we describe: downstream of the knickpoint, the mainstem channel has equilibrated to a lower uplift rate, whereas, upstream of the knickpoint, the channel is steeper than equilibrium and thus still incising at a rate that is faster than the background uplift rate. The hillslopes adjacent to the mainstem canyon respond to this rapid incision rate by steepening.

*Based on the identification of the knickpoint as a migrating feature, the authors use its location to make estimates about the timing of uplift. To make this estimate, the authors assume that erodibility is uniform along the lower part of the profile that the knickpoint travelled across; moreover, they assume that the erodibility of those rocks is the same as granitic rock in the southern Sierra Nevada (p. 31, l. 25). Both of these assumptions are contradicted by the field evidence. In the map below, the section of the Middle Fork below the knickpoint begins at the top left corner and then flows diagonally down towards the northern part of Folsom Lake, through the lake and into the Central Valley. Along this path, it crosses a dozen different geological units, and only one of these is granitic while the rest are various types of metasedimentary and metavolcanic rocks, including a highly sheared subduction melange. As demonstrated in Gabet (2019), the erodibility of these rock types vary greatly; for example the steepness index of the subduction melange is about half that of granitic rock. Furthermore, the Middle Fork crosses six faults, which means that the rocks at those locations will be much weaker than at other spots (Gabet, 2019). Therefore, in addition to problem of assuming that the knickpoint is a migrating feature, the two other assumptions necessary for calculating the timing of uplift are violated.*
We aren't trying to predict knickpoint velocity through these different formations, we are only trying to estimate the timing of cessation of tilt. As you demonstrate, it is more likely that the mean erodibility of these rocks is less than that of granite. Therefore, our estimate of 3 Ma if anything is an overestimate and the perturbation is in fact younger. We take the conservative approach of using the lower erodibility and estimating that perturbation is older. However, as we say throughout the paper, we do not think estimates from a single river basin can provide a robust reconstruction of tectonic history.

*Finally, in this section, the authors estimate the magnitude of recent tilt based on alleged incision depths beneath Mio-Pliocene volcanic deposits (in brown below) and Eocene fluvial gravels (in red). The estimates of incision depths, however, are contradicted by the field evidence. For example, at the site just above the black triangle on the plot above, the authors are claiming that there has been ~600 m of incision since the Miocene-Pliocene (see the regression line) and that this has been driven by recent uplift. However, the elevation of the Eocene deposits proves that there could not have been more than ~450 m of incision since the Eocene. In the illustration to the left, I show the present day profile across the Middle Fork at the top, Beeson and McCoy's interpretation of this profile (left column), and the standard interpretation to the right. The critical point is that the authors are claiming that the Mio-Pliocene volcanics represent a bedrock paleosurface and that all of the relief below the volcanics is due to Pliocene incision into basement rock. However, their interpretation cannot explain the presence of Eocene deposits below the Mio-Pliocene deposits. The only interpretation consistent with the field evidence is that the Eocene gravels represent the bedrock surface during the Eocene when this canyon was already at least 150 m deep. This landscape was then buried by volcanic deposits that are known to have been hundreds of meters thick. To put it bluntly, because 450 < 600 and because the Eocene is older than the Mio-Pliocene, the estimate for recent tilt based on the regression line in the incision plot is invalid. The only accurate statement that can be made is that there's been a maximum of ~450 m of incision since the Eocene – Early Oligocene.*
Note that the x-axis on the incision depth inset is Euclidean distance from the mountain front, not distance upstream. The location with deposits at 450 m occurs upstream of the mainstem knickpoint. To make these measurements, we used a swath profile that was centered on the mainstem river. This means that, at

any given river node, the swath may pick up deposits up to 7.5 km at either greater or lesser Euclidean distances from the mountain front owing to sinuousity of the river channel. Although this method introduces variability in incision measurements, each river node is then associated with elevations of the nearest deposit. This is superior to the alternative method of associating each river node with deposits that are at the same Euclidean distance from the mountain front but may not be the nearest deposit. We chose to regress through these data rather than use a minimum envelope curve that would incorporate the lowest measurements to account for the variability in incision measurements introduced by our method of snapping swaths to sinuous river profiles and the fact that only small-scale digitized geological maps were available and thus deposit elevations may not be precise.

*The use of the Mio-Pliocene volcanic deposits on the interfluves as an indicator of a paleosurface was promoted in papers by Wakabayashi (Wakabayashi and Sawyer, 2001; Wakabayashi, 2013) and this method has since been debunked as older sediments can be found far below them (Gabet, 2014).*
As I said above, Wakabayashi (2013) also showed that in the vast majority of cases the older Eocene-Oligocene deposits are found below, yet quite close in elevation, to the base of the Pliocene volcanic deposits and hence using these two datums comparable incision depths into basement rock are found. In Gabet (2014), you point to an older deposit that is near to river elevation in the S. Yuba, but this deposit is in a fault zone and is hard to interpret. The one Pliocene volcanic deposit very near the river elevation in the upper reaches of the South Fork American in Gabet (2014) is well up stream of the slope-break knickpoint where total incision in expected to be less and does not dismiss the observations made by Wakabayashi (2013) that there has been significant incision into basement rock in the majority of rivers.

*Because the volcanic rocks are remnant patches left on the ridges, the plot above is simply showing that relief increases from the range-front and then decreases as the crest is approached, as shown in (Gabet, 2014); compare the incision pattern of the volcanic rocks in the plot to the left with the plot of relief below (solid line).*
This comment is identical to one on the open discussion forum https://doi.org/10.5194/esurf-2019-24-SC6 which we already responded to in https://doi.org/10.5194/esurf-2019-24-AC5

*6.3 Tilt magnitude recorded in the stream network* (p. 35*) In this section, the authors use a relationship between reach azimuth and gradient to look for signatures of tilt. Based on their analyses, the authors conclude that there has been 2.3° of recent tilt. Their technique can be tested using field observations. As noted by Huber (1981), the upper uneroded surface of a 10 Myo lava flow along the San Joaquin River (which is only 5-10 km north of where the authors did one of their analyses) forms a series of table mountains. The source of this flow was the Sierra Nevada (top-right of the map shown to the left) and it flowed down into the Central Valley (bottom- left of the map). The line in the figure to the left shows the transect plotted on the next figure. The upper surface of the flow is at an angle of 1.37° (first figure below). If we subtract the 2.3° of recent tilt hypothesized by the authors, the upper surface of the flow is now at -0.9° (ie. 1.37 – 2.3 = -0.9; second figure below). This means that, if there had been 2.3° of recent tilting (as the authors claim), this lava would have flowed uphill 10 Mya. This result challenges the approach presented here. The authors state that their results are a "maximum possible tilt magnitude (p. 36, l. 7)". However, the table mountains data show that the maximum is actually ~1.4°, indicating that their technique is, in fact, unable to constrain the maximum amount of tilt. Moreover, without providing any bounds on the minimum possible tilt magnitude, their results are consistent with 0° of tilt. Therefore, this analysis isn't providing any new information. At a minimum, a few other nearby sites should be analyzed because a sample size of 1 is too small to provide confidence in this technique. This would have the benefit of providing some measure of the potential error.*
This analysis shows that the river network reflects a westward tilt. We show that it works in the landscape evolution model and we provide more examples in a forthcoming paper.

[revised manuscript text omitted]

---

## Author Response (AR3)

**Author's response**

We added explicit citations to the interactive comments and replies per request by the Associate Editor. We thank the Editor Dr. Niels Hovius and 
[revised manuscript text omitted]